# The genomic origins of the Bronze Age Tarim Basin mummies

Fan Zhang[1,15], Chao Ning[2,15 ✉], Ashley Scott[2,15], Qiaomei Fu[3], Rasmus Bjørn[2], Wenying Li[4], Dong Wei[5], Wenjun Wang[3], Linyuan Fan[1], Idilisi Abuduresule[4], Xingjun Hu[4], Qiurong Ruan[4], Alipujiang Niyazi[4], Guanghui Dong[6], Peng Cao[3], Feng Liu[3], Qingyan Dai[3], Xiaotian Feng[3], Ruowei Yang[3], Zihua Tang[7], Pengcheng Ma[1], Chunxiang Li[1], Shizhu Gao[8], Yang Xu[1], Sihao Wu[1], Shaoqing Wen[9], Hong Zhu[5], Hui Zhou[1], Martine Robbeets[2], Vikas Kumar[3], Johannes Krause[2,10 ✉], Christina Warinner[2,11 ✉], Choongwon Jeong[12 ✉] & Yinqiu Cui[1,13,14 ✉]

The identity of the earliest inhabitants of Xinjiang, in the heart of Inner Asia, and the languages that they spoke have long been debated and remain contentious[1]. Here we present genomic data from 5 individuals dating to around 3000–2800 BC from the Dzungarian Basin and 13 individuals dating to around 2100–1700 BC from the Tarim Basin, representing the earliest yet discovered human remains from North and South Xinjiang, respectively. We find that the Early Bronze Age Dzungarian individuals exhibit a predominantly Afanasievo ancestry with an additional local contribution, and the Early–Middle Bronze Age Tarim individuals contain only a local ancestry. The Tarim individuals from the site of Xiaohe further exhibit strong evidence of milk proteins in their dental calculus, indicating a reliance on dairy pastoralism at the site since its founding. Our results do not support previous hypotheses for the origin of the Tarim mummies, who were argued to be Proto-Tocharian-speaking pastoralists descended from the Afanasievo[1,2] or to have originated among the Bactria–Margiana Archaeological Complex[3] or Inner Asian Mountain Corridor cultures[4]. Instead, although Tocharian may have been plausibly introduced to the Dzungarian Basin by Afanasievo migrants during the Early Bronze Age, we find that the earliest Tarim Basin cultures appear to have arisen from a genetically isolated local population that adopted neighbouring pastoralist and agriculturalist practices, which allowed them to settle and thrive along the shifting riverine oases of the Taklamakan Desert.

As part of the Silk Road and located at the geographic confluence of Eastern and Western cultures, the Xinjiang Uyghur Autonomous Region (henceforth Xinjiang) has long served as a major crossroads for trans-Eurasian exchanges of people, cultures, agriculture and languages[1,5–9]. Bisected by the Tianshan mountains, Xinjiang can be divided into two subregions referred to as North Xinjiang, which contains the Dzungarian Basin, and South Xinjiang, which contains the Tarim Basin (Fig. 1). The Dzungarian Basin in the north consists of the Gurbantünggüt Desert, which is surrounded by a vast expanse of grasslands traditionally inhabited by mobile pastoralists. The southern part of Xinjiang consists of the Tarim Basin, a dry inland sea that now forms the Taklamakan Desert. Although mostly uninhabitable, the Tarim Basin also contains small oases and riverine corridors, fed by runoff from thawing glacier ice and snow from the surrounding high mountains[4,10,11].

Within and around the Dzungarian Basin, pastoralist Early Bronze Age (EBA) Afanasievo (3000–2600 BC) and Chemurchek (or Qiemu'erqieke) (2500–1700 BC)[12] sites have been plausibly linked to the Afanasievo herders of the Altai–Sayan region in southern Siberia (3150–2750 BC), who in turn have close genetic ties with the Yamnaya (3500–2500 BC) of the Pontic–Caspian steppe located 3,000 km to the west[13–15]. Linguists have hypothesized that the Afanasievo dispersal brought the now extinct Tocharian branch of the Indo-European language family eastwards, separating it from other Indo-European languages by the third or fourth millennium BC (ref. [14]). However, although Afanasievo-related ancestry has been confirmed among Iron Age Dzungarian populations (around 200–400 BC)[7], and Tocharian is recorded in Buddhist texts from the Tarim Basin dating to AD 500–1000 (ref. [13]), little is known about earlier Xinjiang populations and their possible genetic relationships with the Afanasievo or other groups.

[1]School of Life Sciences, Jilin University, Changchun, China. [2]Max Planck Institute for the Science of Human History, Jena, Germany. [3]Key Laboratory of Vertebrate Evolution and Human Origins, Institute of Vertebrate Paleontology and Paleoanthropology, Center for Excellence in Life and Paleoenvironment, Chinese Academy of Sciences, Beijing, China. [4]Xinjiang Institute of Cultural Relics and Archaeology, Ürümqi, China. [5]School of Archaeology, Jilin University, Changchun, China. [6]MOE Key Laboratory of Western China's Environmental Systems, College of Earth & Environmental Sciences, Lanzhou University, Lanzhou, China. [7]Key Laboratory of Cenozoic Geology and Environment, Institute of Geology and Geophysics, Chinese Academy of Sciences, Beijing, China. [8]College of Pharmacia Sciences, Jilin University, Changchun, China. [9]Institute of Archaeological Science, Fudan University, Shanghai, China. [10]Max Planck Institute for Evolutionary Anthropology, Leipzig, Germany. [11]Department of Anthropology, Harvard University, Cambridge, MA, USA. [12]School of Biological Sciences, Seoul National University, Seoul, Republic of Korea. [13]Key Laboratory for Evolution of Past Life and Environment in Northeast Asia, Ministry of Education, Jilin University, Changchun, China. [14]Research Center for Chinese Frontier Archaeology of Jilin University, Jilin University, Changchun, China. [15]These authors contributed equally: Fan Zhang, Chao Ning, Ashley Scott. ✉e-mail: ning@shh.mpg.de; krause@shh.mpg.de; warinner@shh.mpg.de; cwjeong@snu.ac.kr; cuiyq@jlu.edu.cn

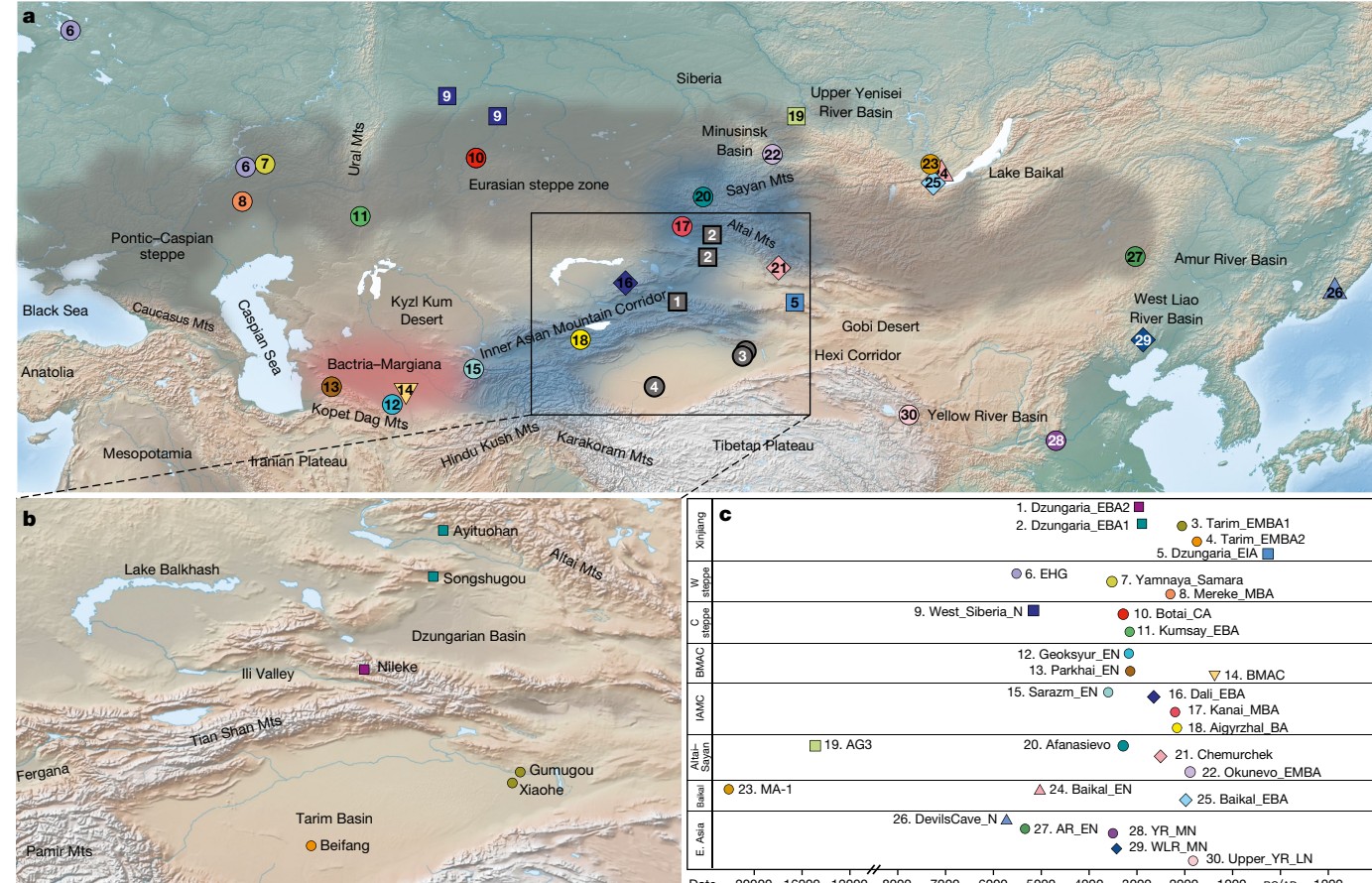

**Fig. 1 | Overview of the Xinjiang Bronze Age archaeological sites analysed in this study. a**, Overview of key Eurasian geographic regions, features and archaeological sites discussed in the text; new sites analysed in this study are shown in grey. **b**, Enhanced view of Xinjiang and the six new sites analysed in this study. **c**, Timeline of the sites in **a**. The timeline is organized by region, and the median date for each studied group is shown. The base maps in **a** and **b** were obtained from the Natural Earth public domain map dataset

(https://www.naturalearthdata.com/downloads/10m-raster-data/10m-cross-blend-hypso/). In the group labels, the suffixes represent the archaeological time periods of each group: N, Neolithic; EN, MN and LN, Early, Middle and Late Neolithic, respectively; EN, Eneolithic for Geoksyur, Parkhai and Sarazm; CA, Chalcolithic Age; BA, Bronze Age; MBA, Middle Bronze Age; EIA, Early Iron Age. MA-1, Mal'ta; EHG, Eastern European hunter-gatherers.

Since the late 1990s, the discovery of hundreds of naturally mummified human remains dating to around 2000 BC to AD 200 in the Tarim Basin has attracted international attention due to their so-called Western physical appearance, their felted and woven woollen clothing, and their agropastoral economy that included cattle, sheep/goats, wheat, barley, millet and even kefir cheese[16–19]. Such mummies have now been found throughout the Tarim Basin, among which the earliest are those found in the lowest layers of the cemeteries at Gumugou (2135–1939 BC), Xiaohe (1884–1736 BC) and Beifang (1785–1664 BC) (Fig. 1, Extended Data Fig. 1 and Extended Data Table 1). These and related Bronze Age sites are grouped within the Xiaohe archaeological horizon on the basis of their shared material culture[13,16,20].

Multiple contrasting hypotheses have been suggested by scholars to explain the origins and Western elements of the Xiaohe horizon, including the Yamnaya/Afanasievo steppe hypothesis[16], the Bactrian oasis hypothesis[21] and the Inner Asian Mountain Corridor (IAMC) island biogeography hypothesis[4]. The Yamnaya/Afanasievo steppe hypothesis posits that the Afanasievo-related EBA populations in the Altai–Sayan mountains spread via the Dzungarian Basin into the Tarim Basin and subsequently founded the agropastoralist communities making up the Xiaohe horizon around 2000 BC (refs. [16,22,23]). By contrast, the Bactrian oasis hypothesis posits that the Tarim Basin was initially colonized by migrating farmers of the Bactria–Margiana Archaeological Complex (BMAC) (around 2300–1800 BC) from the desert oases of Afghanistan,

Turkmenistan and Uzbekistan via the mountains of Central Asia. Support for this hypothesis is largely based on similarities in the agricultural and irrigation systems between the two regions that reflect adaptations to a desert environment, as well as evidence for the ritual use of *Ephedra* at both locations[3,21]. The IAMC island biogeography hypothesis similarly posits a mountain Central Asian origin for the Xiaohe founder population, but one linked to the transhumance of agropastoralists in the IAMC to the west and north of the Tarim Basin[4,24,25]. In contrast to these three migration models, the greater IAMC, which spans the Hindu Kush to Altai mountains, may have alternatively functioned as a geographic arena through which cultural ideas, rather than populations, primarily moved[25].

Recent archaeogenomic research has shown that Bronze Age Afanasievo of southern Siberia and IAMC/BMAC populations of Central Asia have distinguishable genetic profiles[15,26], and that these profiles are likewise also distinct from those of pre-agropastoralist hunter-gatherer populations in Inner Asia[2,5,7,27–30]. As such, an archaeogenomic investigation of Bronze Age Xinjiang populations presents a powerful approach for reconstructing the population histories of the Dzungarian and Tarim basins and the origins of the Bronze Age Xiaohe horizon. Examining the skeletal material of 33 Bronze Age individuals from sites in the Dzungarian (Nileke, Ayituohan and Songshugou) and Tarim (Xiaohe, Gumugou and Beifang) basins, we successfully retrieved ancient genome sequences from 5 EBA Dzungarian individuals (3000–2800 BC)

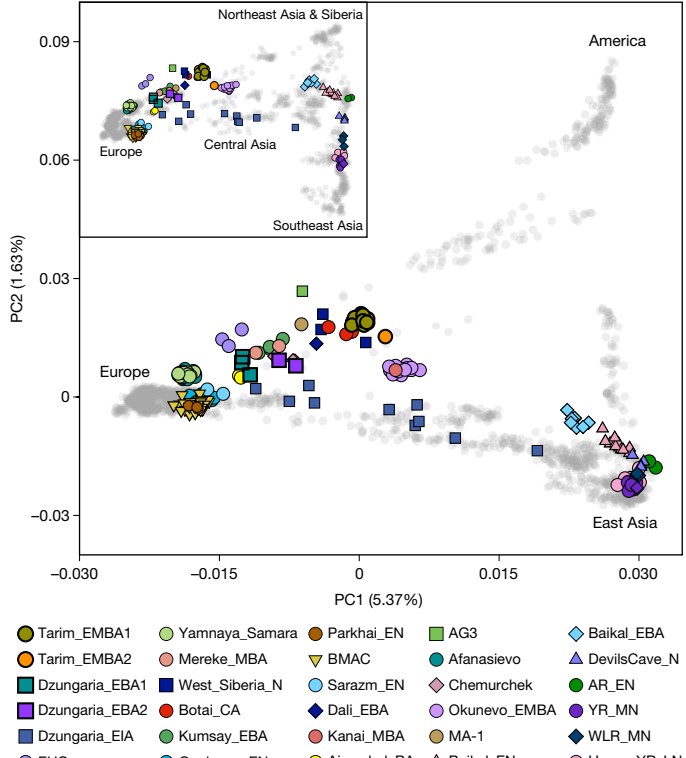

**Fig. 2 | Genetic structure of ancient and present-day populations included in this study.** Principal component analysis of ancient individuals projected onto Eurasian and Native American populations; the inset displays ancient individuals projected onto only Eurasian populations.

culturally assigned as Afanasievo, and genome-wide data from 13 Early–Middle Bronze Age (EMBA) Tarim individuals (2100–1700 BC) belonging to the Xiaohe horizon (Extended Data Table 1 and Supplementary Data 1A). We additionally report dental calculus proteomes of seven individuals from basal layers at the site of Xiaohe in the Tarim Basin (Extended Data Table 2). To the best of our knowledge, these individuals represent the earliest human remains excavated to date in the region.

## Genetic diversity of the Bronze Age Xinjiang

We obtained genome-wide data for 18 of 33 attempted individuals by either whole-genome sequencing or DNA enrichment for a panel of about 1.2 million single-nucleotide polymorphisms (1,240k panel SNPs) (Supplementary Data 1A). Overall, endogenous DNA was well preserved with minimal levels of contamination (Extended Data Table 1 and Supplementary Data 1A). To explore the genetic profiles of ancient Xinjiang populations, we first calculated the principal components of present-day Eurasian and Native American populations onto which we projected those of ancient individuals. Ancient Xinjiang individuals form several distinct clusters distributed along principal component 1 (PC1) (Fig. 2), the main principal component that separates eastern and western Eurasian populations. EBA Dzungarian individuals from the sites of Ayituohan and Songshugou near the Altai Mountains (Dzungaria_EBA1) fall close to EBA Afanasievo steppe herders from the Altai–Sayan mountains to the north. Genetic clustering with ADMIXTURE further supports this observation (Extended Data Fig. 3). The contemporaneous individuals from the Nileke site near the Tianshan mountains (Dzungaria_EBA2) are slightly shifted along PC1 towards the later Tarim individuals. In contrast to the EBA Dzungarian individuals, the EMBA individuals from the eastern Tarim sites of Xiaohe and Gumugou (Tarim_EMBA1) form a tight cluster close to pre-Bronze Age central steppe and Siberian individuals who share a high level of ancient

North Eurasian (ANE) ancestry (for example, Botai_CA). A contemporaneous individual from the Beifang site (Tarim_EMBA2) in the southern Tarim Basin is slightly displaced from the Tarim_EMBA1 towards EBA individuals from the Baikal region.

## Afanasievo genetic legacy in Dzungaria

Outgroup $f_3$ statistics supports a tight genetic link between the Dzungarian and Tarim groups (Extended Data Fig. 2A). Nevertheless, both of the Dzungarian groups are significantly different from the Tarim groups, showing excess affinity with various western Eurasian populations and sharing fewer alleles with ANE-related groups (Extended Data Fig. 2b, c). To understand this mixed genetic profile, we used qpAdm to explore admixture models of the Dzungarian groups with Tarim_EMBA1 or a terminal Pleistocene individual (AG3) from the Siberian site of Afontova Gora[31], as a source (Supplementary Data 1D). AG3 is a distal representative of the ANE ancestry and shows a high affinity with Tarim_EMBA1. Although the Tarim_EMBA1 individuals lived a millennium later than the Dzungarian groups, they are more genetically distant from the Afanasievo than the Dzungarian groups, suggesting that they have a higher proportion of local autochthonous ancestry. Here we define autochthonous to signify a genetic profile that has been present in a region for millennia, rather than being associated with more recently arrived groups.

We find that Dzungaria_EBA1 and Dzungaria_EBA2 are both best described by three-way admixture models (Fig. 3c, Extended Data Table 3 and Supplementary Data 1D) in which they derive a majority ancestry from Afanasievo (about 70% in Dzungaria_EBA1 and about 50% in Dzungaria_EBA2), with the remaining ancestry best modelled as a mixture of AG3/Tarim_EMBA1 (19–36%) and Baikal_EBA (9–21%). When we use Eneolithic and Bronze Age populations from the IAMC as a source, models fail when Afanasievo is not included as a source, and no contribution is allocated to the IAMC groups when Afanasievo is included (Supplementary Data 1D). Thus, Afanasievo ancestry, without IAMC contributions, is sufficient to explain the western Eurasian component of the Dzungarian individuals. We also find that the Chemurchek, an EBA pastoralist culture that succeeds the Afanasievo in both the Dzungarian Basin and Altai Mountains, derive approximately two-thirds of their ancestry from Dzungaria_EBA1 with the remainder from Tarim_EMBA1 and IAMC/BMAC-related sources (Fig. 3, Extended Data Table 3, Supplementary Data 1F and Supplementary Text 5). This helps to explain both the IAMC/BMAC-related ancestry previously noted in Chemurchek individuals[30] and their reported cultural and genetic affiliations to Afanasievo groups[32]. Taken together, these results indicate that the early dispersal of the Afanasievo herders into Dzungaria was accompanied by a substantial level of genetic mixing with local autochthonous populations, a pattern distinct from that of the initial formation of the Afanasievo culture in southern Siberia.

## Genetic isolation of the Tarim group

The Tarim_EMBA1 and Tarim_EMBA2 groups, although geographically separated by over 600 km of desert, form a homogeneous population that had undergone a substantial population bottleneck, as suggested by their high genetic affinity without close kinship, as well as by the limited diversity in their uniparental haplogroups (Figs. 1 and 2, Extended Data Fig. 4, Extended Data Table 1, Supplementary Data 1B and Supplementary Text 4). Using qpAdm, we modelled the Tarim Basin individuals as a mixture of two ancient autochthonous Asian genetic groups: the ANE, represented by an Upper Palaeolithic individual from the Afontova Gora site in the upper Yenisei River region of Siberia (AG3) (about 72%), and ancient Northeast Asians, represented by Baikal_EBA (about 28%) (Supplementary Data 1E and Fig. 3a). Tarim_EMBA2 from Beifang can also be modelled as a mixture of Tarim_EMBA1 (about 89%) and Baikal_EBA (about 11%). For both Tarim groups, admixture models

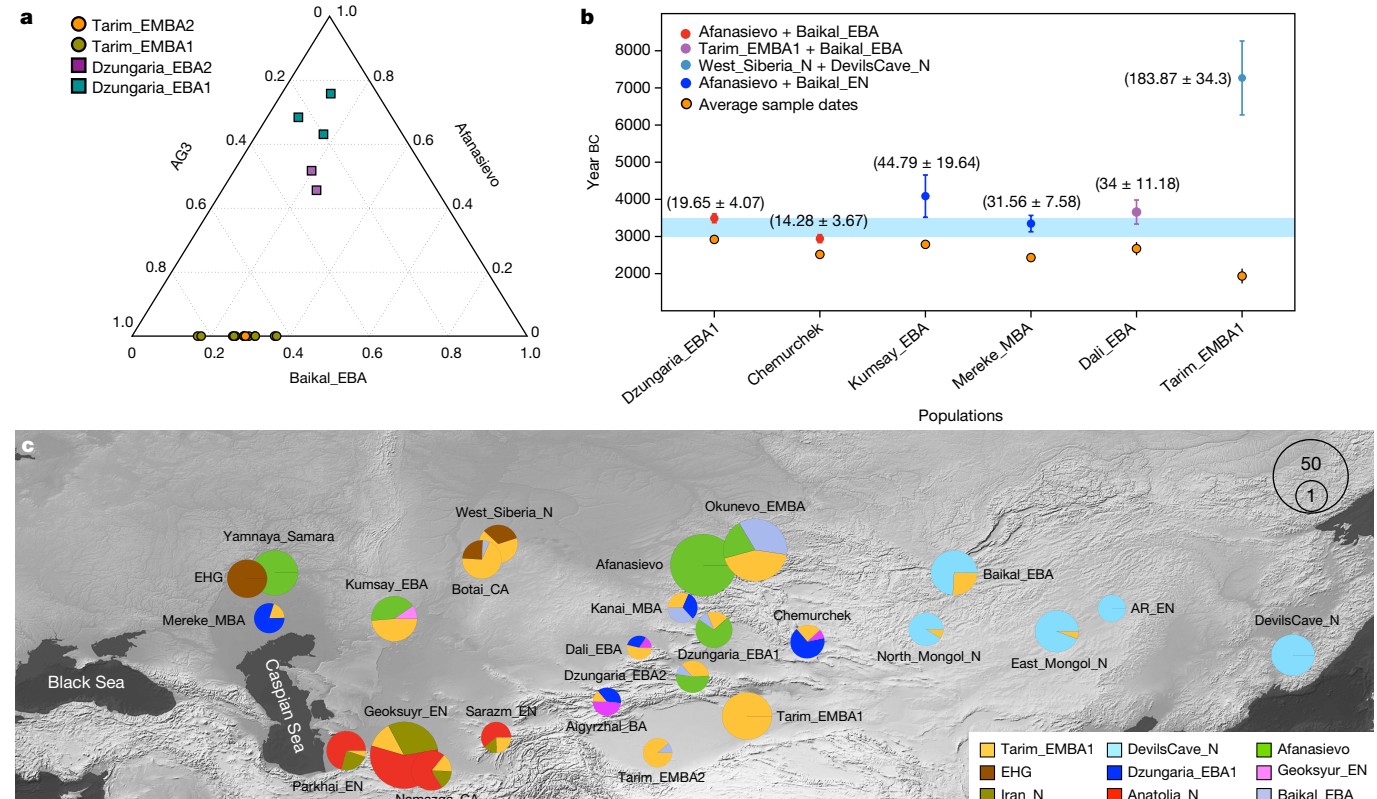

**Fig. 3 | Genetic ancestry and admixture dating of ancient populations from Xinjiang and its vicinity. a**, qpAdm-based estimates of the ancestry proportion of Dzungaria_EBA and Tarim_EMBA from three ancestry sources (AG3, Afanasievo and Baikal_EBA) (Supplementary Data 1D, E). Unlike Dzungaria_EBA individuals, Tarim_EMBA individuals are adequately modelled without EBA Eurasian steppe pastoralist (for example, Afanasievo) ancestry. **b**, Genetic admixture dates for key Bronze Age populations in Inner Asia, including Dzungaria_EBA1 (*n* = 3), Chemurchek (*n* = 3), Kumsay_EBA (*n* = 4), Mereke_MBA (*n* = 2), Dali_EBA (*n* = 1) and Tarim_EMBA1 (*n* = 12). The blue shade represents the radiocarbon dating range of the Yamnaya and Afanasievo individuals. The orange circles and the associated vertical bars represent the averages and standard deviations of median radiocarbon dates, respectively.

The circles above each orange circle represent the estimated admixture dates with a generation time of 29 years, and the vertical bars represent the sum of standard errors of the admixture date and the radiocarbon date estimate. **c**, Representative qpAdm-based admixture models of ancient Eurasian groups (Supplementary Data 1D–I). For Dzungaria_EBA1 and Geoksyur_EN, we show their three-way admixture models including Tarim_EMBA1 as a source. For later populations in Xinjiang, IAMC and nearby regions, we used them as sources, and allocated a colour to each of them (blue for Dzungaria_EBA1; magenta for Geoksyur_EN). The base map in **c** was obtained from the Natural Earth public domain map dataset (https://www.naturalearthdata.com/downloads/10m-raster-data/10m-gray-earth/).

unanimously fail when using the Afanasievo or IAMC/BMAC groups as a western Eurasian source (Supplementary Data 1E), thus rejecting a western Eurasian genetic contribution from nearby groups with herding and/or farming economies. We estimate a deep formation date for the Tarim_EMBA1 genetic profile, consistent with an absence of western Eurasian EBA admixture, placing the origin of this gene pool at 183 generations before the sampled Tarim Basin individuals, or 9,157 ± 986 years ago when assuming an average generation time of 29 years (Fig. 3b). Considering these findings together, the genetic profile of the Tarim Basin individuals indicates that the earliest individuals of the Xiaohe horizon belong to an ancient and isolated autochthonous Asian gene pool. This autochthonous ANE-related gene pool is likely to have formed the genetic substratum of the pre-pastoralist ANE-related populations of Central Asia and southern Siberia (Fig. 3c, Extended Data Fig. 2 and Supplementary Text 5).

## Pastoralism in the Tarim Basin

Although the harsh environment of the Tarim Basin may have served as a strong barrier to gene flow into the region, it was not a barrier to the flow of ideas or technologies, as foreign innovations, such as dairy pastoralism and wheat and millet agriculture, came to form the basis of the Bronze Age Tarim economies. Woollen fabrics, horns and bones of

cattle, sheep and goats, livestock manure, and milk and kefir-like dairy products have been recovered from the upper layers of the Xiaohe and Gumugou cemeteries[33–36], as have wheat and millet seeds and bundles of *Ephedra* twigs[34,37,38]. Famously, many of the mummies dating to 1650–1450 BC were even buried with lumps of cheese[35]. However, until now it has not been clear whether this pastoralist lifestyle also characterized the earliest layers at Xiaohe.

To better understand the dietary economy of the earliest archaeological periods, we analysed the dental calculus proteomes of seven individuals at the site of Xiaohe dating to around 2000–1700 BC. All seven individuals were strongly positive for ruminant-milk-specific proteins (Extended Data Table 2), including β-lactoglobulin, α-S1-casein and α-lactalbumin (Extended Data Fig. 5), and peptide recovery was sufficient to provide taxonomically diagnostic matches to cattle (*Bos*), sheep (*Ovis*) and goat (*Capra*) milk (Extended Data Fig. 5, Extended Data Table 2 and Supplementary Data 3). These results confirm that dairy products were consumed by individuals of autochthonous ancestry (Tarim_EMBA1) buried in the lowest levels of the Xiaohe cemetery (Extended Data Table 2). Importantly, however, and in contrast to previous hypotheses[36], none of the Tarim individuals was genetically lactase persistent (Supplementary Data 1J). Rather, the Tarim mummies contribute to a growing body of evidence that prehistoric dairy pastoralism in Inner and East Asia spread independently of lactase persistence genotypes[28,30].

## Discussion

Although human activities in Xinjiang can be traced back to around 40,000 years ago[24,39], the earliest evidence for sustained human habitation in the Tarim Basin dates only to the late third to early second millennium BC. There, at the sites of Xiaohe, Gumugou and Beifang, well-preserved mummified human remains buried within wooden coffins and associated with rich organic grave good assemblages represent the earliest known archaeological cultures of the region. Since their initial discovery in the early twentieth century and subsequent large-scale excavations beginning in the 1990s (ref. [16]), the Tarim mummies have been at the centre of debates with regard to their origins, their relationship to other Bronze Age steppe (Afanasievo), oasis (BMAC) and mountain (IAMC and Chemurchek) groups, and their potential connection to the spread of Indo-European languages into this region[3,4,40].

The palaeogenomic and proteomic data we present here suggest a very different and more complex population history than previously proposed. Although the IAMC may have been a vector for transmitting cultural and economic factors into the Tarim Basin, the known sites from the IAMC do not provide a direct source of ancestry for the Xiaohe populations. Instead, the Tarim mummies belong to an isolated gene pool whose Asian origins can be traced to the early Holocene epoch. This gene pool is likely to have once had a much wider geographic distribution, and it left a substantial genetic footprint in the EMBA populations of the Dzungarian Basin, IAMC and southern Siberia. The Tarim mummies' so-called Western physical features are probably due to their connection to the Pleistocene ANE gene pool, and their extreme genetic isolation differs from the EBA Dzungarian, IAMC and Chemurchek populations, who experienced substantial genetic interactions with the nearby populations mirroring their cultural links, pointing towards a role of extreme environments as a barrier to human migration.

In contrast to their marked genetic isolation, however, the populations of the Xiaohe horizon were culturally cosmopolitan, incorporating diverse economic elements and technologies with far-flung origins. They made cheese from ruminant milk using a kefir-like fermentation[37], perhaps learned from descendants of the Afanasievo, and they cultivated wheat, barley and millet[37,41], crops that were originally domesticated in the Near East and northern China and which were introduced into Xinjiang no earlier than 3500 BC (refs. [8,42]), probably via their IAMC neighbours[24]. They buried their dead with *Ephedra* twigs in a style reminiscent of the BMAC oasis cultures of Central Asia, and they also developed distinctive cultural elements not found among other cultures in Xinjiang or elsewhere, such as boat-shaped wooden coffins covered with cattle hides and marked by timber poles or oars, as well as an apparent preference for woven baskets over pottery[43,44]. Considering these findings together, it appears that the tightknit population that founded the Xiaohe horizon were well aware of different technologies and cultures outside the Tarim Basin and that they developed their unique culture in response to the extreme challenges of the Taklamakan Desert and its lush and fertile riverine oases[4].

This study illuminates in detail the origins of the Bronze Age human populations in the Dzungarian and Tarim basins of Xinjiang. Notably, our results support no hypothesis involving substantial human migration from steppe or mountain agropastoralists for the origin of the Bronze Age Tarim mummies, but rather we find that the Tarim mummies represent a culturally cosmopolitan but genetically isolated autochthonous population. This finding is consistent with earlier arguments that the IAMC served as a geographic corridor and vector for regional cultural interaction that connected disparate populations from the fourth to the second millennium BC (refs. [24,25]). While the arrival and admixture of Afanasievo populations in the Dzungarian Basin of northern Xinjiang around 3000 BC may have plausibly introduced Indo-European languages to the region, the material culture and genetic profile of the Tarim mummies from around 2100 BC onwards call into question simplistic assumptions about the link between genetics, culture and language and leave unanswered the question of whether the Bronze Age Tarim populations spoke a form of proto-Tocharian. Future archaeological and palaeogenomic research on subsequent Tarim Basin populations—and most importantly, studies of the sites and periods where first millennium AD Tocharian texts have been recovered—are necessary to understand the later population history of the Tarim Basin. Finally, the palaeogenomic characterization of the Tarim mummies has unexpectedly revealed one of the few known Holocene-era genetic descendant populations of the once widespread Pleistocene ANE ancestry profile. The Tarim mummy genomes thus provide a critical reference point for genetically modelling Holocene-era populations and reconstructing the population history of Asia.

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

## Methods

### Sample provenance

The archaeological human remains studied in this manuscript were excavated by the Xinjiang Institute of Cultural Relics and Archaeology from 1979 to 2017. Scientific investigation of these remains was approved by the Xinjiang Cultural Relics and Archaeology Institute, which holds the custodianship of the studied remains, based on the written agreements.

### Radiocarbon dating

Of the 18 individuals reported in this study, 10 were directly dated using accelerator mass spectrometry (AMS) at Beta Analytic, Miami, USA, and/or at Lanzhou University, China. To confirm the reliability of our AMS dating results, 4 out of the 10 individuals were AMS-dated at both Beta Analytic and Lanzhou University. Consistent dates were obtained in all cases (Supplementary Data 1C). The calibration of the dated samples was performed on the basis of the IntCal20 database[45] and using the OxCal v.4.4 program[46]. All of the samples were dated to time periods consistent with those estimated from archaeological stratigraphic layers and excavated grave goods.

### DNA laboratory procedures

Ancient DNA work was conducted in dedicated cleanroom laboratory facilities at the ancient DNA laboratories of Jilin University in Changchun and the Institute of Vertebrate Paleontology and Paleoanthropology in Beijing (Extended Data Table 1 and Supplementary Data 1A). For the 33 individuals initially screened in this study, approximately 50 mg of dentine or bone powder was obtained per individual from either teeth or bones. DNA was extracted following established protocols[47] with slight modifications (https://doi.org/10.17504/protocols.io.baksicwe). A subset of DNA extracts (n = 16) was subjected to a partial uracil-specific excision reagent repair following the methods described in ref. [48] (Extended Data Table 1 and Supplementary Data 1A). All 33 DNA extracts were built into double-stranded dual-index Illumina libraries. Libraries that were prepared in Jilin (n = 26) were directly shotgun sequenced on an Illumina HiSeq X10 or HiSeq 4000 instrument using 2 × 150-base-pair (bp) chemistry, and those with endogenous human DNA higher than 10% (n = 12) were sent for deeper sequencing. One of the 12 individuals (XHBM1) was later excluded from this study owing to high modern human DNA contamination (Supplementary Data 1A). For libraries prepared at the Institute of Vertebrate Paleontology and Paleoanthropology, samples with 0.1% or more human DNA from the initial screening (n = 7) were further enriched for approximately 1.2 million nuclear SNPs and then deeper sequenced on an Illumina HiSeq 4000 instrument using 2 × 150-bp chemistry. Together, a total of 18 individuals yielded sufficient high-quality ancient genomic data for downstream analyses (Extended Data Table 1).

### DNA sequence data processing

Raw read data were processed with EAGER v.1.92.55 (ref. [49]), a pipeline specially designed for processing ancient DNA sequence data. Specifically, raw reads were trimmed for Illumina adaptor sequences, and overlapping pairs were collapsed into single reads using AdapterRemoval 2.2.0 (ref. [50]). Merged reads were mapped to the human reference genome (hs37d5; GRCh37 with decoy sequences) using the aln/samse programs in BWA v.0.7.12 (ref. [51]). PCR duplicates were removed using DeDup v.0.12.2 (ref. [49]). To minimize the effect of postmortem DNA damage on genotyping, we trimmed BAM files generated from samples treated (n = 11) or not (n = 7) with uracil DNA glycosylase (UDG) by soft-masking up to 10 bp on both ends of each read using the trimbam function on bamUtils v.1.0.13 (ref. [52]) on the basis of the DNA misincorporation pattern per library tabulated using mapDamage v.2.0.9 (ref. [53]). For each SNP in the 1,240k panel, a single base from a high-quality read (base and mapping quality score 30 or higher) was randomly sampled to represent a pseudo-diploid genotype using the pileupCaller v.1.4.0.5 downloaded from https://github.com/stschiff/sequenceTools under the random haploid calling mode (-randomHaploid). For the transition SNPs (C/T and G/A), trimmed BAM files were used. For the transversion SNPs, BAM files without trimming were used.

### Ancient DNA authentication

We assessed the authenticity of our ancient DNA data as follows. First, we computed the proportion of C-to-T deamination errors at both the 5′ and 3′ ends of the sequencing reads, and found that all samples exhibited postmortem damage patterns characteristic of ancient DNA (Supplementary Data 1A). We then estimated mitochondrial DNA contamination for all individuals using the Schmutzi v.1.5.1 program[54]. To do this, we mapped adapter-trimmed reads to a 500-bp-extended revised Cambridge Reference Sequence (rCRS) of the human mitochondrial genome (NC_012920.1) to preserve reads passing through the origin, and then wrapped up the alignment to the regular rCRS with the circularmapper v.1.1 (ref. [49]). We successively ran the contDeam and schmutzi modules in the schmutzi program against the worldwide allele frequency database of 197 individuals to estimate the mitochondrial DNA contamination rate. Last, we estimated the nuclear contamination rate on men using ANGSD v.0.910 (ref. [55]), on the basis of the principle that mens have only a single copy of the X chromosome, and thus contamination will introduce extra mismatches among reads in SNP sites but not in the flanking monomorphic sites.

### DNA reference datasets

We compared the genome sequences of our ancient individuals to two sets of worldwide genotype panels, one based on the Affymetrix Axiom Genome-wide Human Origins 1 array (HumanOrigins; 593,124 autosomal SNPs)[56–58] and the other on the 1,240k dataset (1,233,013 autosomal SNPs including all of the HumanOrigins SNPs)[59]. We augmented both datasets by adding the Simons Genome Diversity Panel[60] and published ancient genomes (Supplementary Data 2A).

### Genetic relatedness analysis

We used pairwise mismatch rate (pmr)[61] and lcMLkin v0.5.0 (ref. [62]), to determine the genetic relatedness between ancient individuals. We calculated pmr for all pairs of ancient individuals in this study using the autosomal SNPs in the 1,240k panel and kept individual pairs with at least 8,000 SNPs covered by both to remove noisy estimates from low-coverage samples. We used lcMLkin to validate our observation in pmr analysis and to distinguish between parent–offspring and full sibling pairs.

### Uniparental haplogroup assignment

We aligned the adapter-trimmed reads to the rCRS NC_012920.1, and then generated the mitochondrial consensus sequence of each ancient individual using Geneious software v.11.1.3 (ref. [63]; https://www.geneious.com/). We assigned each consensus sequence into a specific haplogroup using HaploGrep2 (ref. [64]). For the Y chromosome, we used lineage-informative SNPs from the International Society of Genetic Genealogy 2016 tree (https://isogg.org/tree/2016/index16.html). For these SNPs, we called each individual's genotype using bcftools v.1.7 (ref. [51]) mpileup and call modules, after removing reads with mapping quality score < 30 (-q 30) and bases with quality score < 30 (-Q 30). We subsequently removed all heterozygous genotype calls. Then we assigned each individual to a specific Y haplogroup by manually comparing the genotype calls with the International Society of Genetic Genealogy SNPs. Before variant calling, we filtered alignment data using the pysam library v.0.15.2 (https://pysam.readthedocs.io/en/latest/) to reduce false positive variants due to postmortem damage and modern human contamination. We kept an observed base only if it was from a read shorter than 100 bp and the base was more than 10 bp away from the read ends. For transition SNPs, we further removed aligned bases if they were from a read with no postmortem damage pattern (that is, no C-to-T or G-to-A substitution). We determined each individual's Y

haplogroup primarily on the basis of the transversion SNPs and additionally considered transitions if transversions were insufficient.

## Population genetic analysis

We performed principal component analysis as implemented in smartpca v.16000 (ref. [65]) using a set of 2,077 present-day Eurasian individuals from the HumanOrigins dataset (Supplementary Data 2B) with the options 'lsqproject: YES' and 'shrinkmode: YES'. The unsupervised admixture analysis was performed with ADMIXTURE v.1.3.0 (ref. [66]). For ADMIXTURE, we removed genetic markers with minor allele frequency lower than 1% and pruned for linkage disequilibrium using the -indep-pairwise 200 25 0.2 option in PLINK v.1.90 (ref. [67]). We used outgroup $f_3$ statistics[68] to obtain a measurement of genetic relationship of the target population to a set of the Eurasian populations since their divergence from an African outgroup. We calculated $f_4$ statistics with the 'f4mode: YES' function in the ADMIXTOOLS package[58]. $f_3$ and $f_4$ statistics were calculated using qp3Pop v.435 and qpDstat v.755 in the ADMIXTOOLS package.

## Runs of homozygosity

We characterized whether the Bronze Age Xinjiang individuals descended from genetically related parents by estimating the runs of homozygosity (ROH). ROH refers to segments of the genome where the two chromosomes in an individual are identical to each other owing to recent common ancestry. Therefore, the presence of long ROH segments strongly suggests that an individual's parents are related. We applied the hapROH method[69] using the Python library hapROH v.0.3a4 with default parameters. The method was developed to identify ROH from low-coverage genotype data typical of ancient DNA and is still robust enough to identify ROH for individuals with a coverage down to 0.5× (ref. [69]). We reported the total sum of ROH longer than 4, 8, 12 and 20 cM, and visualized the results using DataGraph v.4.5.1.

## Genetic admixture modelling with qpAdm

We modelled our ancient Xinjiang populations using the qpWave/qpAdm programs (qpWave v.410 (ref. [70]) and qpAdm v.810 (ref. [57])). We used the following eight populations in the 1,240k dataset as the base set of outgroups (base) unless explicitly stated otherwise: Mbuti ($n = 5$), Natufian ($n = 6$), Onge ($n = 2$), Iran_N ($n = 5$), Villabruna ($n = 1$), Mixe ($n = 3$), Ami ($n = 2$), Anatolia_N ($n = 23$). This set includes an African outgroup (Mbuti), early Holocene Levantine hunter-gatherers (Natufian), Andamanese islanders (Onge), early Neolithic Iranians from the Tepe Ganj Dareh site (Iran_N), late Pleistocene Western European hunter-gatherers (Villabruna), Central Native Americans (Mixe), an indigenous group native to Taiwan (Ami) and Neolithic farmers from Anatolia (Anatolia_N). To compare competing models, we also took a 'rotating' approach, where we reciprocally added a source from a model to outgroups for a competing model. We specified which outgroups are used for all qpAdm models.

## Admixture dating with DATES

We used DATES v.753 (ref. [26]) for the dating of admixture events of the ancient populations with the pseudo-haploid genotype data under the simplified assumption that gene flow occurred as a single event, and assuming a generation time of 29 years (ref. [58]). The DATES software measures the decay of ancestry covariance to infer the admixture time and estimates jackknife standard errors. In the parameter file for running DATES, we used the options binsize: 0.001, maxdis: 0.5, runmode: 1, qbin: 10 and lovalfit: 0.45 in every run on the pseudo-haploid genotype data. For each target population, we chose a pair of reference populations that we identified as good sources in the qpAdm analysis. In cases in which the qpAdm source had limited sample size or SNP coverage, we chose an alternative that had a similar genetic profile to the qpAdm source but with better data quality to enhance the statistical power of the DATES analysis (Supplementary Data 1D–G). For Dzungaria_EBA1

and Chemurchek, we used the Afanasievo ($n = 20$) and Baikal_EBA ($n = 9$) as the references. For Kumsay_EBA and Mereke_MBA, we used the Afanasievo ($n = 20$) and Baikal_EN ($n = 15$). For Dali_EBA, we used Tarim_EMBA1 ($n = 12$) and Baikal_EBA ($n = 9$). For Tarim_EMBA1, we used West_Siberia_N ($n = 3$) and DevilsCave_N ($n = 4$).

## Protein extraction, digestion and liquid chromatography with tandem mass spectrometry

Total protein extractions were performed on dental calculus obtained from seven Xiaohe individuals excavated from layers 4 and 5 (Extended Data Table 2). Only individuals with calculus deposits >5 mg were analysed, and 5–10 mg of dental calculus was processed for each sample. Samples were extracted and digested using a filter-aided sample preparation, following decalcification in 0.5 M EDTA (ref. [71]). Extracted peptides were analysed by liquid chromatography with tandem mass spectrometry (MS/MS) using a Q-Exactive mass spectrometer (Thermo Scientific) coupled to an ACQUITY UPLC M-Class system (Waters AG) according to previously described protocols[28]. Potential contamination and sample carryover were monitored through the use of extraction blanks as well as injection blanks between each sample.

## Protein database searching

Tandem mass spectra were converted to Mascot generic files by MSConvert version 3.0.11781 using the 100 most intense MS/MS peaks. All MS/MS samples were analysed using Mascot (Matrix Science; v.2.6.0). Mascot was set up to search the SwissProt Release 2019_08 database (560,823 entries) assuming the digestion enzyme trypsin. Mascot was searched with a fragment ion mass tolerance of 0.050 Da and a parent ion tolerance of 10.0 ppm. Carbamidomethylation of cysteine was specified in Mascot as a fixed modification. Deamidation of asparagine and glutamine and oxidation of methionine and proline were specified in Mascot as variable modifications. A subset of samples were analysed in duplicate (Supplementary Data 3), and the results were combined using multidimensional protein identification technology (MudPIT) before analysis.

## Criteria for protein identification

MS/MS-based protein and peptide identifications were validated using Scaffold (version Scaffold_4.9.0, Proteome Software). Peptide identifications were accepted if they could be established at greater than 86.0% probability to achieve a false discovery rate (FDR) less than 1.0% by the Peptide Prophet algorithm[71] with Scaffold delta-mass correction. Protein identifications were accepted if they could be established at an FDR of less than 5.0% and contained at least two unique peptides. Final protein and peptide FDRs were 1.8% and 0.99%, respectively. Protein probabilities were assigned by the Protein Prophet algorithm[72]. After establishing the presence of the milk proteins β-lactoglobulin and α-S1-casein using these criteria, we expanded our analysis to accept further milk proteins identified on the basis of single peptides for high-scoring PSMs (>60), which resulted in the additional identification of α-lactalbumin. Proteins that contained similar peptides that could not be differentiated on the basis of MS/MS analysis alone were grouped to satisfy the principles of parsimony. All samples yielded proteomes typical of a dental calculus oral microbiome, and damage-associated modifications (N and Q deamidation) characteristic of ancient proteins were observed (Supplementary Data 3).

## Reporting summary

Further information on research design is available in the Nature Research Reporting Summary linked to this paper.

## Data availability

The DNA sequences reported in this paper have been deposited in the European Nucleotide Archive under the accession number PRJEB46875.

Haploid genotype data of ancient individuals in this study on the 1,240k panel are available in the EIGENSTRAT format at https://edmond.mpdl.mpg.de/imeji/collection/OMm2fpu0jR3jSqnY. The protein spectra have been deposited in the ProteomeXchange Consortium via the PRIDE partner repository under the accession number PDX027706. The publicly available database SwissProt release 2019_08 is accessible through the UniProt Knowledge Base (https://www.uniprot.org). The basemaps used in Figs. 1, 3 are in the public domain and accessible through the Natural Earth website (https://www.naturalearthdata.com/downloads/10m-raster-data/).

## Code availability

All of the analyses performed in this study are based on publicly available software programs. Specific version information and non-default arguments are described in the Methods.

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

**Acknowledgements** We thank Xinjiang Institute of Cultural Relics and Archaeology and Renmin University of China for providing such valuable samples for study; Lanzhou University for providing the AMS dating results; K. Wang, H. Yu and G. A. Gnecchi-Ruscone for helpful comments on the genetic landscapes of the Eurasian steppe; T. Hermes and R. Flad for helpful comments on the broader archaeological context of the region. This work was supported by the National Key R&D Program of China (grant numbers 2016YFE0203700 and 2018YFA0606402), the National Natural Science Foundation of China (grant number 42072018, 41925009), the Fundamental Research Funds for the Central Universities, the European Research Council under the European Union's Horizon 2020 research and innovation programme (grant agreement numbers 804884-DAIRYCULTURES and 646612-Eurasia3angle), the Humanities and Social Sciences Key Research Base of the Ministry of Education (16JJD780005), the National Research Foundation of Korea grant funded by the Korean Government (MSIT; 2020R1C1C1003879) and the Max Planck Society.

**Author contributions** Y.C., C.J., C.W., C.N. and J.K. conceived and supervised the study. F.Z., A.S, L.F., P.C., R.Y., F.L. and Q.D. performed the laboratory work. Q.F., D.W., W.L., X.H., Q.R., I.A., C.L., S.G., Y.X., S. Wu, S. Wen, H. Zhu, H. Zhou and A.N. provided archaeological materials and associated information. R.B. and M.R. provided the linguistic background and G.D. and Z.T. assisted with the AMS dating. C.N., F.Z., A.S., C.W., C.J., Q.F., P.M., X.F., W.W. and V.K. analysed data. C.N., C.W., C.J., Y.C., F.Z. and A.S. wrote the manuscript with input from all coauthors.

**Funding** Open access funding provided by Max Planck Society.

**Competing interests** The authors declare no competing interests.

**Additional information**
**Correspondence and requests for materials** should be addressed to Chao Ning, Johannes Krause, Christina Warinner, Choongwon Jeong or Yinqiu Cui.

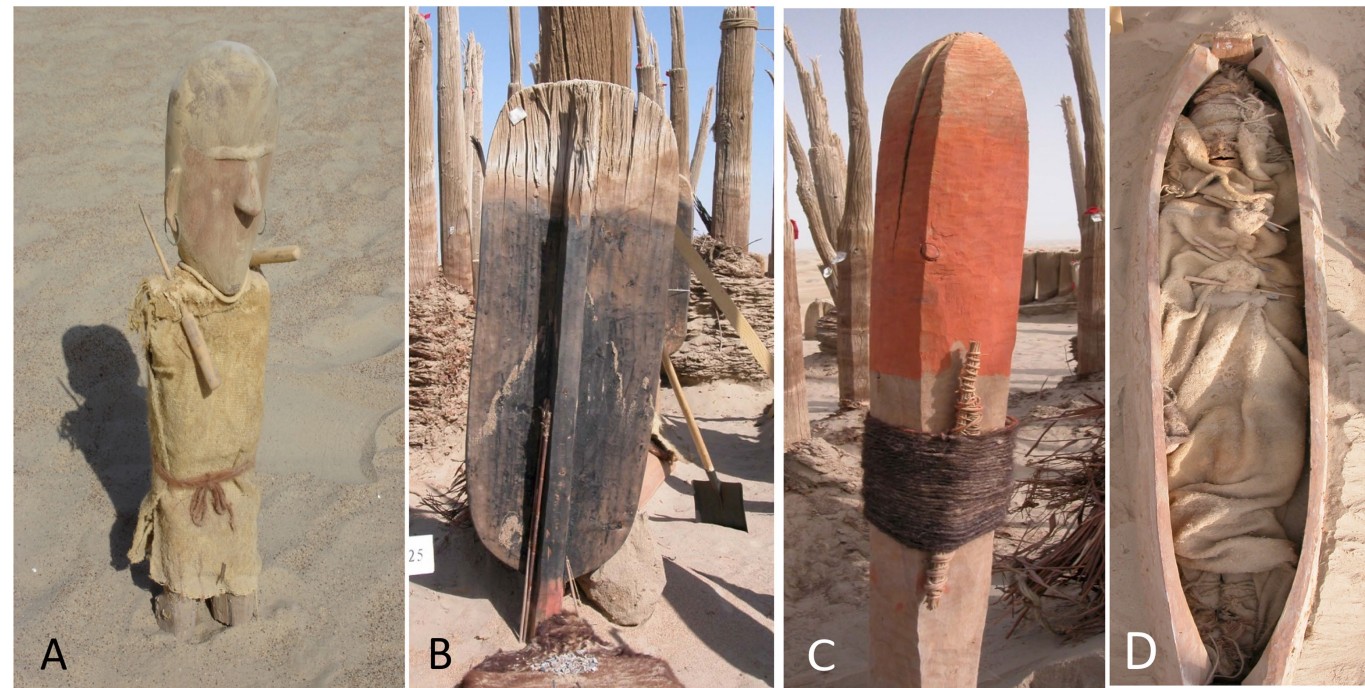

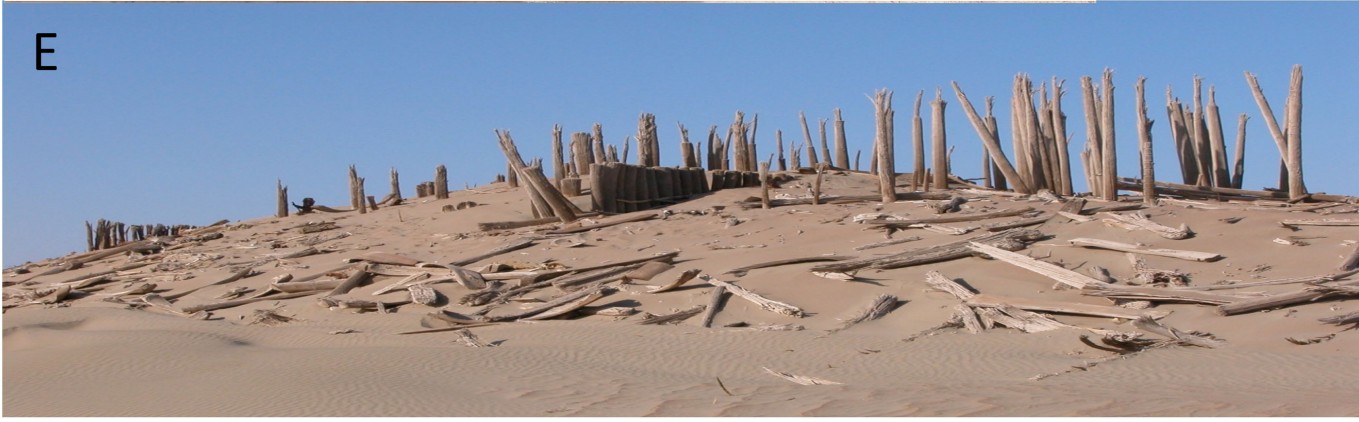

**Extended Data Fig. 1 | Burial goods excavated from the Xiaohe cemetery.**
**A**, a wooden sculpture excavated from the upper layer of a double-layer mud coffin of XHM75. **B**, an oar-plank placed in front of a male burial. **C**, a wooden pole placed in front of a female burial. **D**, Burial XHM66 from layer 4 of the Xiaohe cemetery illustrating typical features of early burials, including boat-shaped coffins and mummified remains dressed in woollen garments. This burial style is common at Bronze Age cemeteries throughout the Tarim Basin, including Beifang and Gumugou. **E**, Side view of the Xiaohe cemetery showing wooden grave markers and fencing.

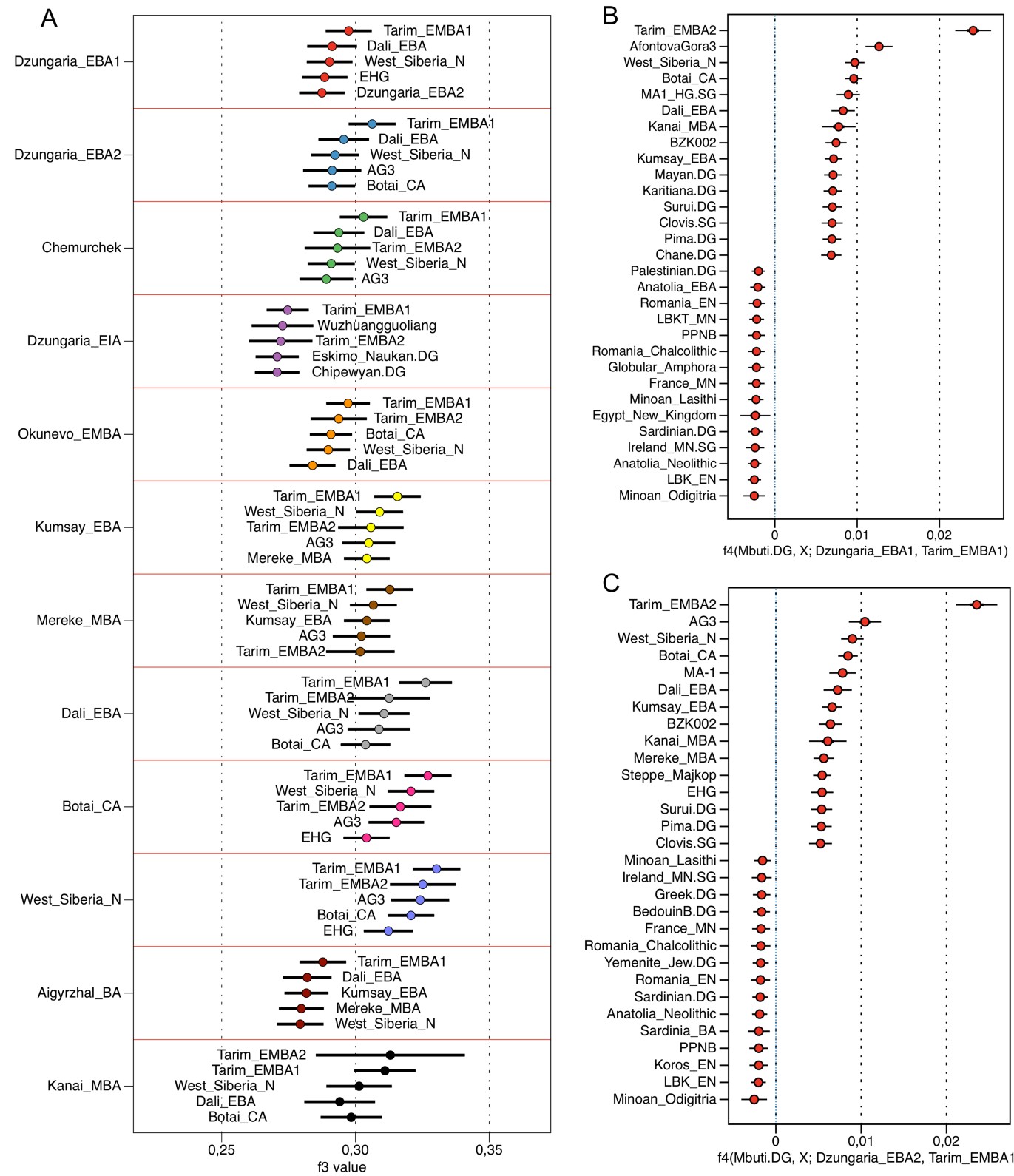

**Extended Data Fig. 2 | F-statistics for the ancient Xinjiang and the Eurasian steppe populations. A**, we show top 5 outgroup *f*3-statistics of the form *f*3(Target, X; Mbuti) for the 361 world-wide populations as contrast populations X, and 8 populations from this study and the Eurasian Steppe as target: Dzungaria_EBA1, Dzungaria_EBA2, Chemurchek, Dzungaria_EIA, Okunevo_EMBA, Kazakhstan_EMBA, Botai_CA, West_Siberia_N, horizontal bars represent ± 1 standard error measure (s.e.m.) calculated by 5 cM block jackknifing. **B**, *f*4-statistics of the form *f*4(Mbuti, X; Dzungaria_EBA1, Tarim_

EMBA1), horizontal bars represent ± 3 (thin) and ± 1 (thick) s.e.m. calculated by 5 cM block jackknifing, and **C**, *f*4-statistics of the form *f*4(Mbuti, X; Dzungaria_EBA2, Tarim_EMBA1), where X is 361 world-wide populations. We show the top and the bottom 15 *f*4 statistics. Horizonal bars represent the point estimate ± 3 (thin) and ± 1 (thick) s.e.m., respectively, as estimated using 5 cM block jackknifing. *F*4 statistics deviating three s.e.m. or more from zero are marked in red.

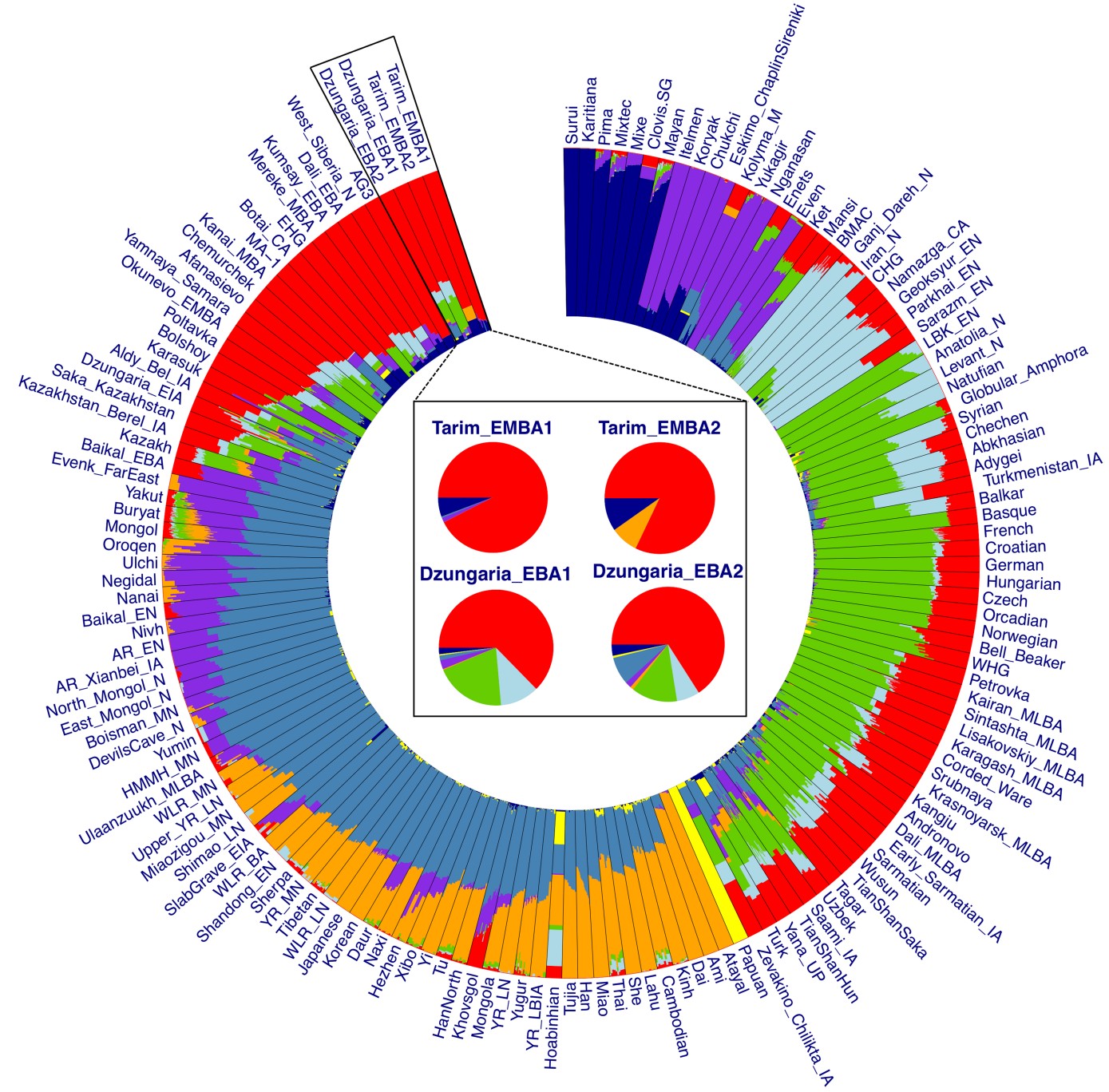

**Extended Data Fig. 3 | Unsupervised ADMIXTURE plot for the Bronze Age Xinjiang individuals.** We plot ancestry component estimates for K = 8 using 'AncestryPainter' (https://www.picb.ac.cn/PGG/resource.php). Dzungaria_ EBA individuals show an ancestry pattern close to Afanasievo and Yamnaya, while Tarim_EMBA individuals show a pattern similar to AG3, West_Siberia_N and Botai_CA from the Eurasia steppe.

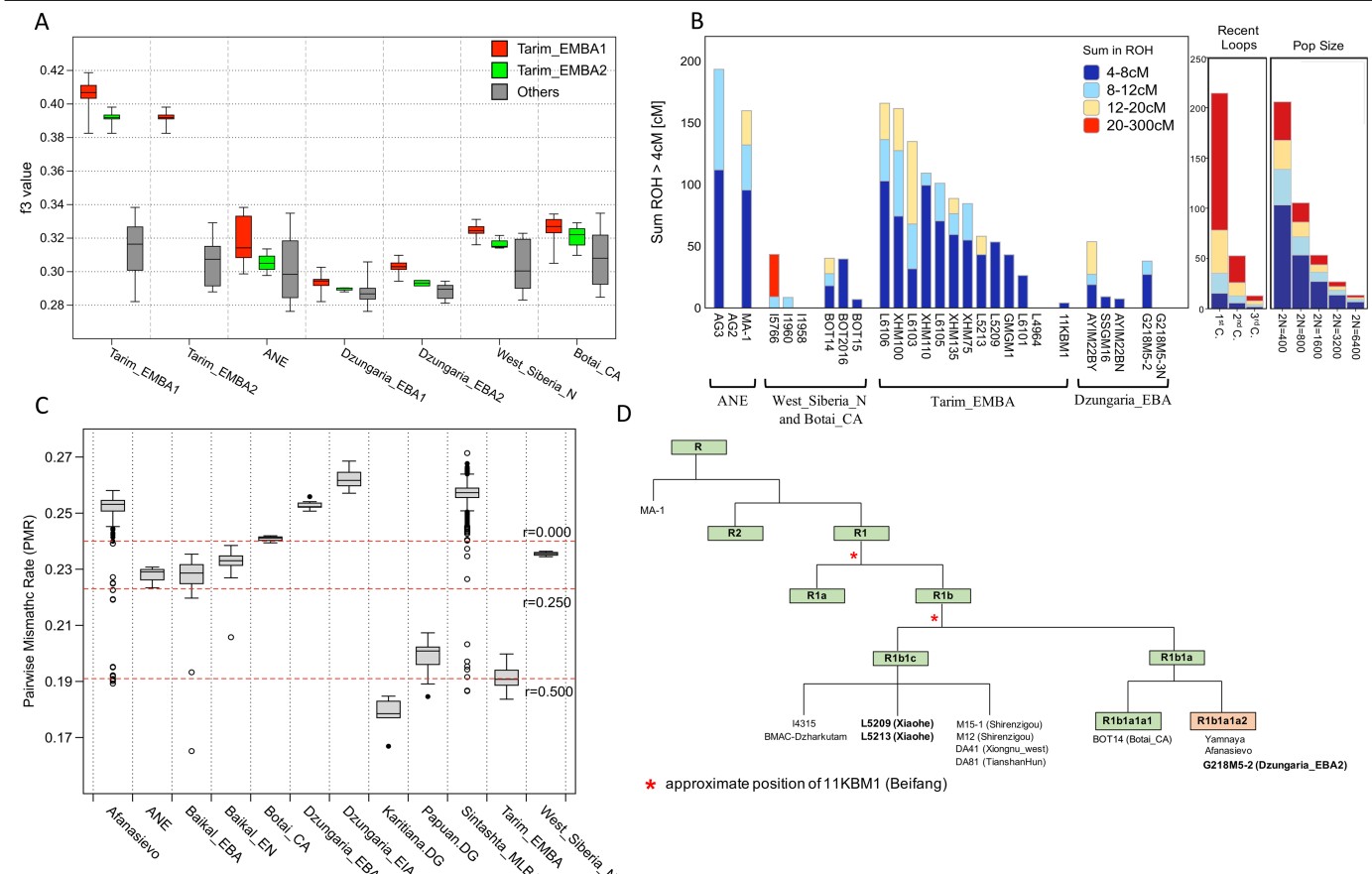

**Extended Data Fig. 4 | Reduced genetic diversity of the Tarim_EMBA individuals. A**, a comparison of individual outgroup *f*3-statistics for the ancient Xinjiang populations and their neighboring populations from Inner Asia, including Tarim_EMBA1 (n = 12), Tarim_EMBA2 (n = 1), ANE (n = 3), Dzungaria_EBA1 (n = 3), Dzungaria_EBA2 (n = 2), West_Siberia_N (n = 3) and Botai_CA (n = 3), which Tarim Basin individuals show the highest affinity to each other. In each boxplot, the box marks the 25th and 75th quartiles of the distribution, respectively, and the horizontal line within the box marks the median. The whisker delineates the maximum and the minimum. **B**, the cumulative distribution of ROH tracts shows that Tarim_EMBA individuals did not descend from close related parents. **C**, pairwise mismatch rate (pmr) between individuals in the ancient populations of Xinjiang and its neighboring regions, including all pairs of individuals within the Afanasievo (n = 27), ANE (n = 3), Baikal_EBA (n = 9), Baikal_EN (n = 15), Botai_CA (n = 3), Dzungaria_EBA (n = 5), Dzungaria_EIA (n = 10), Sintashta_MLBA (n = 51), Tarim_EMBA (n = 13), West_Siberia_N (n = 3), as well as present-day isolated populations such as

Papuan and Karitiana. Tarim_EMBA individuals uniformly show a much reduced pmr value that is equivalent to the first-degree relatives in Afanasievo or Sintashta_MLBA. The red dotted lines mark the expected pmr value for the given coefficient of relationship (*r*), ranging from 0 (unrelated) and 1/4 (second degree relatives) to 1/2 (first degree relatives), based on the mean value of pmr among these populations, respectively. In each box plot, the box represents the interquartile range (the 25th and 75th quartiles), and the horizon line within the box represents the median. Black-filled and open circles represent outliers (1.5 times beyond the IQR) and extreme outliers (3 times beyond the IQR), respectively. The whisker delineates the smallest and the largest non-outlier observations. **D**, Y chromosome phylogeny of the Bronze Age Xinjiang male individuals. Xiaohe male individuals fall into a branch distinct from western Bronze Age steppe pastoralists, such as Afanasievo and Yamnaya. One individual from Beifang falls in a position that is more basal than Xiaohe, but its phylogenetic position cannot be fixed due to low coverage, and its proximate position(s) are instead indicated with an asterisk.

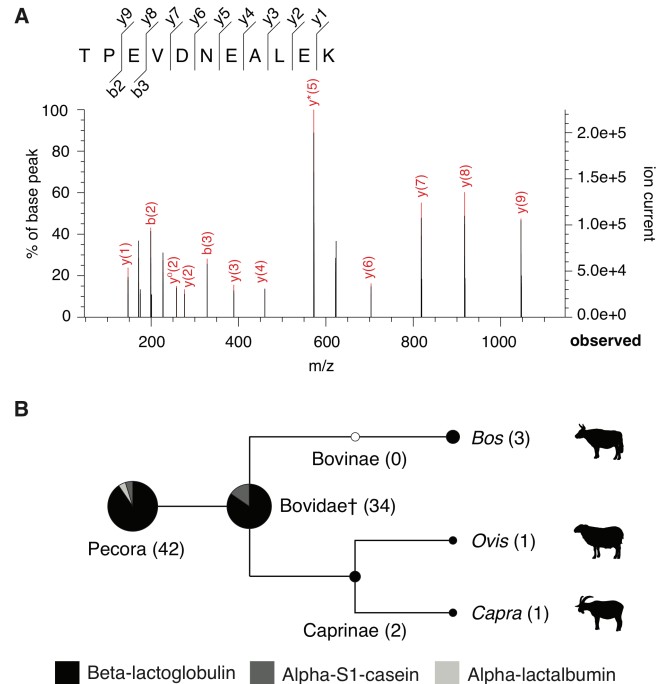

**A**

**B**

Beta-lactoglobulin   Alpha-S1-casein   Alpha-lactalbumin

**Extended Data Fig. 5 | Proteomic evidence for dairy consumption in Xiaohe dental calculus, ca. 2000-1800 BCE. A**, B- and Y-ion series for the frequently observed β-lactoglobulin peptide TPEVD(D/N/K)EALEK, which contains a taxon-specific polymorphic residue: D, Bovinae; N, *Ovis*; K, *Capra*. See SI Appendix. **B**, Taxonomically assigned β-lactoglobulin (black), α-S1-casein (dark grey), and α-lactalbumin peptide spectral matches (PSMs) presented as scaled pie charts on a cladogram of dairy livestock. Bracketed numbers represent the number of PSMs (excluding duplicates) assigned to each node. †Included on the Bovidae node are: 13 PSMs assigned to Bovidae; 21 PSMs assigned to Bovidae but excluding *Capra*.

**Extended Data Table 1 | A summary of the Bronze Age Xinjiang individuals reported in this study**

| Sample ID | Group ID | Date (BCE) | Archaeol-ogical site | Skeletal elements | UDG treatment | SNPs | Mean coverage | Biological sex | mtDNA haplogroup | Y. chr. haplogroup |
|---|---|---|---|---|---|---|---|---|---|---|
| AYIM22BY | Dzungaria_EBA1 | **2843-2811** | Ayituohan | Tooth | no | 495853 | 0.7010 | M | U5a1a1 | Q1b1 |
| AYIM22BN | Dzungaria_EBA1 | 2800-2600 | Ayituohan | Petrous bone | no | 524334 | 0.8990 | F | T2d1a | — |
| SSGM16 | Dzungaria_EBA1 | **2863-2801** | Songshugou | Tooth | no | 785354 | 1.4379 | F | H2b | — |
| G218M5-2 | Dzungaria_EBA2 | **2907-2851** | Nileke | Tooth | no | 790831 | 1.3523 | M | H15b1 | R1b1a1a2a2 |
| G218M5-3N | Dzungaria_EBA2 | **3005-2987** | Nileke | Tooth | no | 415959 | 0.5992 | M | U5a'b | Q1b1 |
| GMGM1 | Tarim_EMBA1 | **2135-2074** | Gumugou | Radius | no | 269355 | 0.3696 | F | C4 | — |
| XHM100 | Tarim_EMBA1 | **1884-1740** | Xiaohe | Tooth | half | 356623 | 0.4014 | F | C4 | — |
| XHM110 | Tarim_EMBA1 | 2000-1800 | Xiaohe | Tooth | half | 514292 | 0.6378 | F | C4 | — |
| XHM135 | Tarim_EMBA1 | **1936-1860** | Xiaohe | Tooth | half | 781191 | 0.6301 | F | C4 | — |
| XHM75 | Tarim_EMBA1 | 2000-1800 | Xiaohe | Tooth | half | 227235 | 0.2471 | F | C4 | — |
| L5209 | Tarim_EMBA1 | 2000-1800 | Xiaohe | Tooth | half | 767789 | 1.2160 | M | C4 | R1b1c |
| L5213 | Tarim_EMBA1 | 2000-1800 | Xiaohe | Tooth | half | 808182 | 1.2051 | M | R1b1 | R1b1c |
| L4964 | Tarim_EMBA1 | 2000-1800 | Xiaohe | Tooth | half | 60893 | 0.0632 | F | C4 | — |
| L6101 | Tarim_EMBA1 | **1767-1623** | Xiaohe | Tooth | half | 160220 | 0.1813 | F | C4 | — |
| L6103 | Tarim_EMBA1 | **1785-1664** | Xiaohe | Tooth | half | 536220 | 1.0811 | F | C4 | — |
| L6105 | Tarim_EMBA1 | 2000-1800 | Xiaohe | Tooth | half | 490934 | 0.5782 | F | C4 | — |
| L6106 | Tarim_EMBA1 | 2000-1800 | Xiaohe | Tooth | half | 247447 | 0.3530 | F | C4 | — |
| 11KBM1 | Tarim_EMBA2 | **1876-1839** | Beifang | Tooth | half | 131030 | 0.1943 | M | C4 | R1 (xR1a, xR1b1) |

In the "Date (BCE)" column, individuals directly dated by AMS are marked in bold (calibrated dates with 95.4% confidence interval) while the remaining dates are based on the archaeological contexts. The "SNPs" column shows the number of SNPs in the 1240k panel covered in each individual. Genome-wide data of seven individuals (L5209, L5213, L4964, L6101, L6103, L6105, L6106) were generated from IVPP by enriching endogenous DNA for the 1240k panel SNPs.

**Extended Data Table 2 | Dietary proteins identified in the dental calculus of individuals analyzed from the Tarim Basin Xiaohe cemetery**

| Sample ID | Date (BCE) | Archaeological layer | Milk Proteins | Total PSMs | Livestock |
|---|---|---|---|---|---|
| XHM100 | **1884-1740** | 5 | β-lactoglobulin | 50 | Cattle or sheep |
| XHM109 | 2000-1800 | 4 | β-lactoglobulin | 29 | Cattle or sheep |
| XHM112 | 2000-1800 | 5 | β-lactoglobulin | 23 | Cattle |
| XHM115 | 2000-1800 | 5 | β-lactoglobulin, α-S1-casein, α-lactalbumin | 98 | Cattle, goat, sheep |
| XHM117 | 2000-1800 | 5 | β-lactoglobulin, α-S1-casein, α-lactalbumin | 132 | Cattle or sheep |
| XHM125 | 2000-1800 | 4 | β-lactoglobulin | 41 | Cattle |
| XHM135 | **1936-1860** | 5 | β-lactoglobulin | 10 | Cattle or sheep |

In the "Date (BCE)" column, individuals directly dated by AMS are marked in bold (calibrated dates with 95.4% confidence interval) while the remaining dates are based on the archaeological contexts. The "Livestock" column shows consensus taxonomic assignment based on observed amino acid variants in milk peptides.

## Extended Data Table 3 | Robustness of key qpAdm admixture models

**A. No BMAC/IAMC-related ancestry component in Dzungaria_EBA**

| Target | Ref1 | Ref2 | Ref3 | Pval | Coef(Ref1) | Coef(Ref2) | Coef(Ref3) | Extra outgroups |
|---|---|---|---|---|---|---|---|---|
| Dzungaria_EBA1 | Afanasievo | Tarim_EMBA1 | Baikal_EBA | 2.90E-01 | 0.717 ± 0.024 | 0.192 ± 0.044 | 0.091 ± 0.026 | |
| | Afanasievo | Tarim_EMBA1 | Baikal_EBA | 3.06E-01 | 0.710 ± 0.022 | 0.212 ± 0.036 | 0.078 ± 0.020 | Geoksyur_EN |
| | Afanasievo | Tarim_EMBA1 | Baikal_EBA | 3.61E-01 | 0.712 ± 0.022 | 0.205 ± 0.036 | 0.083 ± 0.020 | BMAC |
| | Afanasievo | Tarim_EMBA1 | Baikal_EBA | 1.41E-01 | 0.754 ± 0.022 | 0.169 ± 0.026 | 0.078 ± 0.018 | Geoksyur_EN+AG3 |
| Dzungaria_EBA2 | Afanasievo | Tarim_EMBA1 | Baikal_EBA | 1.48E-02 | 0.532 ± 0.030 | 0.359 ± 0.057 | 0.109 ± 0.034 | |
| | Afanasievo | Tarim_EMBA1 | Baikal_EBA | 1.60E-02 | 0.545 ± 0.027 | 0.325 ± 0.044 | 0.130 ± 0.025 | Geoksyur_EN |
| | Afanasievo | Tarim_EMBA1 | Baikal_EBA | 2.16E-02 | 0.540 ± 0.027 | 0.336 ± 0.044 | 0.125 ± 0.025 | BMAC |
| | Afanasievo | Tarim_EMBA1 | Baikal_EBA | 5.08E-01 | 0.527 ± 0.026 | 0.349 ± 0.033 | 0.125 ± 0.023 | Geoksyur_EN+AG3 |

**B. Comparison of Dzungaria_EBA1 and Afanasievo as a source for Chemurchek and IAMC populations**

| Target | Ref1 | Ref2 | Ref3 | Pval | Coef(Ref1) | Coef(Ref2) | Coef(Ref3) | Extra outgroups |
|---|---|---|---|---|---|---|---|---|
| Chemurchek_merged | Tarim_EMBA1 | Dzungaria_EBA1 | Geoksuyr_EN | 4.92E-01 | 0.241 ± 0.044 | 0.673 ± 0.081 | 0.087 ± 0.046 | |
| | Tarim_EMBA1 | Dzungaria_EBA1 | Geoksuyr_EN | 4.74E-01 | 0.260 ± 0.038 | 0.618 ± 0.057 | 0.122 ± 0.029 | Afanasievo |
| | Tarim_EMBA1 | Afanasievo | Geoksuyr_EN | 4.29E-03 | 0.466 ± 0.021 | 0.440 ± 0.045 | 0.094 ± 0.039 | |
| | Tarim_EMBA1 | Afanasievo | Geoksuyr_EN | 5.24E-04 | 0.454 ± 0.021 | 0.403 ± 0.044 | 0.143 ± 0.036 | Dzungaria_EBA1 |
| Dali_EBA | Tarim_EMBA1 | Dzungaria_EBA1 | Geoksuyr_EN | 6.03E-01 | 0.537 ± 0.053 | 0.303 ± 0.096 | 0.159 ± 0.055 | |
| | Tarim_EMBA1 | Dzungaria_EBA1 | Geoksuyr_EN | 7.05E-01 | 0.529 ± 0.049 | 0.326 ± 0.073 | 0.145 ± 0.036 | Afanasievo |
| | Tarim_EMBA1 | Afanasievo | Geoksuyr_EN | 7.58E-01 | 0.634 ± 0.029 | 0.248 ± 0.065 | 0.118 ± 0.055 | |
| | Tarim_EMBA1 | Afanasievo | Geoksuyr_EN | 8.18E-01 | 0.633 ± 0.029 | 0.245 ± 0.060 | 0.122 ± 0.047 | Dzungaria_EBA1 |
| Aigyrzhal_BA | Tarim_EMBA1 | Dzungaria_EBA1 | Geoksuyr_EN | 1.69E-01 | 0.140 ± 0.042 | 0.377 ± 0.079 | 0.483 ± 0.045 | |
| | Tarim_EMBA1 | Dzungaria_EBA1 | Geoksuyr_EN | 1.93E-01 | 0.160 ± 0.035 | 0.326 ± 0.054 | 0.514 ± 0.029 | Afanasievo |
| | Tarim_EMBA1 | Afanasievo | Geoksuyr_EN | 1.29E-02 | 0.278 ± 0.020 | 0.211 ± 0.050 | 0.512 ± 0.043 | |
| | Tarim_EMBA1 | Afanasievo | Geoksuyr_EN | 1.60E-02 | 0.273 ± 0.020 | 0.203 ± 0.046 | 0.525 ± 0.038 | Dzungaria_EBA1 |
| Kanai_MBA | Tarim_EMBA1 | Dzungaria_EBA1 | Baikal_EBA | 8.74E-01 | 0.324 ± 0.118 | 0.380 ± 0.076 | 0.296 ± 0.058 | |
| | Tarim_EMBA1 | Dzungaria_EBA1 | Baikal_EBA | 6.89E-01 | 0.211 ± 0.092 | 0.427 ± 0.071 | 0.363 ± 0.037 | Afanasievo |
| | Tarim_EMBA1 | Afanasievo | Baikal_EBA | 7.65E-01 | 0.316 ± 0.102 | 0.316 ± 0.054 | 0.368 ± 0.060 | |
| | Tarim_EMBA1 | Afanasievo | Baikal_EBA | 8.93E-01 | 0.371 ± 0.075 | 0.293 ± 0.047 | 0.336 ± 0.043 | Dzungaria_EBA1 |
| Kumsay_EBA | Tarim_EMBA1 | Dzungaria_EBA1 | Geoksuyr_EN | 5.95E-04 | 0.317 ± 0.043 | 0.561 ± 0.079 | 0.122 ± 0.043 | |
| | Tarim_EMBA1 | Dzungaria_EBA1 | Geoksuyr_EN | 1.18E-03 | 0.304 ± 0.036 | 0.592 ± 0.054 | 0.104 ± 0.027 | Afanasievo |
| | Tarim_EMBA1 | Afanasievo | Geoksuyr_EN | 7.97E-02 | 0.488 ± 0.018 | 0.416 ± 0.041 | 0.097 ± 0.034 | |
| | Tarim_EMBA1 | Afanasievo | Geoksuyr_EN | 9.14E-02 | 0.488 ± 0.018 | 0.388 ± 0.040 | 0.124 ± 0.031 | Dzungaria_EBA1 |
| Mereke_MBA | Tarim_EMBA1 | Dzungaria_EBA1 | Geoksuyr_EN | 1.60E-01 | 0.195 ± 0.048 | 0.810 ± 0.087 | -0.005 ± 0.048 | |
| | Tarim_EMBA1 | Dzungaria_EBA1 | Geoksuyr_EN | 9.55E-02 | 0.229 ± 0.039 | 0.718 ± 0.058 | 0.052 ± 0.028 | Afanasievo |
| | Tarim_EMBA1 | Afanasievo | Geoksuyr_EN | 1.84E-01 | 0.455 ± 0.021 | 0.566 ± 0.047 | -0.021 ± 0.039 | |
| | Tarim_EMBA1 | Afanasievo | Geoksuyr_EN | 2.29E-03 | 0.443 ± 0.021 | 0.503 ± 0.044 | 0.055 ± 0.034 | Dzungaria_EBA1 |

**C. Modeling of ANE-rich pre-Bronze Age populations in Central Asia**

| Target | Ref1 | Ref2 | Ref3 | Pval | Coef(Ref1) | Coef(Ref2) | Coef(Ref3) | Extra outgroups |
|---|---|---|---|---|---|---|---|---|
| Botai_CA | Tarim_EMBA1 | Baikal_EBA | EHG | 2.57E-01 | 0.695 ± 0.057 | 0.056 ± 0.029 | 0.249 ± 0.034 | |
| | Tarim_EMBA1 | Baikal_EBA | EHG | 1.39E-01 | 0.554 ± 0.036 | 0.133 ± 0.019 | 0.312 ± 0.028 | AG3 |
| | Tarim_EMBA1 | Baikal_EBA | EHG | 1.90E-02 | 0.581 ± 0.039 | 0.121 ± 0.017 | 0.298 ± 0.029 | Afanasievo |
| | Tarim_EMBA1 | Baikal_EBA | EHG | 1.78E-01 | 0.639 ± 0.039 | 0.084 ± 0.020 | 0.277 ± 0.027 | Geoksyur_EN |
| West_Siberia_N | Tarim_EMBA1 | - | EHG | 3.92E-01 | 0.671 ± 0.030 | - | 0.329 ± 0.030 | |
| | Tarim_EMBA1 | - | EHG | 2.22E-01 | 0.674 ± 0.033 | - | 0.326 ± 0.033 | AG3 |
| | Tarim_EMBA1 | - | EHG | 2.81E-01 | 0.669 ± 0.030 | - | 0.331 ± 0.030 | Afanasievo |
| | Tarim_EMBA1 | - | EHG | 4.10E-01 | 0.680 ± 0.029 | - | 0.320 ± 0.029 | Geoksyur_EN |

We present details of key qpAdm admixture models reported in this study with alternative outgroup sets including sources from competing admixture models. "Coef" columns show the ancestry proportion and its standard error, calculated by 5 cM block jackknifing. Extra outgroup shows outgroups added to the base set. (A) Admixture models for Dzungaria_EBA do not change when BMAC/IAMC-related populations are added to the outgroup, supporting no contribution from them. (B) For Chemurchek and IAMC populations, we compare models including Dzungaria_EBA1 or Afanasievo as a competing source. Dzungaria_EBA1 works better for Chemurchek, Aigyrzhal_BA, Mereke_MBA, while Afanasievo works better for Kumsay_EBA. (C) Admixture models for Botai_CA and West_Siberia_N robustly hold when Afanasievo, Geoksyur_EN, or AG3 are included as an additional outgroup. Pval represents qpAdm p-value for the one-sided likelihood ratio test comparing the nested model (i.e. the target population is a mixture of the given references) with the nesting one (i.e. the target population cannot be sufficiently modeled as mixture of the given references). P-values are not multiple-testing corrected. Standard error measures were calculated with 5 cM block jackknifing.

# nature research

# Reporting Summary

Nature Research wishes to improve the reproducibility of the work that we publish. This form provides structure for consistency and transparency in reporting. For further information on Nature Research policies, see our Editorial Policies and the Editorial Policy Checklist.

## Statistics

For all statistical analyses, confirm that the following items are present in the figure legend, table legend, main text, or Methods section.

| n/a | Confirmed | |
|---|---|---|
| ☐ | ☒ | The exact sample size (*n*) for each experimental group/condition, given as a discrete number and unit of measurement |
| ☐ | ☒ | A statement on whether measurements were taken from distinct samples or whether the same sample was measured repeatedly |
| ☐ | ☒ | The statistical test(s) used AND whether they are one- or two-sided<br>*Only common tests should be described solely by name; describe more complex techniques in the Methods section.* |
| ☒ | ☐ | A description of all covariates tested |
| ☐ | ☒ | A description of any assumptions or corrections, such as tests of normality and adjustment for multiple comparisons |
| ☒ | ☐ | A full description of the statistical parameters including central tendency (e.g. means) or other basic estimates (e.g. regression coefficient) AND variation (e.g. standard deviation) or associated estimates of uncertainty (e.g. confidence intervals) |
| ☐ | ☒ | For null hypothesis testing, the test statistic (e.g. *F*, *t*, *r*) with confidence intervals, effect sizes, degrees of freedom and *P* value noted<br>*Give P values as exact values whenever suitable.* |
| ☒ | ☐ | For Bayesian analysis, information on the choice of priors and Markov chain Monte Carlo settings |
| ☒ | ☐ | For hierarchical and complex designs, identification of the appropriate level for tests and full reporting of outcomes |
| ☒ | ☐ | Estimates of effect sizes (e.g. Cohen's *d*, Pearson's *r*), indicating how they were calculated |

*Our web collection on statistics for biologists contains articles on many of the points above.*

## Software and code

Policy information about availability of computer code

| | |
|---|---|
| Data collection | Illumina sequence data were processed using the following programs to obtain genotype data used in the analysis: EAGER v1.92.55, AdapterRemoval v2.2.0, BWA v0.7.12, DeDup v0.12.2, bamUtils v1.0.13, mapDamage v2.0.9, pileupCaller v1.4.0.5 (https://github.com/stschiff/sequenceTools), Schmutzi v1.5.1, circularmapper v1.1, ANGSD v0.910. These programs are publicly available. |
| Data analysis | Calibration of AMS 14C dating results was done by OxCal v4.4, using the IntCal20 database. Population genetic data analysis in this study was performed using the following publicly available programs: lcMLkin v0.5.0, Geneious v11.1.3, HaploGrep2, bcftools v1.7, pysam v0.15.2, smartpca v16000, ADMIXTURE v1.3.0, PLINK v1.90, qp3Pop v435, qpDstat v755, hapROH v0.3a4 (https://pypi.org/project/hapROH/), DataGraph v4.5.1, qpWave v410, qpAdm v810, DATES v753. Non-default parameters used in our analysis are described in the Methods section. Protein mass spectrometry data analysis was performed using the following programs: MSConvert v3.0.11781, Mascot v2.6.0, Scaffold v4.9.0. |

For manuscripts utilizing custom algorithms or software that are central to the research but not yet described in published literature, software must be made available to editors and reviewers. We strongly encourage code deposition in a community repository (e.g. GitHub). See the Nature Research guidelines for submitting code & software for further information.

## Data

Policy information about availability of data

All manuscripts must include a data availability statement. This statement should provide the following information, where applicable:

- Accession codes, unique identifiers, or web links for publicly available datasets
- A list of figures that have associated raw data
- A description of any restrictions on data availability

The DNA sequences reported in this paper have been deposited in the European Nucleotide Archive (ENA) under accession PRJEB46875. Haploid genotype data of

April 2020

# Field-specific reporting

Please select the one below that is the best fit for your research. If you are not sure, read the appropriate sections before making your selection.

☐ Life sciences      ☐ Behavioural & social sciences      ☒ Ecological, evolutionary & environmental sciences

For a reference copy of the document with all sections, see nature.com/documents/nr-reporting-summary-flat.pdf

# Ecological, evolutionary & environmental sciences study design

All studies must disclose on these points even when the disclosure is negative.

| Study description | This study includes whole genome or genome-wide sequencing of 18 Early/Middle Bronze Age ancient individuals from Xinjiang, out of 33 skeletal elements screened, ranging between 3000 and 1700 BC. Sequencing coverage ranges 0.06-1.44x. Ancient genomes come from both the Dzungarian basin (n=5) and the Tarim basin (n=13). |
|---|---|
| Research sample | Research samples are composed of 18 ancient genomes from various archaeological sites in Xinjiang. They are chosen to cover the archaeological cultures so far excavated in northern and southern Xinjiang. We separated them into four analysis units based on their geographic origin, time period, and individual genetic profiles. The analysis units include: Dzungaria_EBA1 (n=3), Dzungaria_EBA2 (n=2), Tarim_EMBA1 (n=12), Tarim_EMBA2 (n=1). Among the 18 ancient individuals, there are six genetic males and 12 genetic females. Individual genetic sex information is available at the Extended Data Table 1. |
| Sampling strategy | Sample size was determined by the availability of relevant archaeological specimen and therefore no pre-selection of sample size was performed prior to the study. To produce ancient genomes reported in this study, we screened the accessible skeletal elements from the relevant geographic regions and time periods, and produced in-depth sequencing data for those with sufficient endogenous DNA preservation and without substantial contamination. |
| Data collection | Yinqiu Cui and Qiaomei Fu were present for data collection. Libraries that were prepared in Jilin (n=26) were directly shotgun sequenced on an Illumina HiSeq X10 or HiSeq 4000 instrument at the Novogene company, China, in the 150-bp paired-end sequencing design. Libraries prepared at IVPP, samples with 0.1% or more human DNA from the initial sequencing (n=7) were enriched at approximately 1.2 million nuclear SNPs and were sequenced on an Illumina HiSeq 4000 instrument in Beijing, China using 2x150bp chemistry. |
| Timing and spatial scale | Laboratory works and sequencing were conducted over the period from March 2018 to June 2019. Samples were taken from various archaeological sites in Xinjiang, China. Detailed information of the archaeological samples studied in this manuscript is provided in Fig. 1 and Supplementary Data S1-v4.xlsx. |
| Data exclusions | We excluded samples only if the samples do not meet the quality criteria, either by having low level of endogenous human DNA prohibiting genome-scale sequencing or by showing high level of contamination estimates. For population genetic analysis that requires exclusion of genetic relatives, we excluded closely related individuals (1st degree relatives) by removing one with lower coverage from each pair. |
| Reproducibility | We took multiple individuals from each archaeological site, if available, to support the representativeness of their genetic and dental proteomic profiles. For each sample, we estimated contamination level to support the authenticity of data. Importantly, two labs produced genome data of different individuals from the Xiaohe site: Jilin (n=4) and IVPP (n=7). Repeated genome data collection on the same individual is not considered as necessary nor useful in the field standard and therefore was not performed. |
| Randomization | Ancient genomes were first analyzed by each individual, and then were allocated into the analysis group based on their archaeological context, absolute date (14C dating), and their individual genetic profile. Randomization is not applicable because this study is observational and includes no treatment nor case/control comparison. |
| Blinding | There was no experimental treatment of samples involved in this study that requires blinding. Data analysis was performed based on the analysis groups that were defined by external information (archaeological context and date). |

Did the study involve field work?      ☐ Yes      ☒ No

# Reporting for specific materials, systems and methods

We require information from authors about some types of materials, experimental systems and methods used in many studies. Here, indicate whether each material, system or method listed is relevant to your study. If you are not sure if a list item applies to your research, read the appropriate section before selecting a response.

## Materials & experimental systems

| n/a | Involved in the study |
|---|---|
| ☒ | Antibodies |
| ☒ | Eukaryotic cell lines |
| ☐ | Palaeontology and archaeology |
| ☒ | Animals and other organisms |
| ☒ | Human research participants |
| ☒ | Clinical data |
| ☒ | Dual use research of concern |

## Methods

| n/a | Involved in the study |
|---|---|
| ☒ | ChIP-seq |
| ☒ | Flow cytometry |
| ☒ | MRI-based neuroimaging |

# Palaeontology and Archaeology

| | |
|---|---|
| Specimen provenance | The archaeological human remains studied in this manuscript were excavated by the Xinjiang Institute of Cultural Relics and Archaeology during 1979-2017. Scientific investigation of these remains were approved by the Xinjiang Cultural Relics and Archaeology Institute, which holds the custodianship of the studied remains, based on the written agreements. |
| Specimen deposition | The archaeological human remains studied in this manuscript are being housed in and managed by the Xinjiang Institute of Cultural Relics and Archaeology. |
| Dating methods | Of the 18 individuals reported this study, 10 were directly dated using accelerator mass spectrometry (AMS) at Beta Analytic, Miami, USA and/or at Lanzhou University, China. To confirm the reliability of our AMS datings, 4 out of the 10 individuals were AMS dated at both Beta Analytic and Lanzhou University. Consistent dates were obtained in all cases (Supplementary Data S1C). The calibration of the dated samples were performed based on the IntCal20 database and using the OxCal v4.4 program. |

☒ Tick this box to confirm that the raw and calibrated dates are available in the paper or in Supplementary Information.

| | |
|---|---|
| Ethics oversight | This study is based on previously excavated archaeological remains and included no new excavation effort nor study of live human or animal subjects. Therefore the study protocols used in this study are not the subject of approval by IRB/IACUC. The access to the remains was approved by the Xinjiang Institute of Cultural Relics and Archaeology based on the written agreements. |

Note that full information on the approval of the study protocol must also be provided in the manuscript.

