## [Peer Review File · Nature]

Manuscript Title: The genomic origins of the Bronze Age Tarim Basin mummies

Reviewer Comments & Author Rebuttals

Reviewer Reports on the Initial Version:

Referee #1 (Remarks to the Author):

Review:

This is a fascinating study, which introduces a number of key new human genomes that are of both general and specific interest to the scientific community, thus at its core, this is a study worthy of consideration in Nature. Generally, I think the paper is well executed and my understanding of extant archaeological evidence is in agreement with the final observation made by the authors – that the middle Bronze Age Tarim mummies (ca. 1800 > BC) represent local/primordial Eurasian genomic ancestry without a strong genetic link to the Afanasievo or BMAC (in direct demographic terms). Thus, the conclusion that the Tarim populations tested represent largely local (and deep time) genetic ancestry is interesting and important for many discussions relevant to Eurasian prehistory.

The article begins by positioning the study in terms of two “contrasting” arguments concerning the archaeological origins and (demographic) composition of high profile sites in the Tarim Basin – such as Gumougou and Xiaohe (which have been argued as speakers of Tocharian languages). These are cited as the “steppe origins” and “BMAC origins” models.

One should note that these models are generally seen as outdated and oversimplified “origins” models and do not reflect current thinking about the nature of connection in terms of (material) culture or society in the middle Bronze Age Tarim basin. The author's should highlight more recent revision and considerations of the archaeology, such as those by Betts et al 2019 on Xiaohe origins, for example, or Li, Yuqi 2020 on Xinjiang agricultural connections, references provided below.

The models currently referenced in the paper are also less favorable in the sense that they rely on outdated understandings of direct “migration” patterns. While the Yamnaya “migration” does appear to be a stochastic and rapid demographic displacement, the bulk of genomes recorded across the steppe (in the EBA or later BA) are (now) better understood as the result of interaction/admixture processes that take place over centuries if not millennia that do NOT include Yamnaya/Afanasievo ancestry, so it is not surprising that a simple vector of ancestry from either essentialized communities of the “Steppe” or “BMAC” will not provide easy fits for sites like Xiaohe. Other models with multiple components, outside of a direct Afanasievo source even, may be more likely. As such, I think the authors could productively revise their conceptual framing of the study by not relying too heavily upon the “Steppe Hypothesis or BMAC hypothesis” (each well over 15 years or more), but rather update the framework of their study on the basis of abundant new archaeological data, perhaps to test more specific and nuanced cases:

1) Updating the models tested

Given the recent advances in a number of areas of archaeological research, the most cited “current” models to explore might more productively be summarized as:

- a) an Afanasievo or Chemurchek admixtures for the Tarim sites (which the authors rightly undertake, for both EBA and LBA periods) --
- b) the Inner Asian Mt. Corridor Model (not tested, described below)

One should note, the “Afanasievo” should not be conflated with the “Steppe” in the 4th and 3rd

mill BC, (lines 97-101), since the archaeology AND genetics of the Early Bronze Age Steppe (in the strict geographic sense) are significantly different from those documented among the Yamnaya/Afanasievo populations [there are very few sites between these areas with similar genomic profiles, but rather Eneo/EBA sites like Botai, Dali, and Kanai are more representative of a Steppe substrate, with large % ANE ancestry.

In the paper, the authors model samples Dhungaria_EBA1 &2 (and test later Tarim samples) using Yamnaya/Afanasievo genetics – which makes perfectly good sense, but they should be careful not to conflate this with testing a “steppe” origin. (Using the “steppe” in non-specific terms makes things more confusing).

This should simply be called an “Afanasievo/Altai model.

Furthermore, newer more nuanced models, such as that presented by Betts et al. 2019, offer a much more complex intersection of likely communities who may have had diverse cultural influences at sites in EBA and BA Xinjiang, like Xiaohe. These deserve consideration in testing the genomic ancestry. In fact, Betts et al. builds largely on Frachetti’s 2012 “Inner Asian Mountain Corridor – or “IAMC” theory of complex interaction between agro-pastoralists throughout mountain areas spanning and (archaeologically) connecting communities from Southern Central Asia (e.g. Sarazm) and the Altai (Afanasievo and Chemurchek), and ultimately Xinjiang.

New genomic work carried out by Narasimhan et al 2019 (cited) across the interactive territory of the IAMC strongly supports the mountain chain as a vector for both human admixture as well as abundant archaeological elements found in BA Xinjiang, thus it is also worth testing, not least of all due to possible explanatory synergy with the proteomics data presented confirming pastoralism at Xiaohe. The argument that sheep/goat pastoralism (which likely diffused through the IAMC from the 3rd to 2nd mill BC) did not accompany genomic admixture is not, in fact, tested unless genomes from the IAMC are included in the admixture model as well (i.e., the Afanasievo are not the only source of sheep/goat)

If one were looking purely at the genomics of the Xiaohe samples (as presented), it looks very much the primordial ANE signature documented is also prevalent in various samples among Eurasian populations from the Neolithic and Chalcolithic (e.g. Botai), but also from the Early Bronze Age sites closer to Xinjiang (e.g. Dali, discussed below in detail).

2) Testing IAMC model for modeling the EBA + later Tarim:

Since at least 2010 the issues surrounding “western” influences in Xinjiang during the Bronze Age have been significantly updated on account of new and unambiguous archaeological facts which document clear transmissions of agricultural and pastoral products (plants/animals/practices), metal technologies, and other cultural practices along the so-called “Inner Asian Mountain Corridor” (coined by Frachetti 2012, *Current Anthro*). The “IAMC” is defined by the foothill territory spanning between the Pamir Mts and the Altai Mts, where the transfer of domesticated grains such as wheat, barley and millet, as well as human genetic admixture can be traced along clines both northeast and southwest from at least 3500-1000 BC, and importantly introducing wheat and barley to Xinjiang around 1800 BC (See comprehensive summary of BA agriculture in Xinjiang by Li, Y. 2020).

Of relevance to the current study, the northeast transfers of wheat and barley and sheep/goat along the IAMC are also now demonstrated among some Afanasievo communities in the Altai ca 3100 BC, while millet is well documented passing from NW Xinjiang to Southwest Central Asia along the IAMC at roughly the same time (ca. 2800 BC) (see supporting evidence by Zhou et al. 2020 (cited), Hermes et al. 2019, Hermes et al 2018, Spengler et al. 2014, Stevens et al. 2018, Liu X. et al. 2019). Cultural and economic connections along the IAMC perpetuated throughout the 3rd and 2nd millennia BC, and are considered by many as the source for the introduction

agropastoralism in the eastern Tian Shan and Xinjiang after 2000 BC, including at sites in the Tarim Basin. Wheat and Barley, in particular, recovered at Gumougou and Xiaohe are both seen morphologically as similar to assemblages recovered along the IAMC exchange system (Li 2020, Liu 2019, Spengler 2015). There is also strong evidence for a variety of additional cultural and technological links between IAMC sites and those of Xinjiang throughout both the 3rd and 2nd mill BC (e.g. significant similarity in textiles with Tarim sites, Doumani-Dupuy et al 2017).

The archaeological evidence is strong enough that the authors should consider how this recent array of evidence for connectivity along the IAMC figures into their admixture models, and might influence their conclusions.

3) qpAdam modeling including IAMC sites, Sarazm, Dali, Aigyrzhal-2, rather than solely "BMAC"

From a genetic perspective, the "Inner Asian Mt. Corridor" (IAMC) model provides relevant testable scenarios that should be considered in the admixture modelling for the Early Bronze Age Xinjiang (at least), and possibly later Bronze Age. Narasimhan et al (2019) show a clear genomic cline of shared ancestry between predominantly pre-Bronze Age steppe communities (coined WSHG, here ANE) and southwestern C. Asian communities (modelled as Iranian_neolithic) distributed along the IAMC in a bidirectional admixture as early as ~3000 BC.

Specifically, Narasimhan et al. 2019 model the EBA genome from Dali (Dali_EBA), which dates to the late Afanasievo period (ca. 2800 BC) as roughly 80% WSHG (similar to ANE) and ~20% Iranian (note: Jeong et al 2020 describe this Iranian neolithic component as "BMAC", even though Dali predates the BMAC by ~500yrs). This southern Central Asian ancestry can also be modeled using Sarazm, or Iran_EN). *[Note, The ancestry pie graph in the current article includes an additional element at Dali, modeling which is not discussed?] Other key genomes along the IAMC cline include Sarazm, Aigyrzhal-2, Dali, & Kanai, none of which are considered.

Admixture modeling of the Dzhungarian_EBA using these IAMC sites would at very least serve to confirm whether the small percentages of "Iranian" ancestry (like those found at northern sites along the IAMC) with abundant ANE ancestry (though smaller % at Sarazm) were contributing to populations in Xinjiang, either in the EBA or later. While the authors clearly show BMAC ancestry may be too distant genetically, admixed IAMC genomes with earlier Iranian elements may be more significant (given their reduced percent of Iranian_Neolithic ancestry). Parkhai (eneolithic) may also be useful for modeling.

Interestingly, Jeong et al 2020 directly relate the ancestry of the Dali_EBA sample (ca 2800 BC on the IAMC) to EBA communities of the southern Chemurchek in the Altai/NW Xinjiang, pointing out the existence of Iranian ancestry there ca. 2700 BC. Notably this ancestry mixture is mirrored in correlations of ceramic finds at both Dali and Chemurchek sites (see Betts et al 2019 for more discussion). In the outgrouping analysis of the current study, (Extended data Fig 2) Dali_EBA ranks in the top range for both Dzhungaria_EBA1, EBA2, and Chemurchek.

Yet in the current paper the the EBA genome from Dali was not included in any of the qpAdm modeling (at least it is not listed in the supplementary tables) in spite of the fact that the sites is only ~250 km west from the site of Nileke (Authors' sample Dzhungaria_EBA2), and is only 100 yrs different in chronological terms, and Chemurchek sites are modelled/illustrated wholly differently from Jeong et al, 2020 without a clear explanation as to why (in figure 3 for example)? This is confusing and should be explained, if there was an error in the Cell paper concerning the Chemurchek ancestry, or why such modeling has changed (see figure 3c).

Given the above, I was also confused as to why the authors chose to model Dzhungaria_EBA1 and EBA2, using exogenous Okunev ancestry, since Okunev genomes are much later (+800 yrs) than the EBA samples and not clearly linked archaeologically.

*[Please also note, the location of the Dali site is in the wrong place on Figure 1.], it is located in the northern Dzhungar Mountains.

Summary:

Given that there are archaeologically sound reasons to think that populations along the IAMC spread key domesticates to the Altai and Xinjiang through their relationships with neighboring groups (such as at sites for Dzhungaria_EBA1 and 2, Chemurchek, etc.), it seems important to include Dali_EBA, Sarazm, Aigyzhal-2, Kanai and other sites along the IAMC in the qpAdm model, to test/rule out these intermediate IAMC sites as genetic sources for either the EBA sites and later 2nd mill BC sites in Xinjiang. Given the abundant cultural ties between the IAMC and Xinjiang sites evident at (later) sites such as Xiaohe, Lop Nor, Gumougou and more (grains, ceramics, textiles, etc.), the high ANE component at Xiaohe needs to be explored as part of a wider geography of communities who are unrelated to Afanasievo ancestry.

Since agropastoralism (at least key domesticates) clearly passed through the IAMC into western China around the end of the 3rd millennium and continued to do so throughout the 2nd mill. BC, the authors should engage with this more nuanced "compromise" model that might serve as an intermediation between the more essential "BMAC" or "Afanasievo" explanations.

If after being modeled, there are no genetic components in Xinjiang that can be reliably linked to genomes from IAMC sites, then the authors should contextualize this fact in light of the abundant cultural transmissions that tied Xinjiang to the Mountain Corridor throughout the 3rd and 2nd mill. BC. Since the IAMC genomic cline has clear parallels in cultural transmission processes that have strong archaeological correlates in Xinjiang – it would be interesting to test if there was shared ancestry through time. The outgroup analysis would suggest as much, as well as the fact that the Dzhungaria_EIA sample has Iranian admixture.

One final note: It is true that there has been abundant popular speculation about the supposed "western" origins of the Tarim mummies, and much of this is spurred by relatively conjectural ideas about the "looks" of the mummies, their (apparent) hair color and other phenotypical factors (noted in lines 87-93). One should bear in mind that this conjectural line of argumentation has – to this reviewer's mind – never amounted to more than loose speculation and proponents of such speculations are usually taken with abundant "grains of salt", at least by regional specialists. In as much, I would not recommend devoting as many words to such viewpoints as viable reasons for this study, since they do not represent notions of the majority of archaeologists who have done research on the region in the last 10-15 years using actual scientific methods. Thus, while Prof. Mair, who is cited extensively on this point at the start of the paper, is a pioneering and preeminent Sinologist, he not a formally trained archaeologist and his popularized viewpoints espousing wildly distant western origins at Xiaohe (and Xinjiang generally, eg. Scotland on account of tartan textiles!) are not taken very seriously by many in the field. Therefore, one might not want to spotlight this in the current study and I recommend shortening or removing this portion of the lead-in to make room for more serious issues of debate, which the authors quickly turn to in their introduction. Anyway, I am sure such points will be raised during the media coverage on the article, and it is in this context that such speculations can be easily dismissed (esp. given the results presented here).

In general, I congratulate the authors on a fascinating piece of research and, with some updates and repositioning of the data, this paper will serve to illustrate many new things about the prehistoric population history of Xinjiang and Eurasia more broadly. I suggest that the paper be reviewed again before forwarding it for publication.

A few references mentioned (and not referenced in submission)

Doumani-Dupuy, P. et al. 2017 Eurasian textiles: Case studies in exchange during the incipient and later Silk Road periods *Quat. International* 468 <https://doi.org/10.1016/j.quaint.2016.09.067>

Frachetti MD (2012) Multiregional emergence of mobile pastoralism and nonuniform institutional complexity across Eurasia. *Curr Anthropol* 53:2–38. <https://doi.org/10.1086/663692>

Frachetti MD, Spengler RN, Fritz GJ, Mar'yashev AN (2010) Earliest direct evidence for broomcorn millet and wheat in the central Eurasian steppe region. *Antiquity* 84:993–1,010

Li, Yuqi. Agriculture and palaeoeconomy in prehistoric Xinjiang, China (3000–200 BC). *Veget Hist Archaeobot* (2020). <https://doi.org/10.1007/s00334-020-00774-2>

Liu X, Jones PJ, Motuzaitė Matuzevičiūtė G et al (2019) From ecological opportunism to multi-cropping: mapping food globalisation in prehistory. *Q Sci Rev* 206:21–28

Referee #2 (Remarks to the Author):

The current manuscript reports ancient DNA from the Xinjiang area of Central Asia. The paper seems to solve a major research question, namely the ancestry of the Tarim mummies in the light of their 'Western' physical appearance, association with Tocharian languages and noteworthy cultural elements. The study finds that, rather than supporting the predictions that would arise from two major hypothesis about the origins of the Tarim mummies, the ancestry of the group is neither derived from BMAC-associated individuals or Afanasievo steppe-related ancestry, but instead show strong links with Pleistocene 18-24 kya individuals from the Lake Baikal regions (Mal'ta and Afontova-Gora), which were the first ancient DNA to be described from this region (Raghavan et al. 2014). The ADMIXTURE analysis is striking, where Afontova-Gora3 appears to be a nearly perfect match to the Tarim mummies, despite the time and genetic drift separating them. This is very interesting, and sheds new light on the origins of this group for which there exists extensive archaeological information. In particular so, because it provides a plausible explanation for some of the previous observation, the 'Western' appearance is explicable by the indeed western genetic affinities of the Mal'ta and Afontova-Gora ~20kya individuals, but these were previously not known to have as close descendants present in the Holocene. In contrast, individuals from the Dzungarian basin show stronger affinities with Afanasievo.

I have several comments about the study as performed and reported:

I think the term indigenous should not be used in this manuscript. Surely indigeneity is primarily a concept relating to identity, and I do not see it as useful to apply it to ancient groups, even if here there is a suggestion of long-standing occupation. We do not know how the ancient individuals self-identified, and indeed people in prehistory with Afanasievo ancestry might have self-identified as indigenous as well. Indigenous could also be interpreted in different ways, including those which lose the scientific basis of the Tarim groups being closely related to Pleistocene Baikal and groups in Central Asia. Are present-day groups in the region only indigenous if they show substantial ancestry from the EMBA groups? In this particular case, there is a risk of harm being made (upon communication of the study to the public) to a currently persecuted group in a disputed region, the Uyghur, and so the communication involved is of prime importance. In this context, indigeneity becomes a concept that may be primarily about politics and identity, and of little or no scientific use for an ancient DNA study.

The qpAdm ancestry models are of central importance to the conclusions about ancestry, and all

other analyses (PCA, ADMIXTURE, outgroup-f3 etc) are primarily explorative in comparison. However, when different populations were tested as sources in qpAdm, other candidate source populations were not included in the list of reference populations. This 'rotation' is valuable as it shows that good-fit source populations are good fits also in context of other groups in direct model competition (Harney et al. 2020). Otherwise, if the choice of reference populations does not include the source populations that are competing, there are questions e.g. if the number of SNPs included makes a difference in rejected/non-rejected models.

Additionally, given how central the qpAdm models are, I would suggest including more text or display items describing how the competitive model testing was set up, which populations were included as candidate source populations, which were included as reference/outgroup populations, and how the rationale for model-testing proceeded. This should not be relegated to supplementary excel tables, and if space is needed, other analyses that do not add much, e.g. PCA, can be removed from the main article.

The main result is that the Tarim mummies are largely descendants of Pleistocene Baikal populations, but the authors do not cite the paper that centrally revealed this ancestry (Raghavan et al. 2014) in the main text. Furthermore, I find the ANE terminology unnecessary, in this case the link to LGM Baikal groups is fairly direct and the broader term doesn't seem needed.

I am puzzled by the apparently extreme patterns of low conditional nucleotide mismatch rate between the Tarim individuals. The authors note that the rate is similar to 1st degree relatives in other groups, but the groups don't show recent runs of homozygosity. I believe this result should be featured in the main text. The Extended Data Figure 4A also seems to be missing some colours in the legend. Is the mismatch rate as low as the most low-diversity groups today, e.g. in the Amazon or Oceania? What is the mismatch rate to Mal'ta and Afontova Gora: could the low diversity have originated in those Pleistocene populations?

Clarify that by formation of the Tarim_EMBA1 gene pool you are referring to the mixture between the two inferred sources. A date of 183 generations into the past seems quite ancient, beyond the limit of ~130 generations that have often been used as a rule-of-thumb in human genetics (see Loh et al. 2013, ALDER, and Moorjani et al., ROLOFF). Can the authors rule out that this admixture linkage disequilibrium is not confounded by the extremely low genetic diversity of Tarim_EMBA1? See e.g. Loh et al. 2013, Figure 5, for an analysis showing that intra-population linkage disequilibrium in the Andamanese can misleadingly be interpreted as admixture linkage disequilibrium.

I find the paper well and objectively written, and do not see indications of politics influencing the study. However, we can note that a natural question that remains unaddressed in this manuscript is the relationship between the ancient groups in the Tarim and Dzungarian basins to modern-day populations in Xinjiang, including notably the Turkic-speaking Uyghur which are widely recognized as persecuted by the Chinese government. Indeed, the question of the Tarim mummies has been political for decades. A higher-level explorative analysis with ADMIXTURE is reported in Extended Data Figure 3, but finer-resolution K clustering, or qpAdm ancestry models, that studies Xinjiang present-day populations and asks whether the Tarim and Dzungarian ancient individuals are closely related to them or not would be mandated, as the question of relationships to later ancestry arises naturally from the revelations in this ancient DNA study.

Is Figure 1E with a human remains necessary? Human remains should only be displayed if scientifically motivated.

The authors write " we successfully retrieved ancient genome sequences from 5 EBA Dzungarian individuals (3000-2800 BCE) and 13 Early-Middle Bronze Age (EMBA) Tarim individuals (2100-1700 BCE)". However, this is incorrect as these are not all genome sequences. Only 12 of the samples processed by the Jilin lab were directly whole-genome (shotgun) sequenced, and the ones

processed at IVPP were only enriched for a selected panel of 1240k SNPs. Given the importance of this material and the destructive sampling, it would be motivated to directly sequence whole-genome data from all individuals that are amenable to such analysis (e.g. using the >10% endogenous human DNA threshold applied in the Jilin processing).

Referee #3 (Remarks to the Author):

A. Summary of the key results

I see the key results of this paper being that the "Tarim mummies represent a culturally cosmopolitan but genetically isolated indigenous population" (Line 300-301). This strong statement, well supported by the data, demonstrates the great significance of this targeted study of the genomic history of the Tarim Basin Bronze Age populations. The findings challenge many longstanding hypotheses about the BA in this region, and based on this fact, I recommend it be published after some changes are made. Please see below for more details.

B. Originality and significance: if not novel, please include reference

Very significant but not well outlined. Emphasis needs to be shifted to make a greater impact and make the contribution clear for the journal's wider audience. This paper, if published, will be one among several aDNA studies newly published and therefore needs to move beyond the stated goal of "reconstructing the population histories" (Line 112) of a region. More contextualization with the broader problems of archaeology are needed, as well as clear comparison to the contrast/comparison in findings from other papers in this field.

C. Data & methodology: validity of approach, quality of data, quality of presentation

-Validity of approach: extremely valid approach, essential in order to address years of unsolved questions about human origins, population movement and large-scale/long-distance interaction during prehistory.

-Quality of data: High particularly given the challenges of access in Xinjiang.

-Quality of presentation: Moderate. Significance and aims of the study could be stated more clearly. As a regional specialist it is clear to me why this study is important, but for the wider readership of this journal, the contribution of the study may not be recognized.

D. Appropriate use of statistics and treatment of uncertainties

Not qualified to comment on appropriate use of statistics beyond the approach appearing to meet current standards for high quality aDNA analysis, and results are able to be compared with other large studies of aDNA, etc.

E. Conclusions: robustness, validity, reliability

Conclusions are strong, but do not link well to the Introduction or Summary. The organization of the paper is such that the main point of the paper is not clear until the end. Authors are relying too heavily on the reader consulting the supplementary material where the main sites, archaeology, important geographic features, chronology, and data sets are well laid out. However, the primary text does not adequately orient the reader.

F. Suggested improvements: experiments, data for possible revision

I have several suggestions to improve the readability and clarify of the manuscript. They are primarily organizational and suggestions for greater emphasis being placed on the conclusions/discussion.

-The introduction provides a brief description of the geographic area of study, both north and south Xinjiang and a discussion on the problem of the origin of the Tocharian language and associated populations. This focus does not adequately demonstrate the relevance for this choice of discussion. This could be moved down or rephrased to keep the reader on point.

-May include a clearer statement of research goals and question in the introduction. What is the problem being addressed? For instance, at Line 111, it's still unclear what the aims and research

questions of this paper are beyond reconstructing population history (as either supporting the Steppe or the BMAC hypothesis).

-(Line 95) Authors present the 2 main hypotheses for Xiaohu origins, which largely shape their hypothesis testing. However, the archaeology of Xinjiang (and neighboring regions) has developed significantly since these hypotheses were initially proposed, and hence the authors should contextualize their study with more recent opinions and proposed models.

-While the authors offer a brief description of the geography, it is not made clear where the sites that are part of their study are located (or what the sites are called). While this information is well outlined in the supplementary text, it should be included somehow in the main text body. A possible solution: Lines 89-93 mention the relevant TB sites, and so the authors could state that these are also the sites being used in their own study. They must do the same for the Dzungarian sites somewhere in the Introduction.

-The reason for conducting a combined aDNA and proteomics study is not well explained in the introduction, nor is the relevance for discussing the Tocharian language problem.

-Lines 124-143 describe the genetic profiles of samples, but it's unclear to me what is original data and what is pre-existing published data that is helping to build interpretations.

-Line 155 mentions Okunev for the first time. General readers will have to consult supplementary text in order to follow.

-The findings on lactase persistence within the Tarim Basin populations as well as the range of dairy sources are very interesting. But how does it impact on their research question?

Suggestions based on the Discussion content:

-Line 256 of the main text gives the first mention of the sites used for the study. Too late.

-The 3rd paragraph of discussion is very strong, but it should be moved (or something similarly composed) to the start of the paper to replace the current focus on geography and Tocharian language, etc.

-Authors don't give due emphasis to the impact of their findings on the long-term adherence to migration-focused models to explain the population history and archaeological patterns of Bronze Age Eurasia (particularly for Xinjiang). For instance, they suggest that the Tarim Basin mummies (of their study) provide the clearest representation of pre-pastoralist populations in eastern Eurasia, hence we have an indigenous population! This is a highly significant finding and the opportunity to really tackle these issues should not be overlooked. For example, how can we think differently about the processes of cultural exchange and technology transfer given such findings?

-Authors could give more time to discussing the millennia gap between the Afanasevo ancestry and the indigenous ancestry of the Tarim Basin populations (which they do briefly). For instance, what happens to the Afanasevo populations, what do they contribute, what is the relationship between the 2?

-The implications of this study's findings are significant, but they are not adequately showcased by the authors. I mention this because:

1. This robust and scientific study will make it possible to speak with much greater certainty about the BA in Xinjiang (+central Eurasia/Asia/east Asia). The traditional emphasis on "Western" influence on the prehistory of Xinjiang/population/material culture origins has shaped the academic and public dialogue since Mallory and Mair first wrote about it. Now these views are greatly challenged a strong scientific backing. This point should be emphasized.

2. We can put a lot of earlier hypotheses to bed and start to consider new possibilities for how the social dynamics played out in the BA. These are going to be more challenging to develop, but they are absolutely crucial if we are to produce to an accurate picture of what was happening.

3. Migration narratives – we now have a growing body of scientific evidence to suggest that populations movements were more varied. How do the author's findings differ from what other larger scale populations studies are finding/suggesting?

4. The site of Nileke is interesting because it suggests the lack of correlation between material culture and genetics. The new questions that emerge about the Afanasevo 'spread' based on this data is significant and could be emphasized.

G. References: appropriate credit to previous work?

Betts et al. 2019 (ref No.13) should be referenced for its examination and investigation of Tarim genetic ancestry, not just for the geographic description. Perhaps at Line 252.

H. Clarity and context: lucidity of abstract/summary, appropriateness of abstract, introduction and conclusions

The clarity and context of the study is sound. However, the presentation weakens this piece. This could be remedied with some reorganization and a shifted emphasis away from the points presented in the introduction and summary to those that are focused on in the discussion. (which I outline in detail above).

Referee #4 (Remarks to the Author):

I believe this is an extremely well-written manuscript that adds an important piece of the puzzle in our understanding of human prehistory.

I would like to say upfront that the size of the dataset (18 low coverage genomes) is not super impressive and the study is also not driven by methodological innovation. So we need to ask instead if results related to these particular mummies are important enough to warrant publication in the highest ranking journal? To answer that I hope that we have some archaeologists and/or linguists included in the reviewer panel that can comment on this. All I can say is that ever since I entered the field of ancient DNA research, I have heard talk about the Tarim mummies, and how they are central to the language origin debate, so despite not having a lot of knowledge on these mummies myself, I do believe that these results will have a wide impact in the fields of ancient DNA, archaeology and linguistics. The labwork meets the highest ancient DNA standards and the bioinformatic and analytical work is high quality too.

In my view (as an aDNA researcher), the paper presents two major results:

- 1) Shows a genetic link between the Afanasievo culture in Altai and the Dzungarian Basin
- 2) Shows that the Tarim mummies have no genetic link to Afanasievo

I think, the first result is just as important as the second one, despite not getting the same level of attention in the manuscript. The first result confirms a major hypothesis proposed in both archaeological and genetic literature about the Yamnaya-Afanasievo migrations extending beyond Altai and all the way to China. What a fantastic result! Congratulations! Why not have a map showing the extend of this massive east-west connection across the continents? It seems to me that the authors are a bit too eager to reject the Tarim-Afanasievo link that they forget just how amazingly close the previous archaeological and genetic studies have been in their predictions. For the same reason it would be interesting hearing more about these Dzungarian individuals in the main paper. How do they differ from Tarim culturally? Are they also mummies? Is the Afanasievo link a surprise (a quick look in the archeological supplement tells me that this is not a major surprise, but this is very relevant information to have in the main paper!).

The second result is mainly interesting in the light of how iconic these mummies are and their central position in the language debate. To put this into perspective though, they lived 1000 years after Afanasievo, so perhaps it is not super surprising that they don't have Afanasievo ancestry? I'm aware that this is not your own hypothesis that you are rejecting here, but it is still worth reflecting a bit on this gap in time? How likely is it actually that someone with Afanasievo ancestry

brought Tocharian to the Tarim basin if the population in Tarim was not established until 1000 years after Afanasievo vanished? In the light of your results, I think this gap in time deserves a few comments in the manuscript.

Secondly, the sampled Tarim individuals lived 3000 years before the Tocharian texts were produced (if I am reading the information in this manuscript correctly), so perhaps it is also worth reflecting on just how well these particular mummies could represent potential Tocharian speakers? As readers we need a better understanding as to why these mummies could be directly related to manuscripts produced by monks 3000 years later? I'm sure there have been many books written about this, but I think readers of Nature should be provided with this information directly in your manuscript.

Also, given that the history of the Tarim mummies cover 1800 years, could it not easily be that some of the later mummies from this region (closer in time to the production of the texts) would indeed display Afanasievo ancestry and spoke Tocharian? I'm aware that this is a counter-argument to my comment above about the 1000-year-gap, but it's just to highlight that there are large gaps on either side of "your" mummy samples when compared to the two events central to your discussion (i.e., Afanasievo presence in Altai, and production of manuscripts).

Similarly, could there be spatial population structure in the Tarim basin, so you would potentially have neighboring populations with Afanasievo ancestry? The mummies carry signatures of small population sizes and you also write somewhere that this is a very dry area with only few oases. Taken together this could suggest highly structured populations in this area? If you want to entirely reject an Afanasievo link to "everything Tarim", then it is important to discuss how confident you are that your 13 Tarim mummy samples are indeed representative for the populations in this basin through space and time? I think addressing these questions are essential to really understand the impact of your second finding. If not sufficient space in the main text, then perhaps draft a paragraph in the supplement section directly addressing these concerns.

Aside from the points above, I just want to emphasize again that it is indeed a result of major importance to be able to establish the ancestry of these mummies. As a reader I just need some context that allows me to evaluate to what extent these 13 early mummies are good representatives of the whole 1800-year history covered by the Tarim mummies.

For the same reasons I am not a big fan of this statement in the discussion:

"While the arrival and admixture of Afanasievo populations in the Dzungarian Basin of northern Xinjiang ca. 3000 BCE may have plausibly introduced Indo-European languages to the region, the material culture and genetic profile of the Tarim mummies from ca. 2100 BCE onwards call into question simplistic assumptions about the link between genetics and language and leaves unanswered the question of whether the Bronze Age Tarim populations spoke a form of proto-Tocharian."

It is very challenging to draw tight links between DNA and language because they are two proxies that can never fully be aligned (unless perhaps you find an actual book on the ancient individual you have sequenced...). In this case, we have no written sources from the Yamnaya/Afanasievo and we have no written sources from the Tarim mummies (as I understand it), but we are still trying to infer what language they spoke and how the language could have arrived there. Given the research history of these mummies, I do agree that your result has linguistic implications but I don't think you use the result correspondingly. In your study, you argue that Afanasievo likely introduced some form of Indo-European southwards but clearly they did not have contact with Tarim. In that case is the most parsimonious conclusion not that these mummies had nothing to do with Indo-European language? Since you are essentially rejecting the null-hypothesis that clearly had language implication (i.e. the Afanasievo-Indo-European link), shouldn't the alternative hypothesis also have language implications? Given your whole setup, I don't think it works well to

sit on the fence with this and simply say that more research is needed.

Some minor comments:

Figure 1: Can you make it clearer on these figures which samples are from your own project? Somehow make a visual distinction between your data and those published previously.

There is something wrong with this bit:

"Its eastward spread has been plausibly linked to the expansion of steppe pastoralists such as the Yamnaya (3500-2500 BCE) of the Pontic-Caspian steppe and the Afanasievo (3150-2750 BCE) of the Altai-Sayan region^{15,16}, who have close archaeological and genetic links¹⁷."

Mathieson et al. 2015 (ref 17) did not provide genetic data on the Afanasievo culture. The paper you have to cite here is Allentoft et al. (2015) which sequenced Afanasievo skeletons for the first time and thus established the close genetic link to Yamnaya. Your work is building directly on this study by extending the Afanasievo/Yamnaya ancestry even further (as predicted in the same study from 2015). Here is a relevant citation from Allentoft et al. (2015) that appear somewhat ignored in your current manuscript:

"It seems plausible that Afanasievo, with their genetic western (Yamnaya) origin, spoke an Indo-European language and could have introduced this southward to Xinjiang and Tarim (38). Importantly, however, although our results support a correspondence between cultural changes, migrations, and linguistic patterns, we caution that such relationships cannot always be expected but must be demonstrated case by case."

Author Rebuttals to Initial Comments:

Referee #1, archaeology expertise (Remarks to the Author):

Comment 1: This is a fascinating study, which introduces a number of key new human genomes that are of both general and specific interest to the scientific community, thus at its core, this is a study worthy of consideration in Nature. Generally, I think the paper is well executed and my understanding of extant archaeological evidence is in agreement with the final observation made by the authors – that the middle Bronze Age Tarim mummies (ca. 1800> BC) represent local/primordial Eurasian genomic ancestry without a strong genetic link to the Afanasievo or BMAC (in direct demographic terms). Thus, the conclusion that the Tarim populations tested represent largely local (and deep time) genetic ancestry is interesting and important for many discussions relevant to Eurasian prehistory.

We appreciate the reviewer's positive assessment on our study.

Comment 2: The article begins by positioning the study in terms of two "contrasting" arguments concerning the archaeological origins and (demographic) composition of high profile sites in the Tarim Basin – such as Gumougou and Xiaohe (which have been argued as speakers of Tocharian languages). These are cited as the "steppe origins" and "BMAC origins" models. One should note that these models are generally seen as outdated and oversimplified "origins" models and do not reflect current thinking about the nature of connection in terms of (material) culture or society in the middle Bronze Age Tarim basin. The authors should highlight more recent revision and considerations of the archaeology, such

as those by Betts et al 2019 on Xiaohe origins, for example, or Li, Yuqi 2020 on Xinjiang agricultural connections, references provided below.

We have updated the text to include the IAMC models described in Betts et al. 2019 and Li 2020. We have also reduced the emphasis placed on the Afanasievo steppe and BMAC oasis models, although we do still include them because they are well known and still widely cited. With our study's data, we are able to definitively refute them.

Comment 3: The models currently referenced in the paper are also less favorable in the sense that they rely on outdated understandings of direct “migration” patterns. While the Yamnaya “migration” does appear to be a stochastic and rapid demographic displacement, the bulk of genomes recorded across the steppe (in the EBA or later BA) are (now) better understood as the result of interaction/admixture processes that take place over centuries if not millennia that do NOT include Yamnaya/Afanasievo ancestry, so it is not surprising that a simple vector of ancestry from either essentialized communities of the “Steppe” or “BMAC” will not provide easy fits for sites like Xiaohe. Other models with multiple components, outside of a direct Afanasievo source even, may be more likely. As such, I think the authors could productively revise their conceptual framing of the study by not relying too heavily upon the “Steppe Hypothesis or BMAC hypothesis” (each well over 15 years or more), but rather update the framework of their study on the basis of abundant new archaeological data, perhaps to test more specific and nuanced cases. Updating the models tested. Given the recent advances in a number of areas of archaeological research, the most cited “current” models to explore might more productively be summarized as:

- a) an Afanasievo or Chemurchek admixtures for the Tarim sites (which the authors rightly undertake, for both EBA and LBA periods) --
- b) the Inner Asian Mt. Corridor Model (not tested, described below)

We have added additional models and consider more complex admixture processes than simple migration vectors. In addition to showing that the EMBA Tarim populations do not descend from geographically distant Afanasievo or BMAC populations, we also show that they do not descend from geographically proximal admixed descendent populations in Dzungaria (i.e., Dzungaria Afanasievo, Chemurchek) or the IAMC (i.e., Sarazm, Dali, and Aigyrzhal).

Comment 4: One should note, the “Afanasievo” should not be conflated with the “Steppe” in the 4th and 3rd mill BC, (lines 97-101), since the archaeology AND genetics of the Early Bronze Age Steppe (in the strict geographic sense) are significantly different from those documented among the Yamnaya/Afanasievo populations [there are very few sites between these areas with similar genomic profiles, but rather Eneo/EBA sites like Botai, Dali, and Kanai are more representative of a Steppe substrate, with large % ANE ancestry.

We have clarified and more clearly defined our usage of the term steppe. We have also updated Figure 1 to aid in being more geographically precise in the manuscript.

Comment 5: In the paper, the authors model samples Dhungaria_EBA1 &2 (and test later Tarim samples) using Yamnaya/Afanasievo genetics – which makes perfectly good sense, but they should be

careful not to conflate this with testing a “steppe” origin. (Using the “steppe” in non-specific terms makes things more confusing). This should simply be called an “Afanasiovo/Altai model.

Corrected. Please see our reply to the reviewer’s previous comment.

Comment 6: Furthermore, newer more nuanced models, such as that presented by Betts et al. 2019, offer a much more complex intersection of likely communities who may have had diverse cultural influences at sites in EBA and BA Xinjiang, like Xiaohe. These deserve consideration in testing the genomic ancestry. In fact, Betts et al. builds largely on Frachetti’s 2012 “Inner Asian Mountain Corridor – or “IAMC” theory of complex interaction between agro-pastoralists throughout mountain areas spanning and (archaeologically) connecting communities from Southern Central Asia (e.g. Sarazm) and the Altai (Afanasiovo and Chemurchek), and ultimately Xinjiang.

We have added a test of the IAMC model to our analyses.

Comment 7: New genomic work carried out by Narasimhan et al 2019 (cited) across the interactive territory of the IAMC strongly supports the mountain chain as a vector for both human admixture as well as abundant archaeological elements found in BA Xinjiang, thus it is also worth testing, not least of all due to possible explanatory synergy with the proteomics data presented confirming pastoralism at Xiaohe. The argument that sheep/goat pastoralism (which likely diffused through the IAMC from the 3rd to 2nd mill BC) did not accompany genomic admixture is not, in fact, tested unless genomes from the IAMC are included in the admixture model as well (i.e., the Afanasiovo are not the only source of sheep/goat)

Following the reviewer’s suggestion, we now incorporate the IAMC populations (e.g. Sarazm_EN, Dali_EBA, Kanai_MBA, and Aigyrzhal_BA) into our admixture modeling. Our major finding, which do not change by adding the IAMC populations into modeling, is that the Tarim_EMBA individuals do not have a genetic component that can be linked to populations with herding of sheep/goat, including Afanasiovo, IAMC, and pre-BMAC populations. Please find the model details in ‘Supplementary Data S1E’.

Comment 8: If one were looking purely at the genomics of the Xiaohe samples (as presented), it looks very much the primordial ANE signature documented is also prevalent in various samples among Eurasian populations from the Neolithic and Chalcolithic (e.g. Botai), but also from the Early Bronze Age sites closer to Xinjiang (e.g. Dali, discussed below in detail).

Dali_EBA or any other IAMC populations do not fit as source populations in admixture models explaining the ancestry composition of Tarim_EMBA. Instead, we find the opposite: modeling the IAMC populations as a mixture of Tarim_EMBA and other sources (including a BMAC/pre-BMAC source), fit the observed data well.

Comment 9: Testing IAMC model for modeling the EBA + later Tarim. Since at least 2010 the issues surrounding “western” influences in Xinjiang during the Bronze Age have been significantly updated on account of new and unambiguous archaeological facts which document clear transmissions of agricultural and pastoral products (plants/animals/practices), metal technologies, and other cultural practices along the so-called “Inner Asian Mountain Corridor” (coined by Frachetti 2012, *Current Anthro*). The “IAMC” is defined by the foothill territory spanning between the Pamir Mts and the Altai Mts, where the transfer of domesticated grains such as wheat, barley and millet, as well as human genetic admixture can be traced along clines both northeast and southwest from at least 3500-1000 BC, and importantly introducing wheat and barley to Xinjiang around 1800 BC (See comprehensive summary of BA agriculture in Xinjiang by Li, Y. 2020).

See our answers to Comments 7 and 8.

Comment 10: Of relevance to the current study, the northeast transfers of wheat and barley and sheep/goat along the IAMC are also now demonstrated among some Afanasievo communities in the Altai ca 3100 BC, while millet is well documented passing from NW Xinjiang to Southwest Central Asia along the IAMC at roughly the same time (ca. 2800 BC) (see supporting evidence by Zhou et al. 2020 (cited), Hermes et al. 2019, Hermes et al 2018, Spengler et al. 2014, Stevens et al. 2018, Liu X. et al. 2019). Cultural and economic connections along the IAMC perpetuated throughout the 3rd and 2nd millennia BC, and are considered by many as the source for the introduction agropastoralism in the eastern Tian Shan and Xinjiang after 2000 BC, including at sites in the Tarim Basin. Wheat and Barley, in particular, recovered at Gumougou and Xiaohe are both seen morphologically as similar to assemblages recovered along the IAMC exchange system (Li 2020, Liu 2019, Spengler 2015). There is also strong evidence for a variety of additional cultural and technological links between IAMC sites and those of Xinjiang throughout both the 3rd and 2nd mill BC (e.g. significant similarity in textiles with Tarim sites, Doumani-Dupuy et al 2017). The archaeological evidence is strong enough that the authors should consider how this recent array of evidence for connectivity along the IAMC figures into their admixture models, and might influence their conclusions.

Following the reviewer’s suggestion, we co-analyzed the IAMC populations from Narasimhan et al (2019; *Science*) with our data set. As described above, we find a close genetic connection between Tarim_EMBA and the IAMC populations. However, our modeling shows that the IAMC populations are a mixture of Tarim_EMBA, a BMAC/pre-BMAC-related source, and an Afanasievo/Dzungaria_EBA related one (Supplementary Data S1G). The Xiaohe population may indeed have been culturally influenced by the surrounding IAMC populations and this may be the origin of their agricultural crops and possibly also livestock, but the Xiaohe population did not descend from these IAMC populations. To date, all studied IAMC sites show admixture with a BMAC/pre-BMAC-like source that is absent in the Xiaohe population.

Comment 11: qpAdm modeling including IAMC sites, Sarazm, Dali, Aigyrzhal-2, rather than solely “BMAC”. From a genetic perspective, the “Inner Asian Mt. Corridor” (IAMC) model provides relevant testable scenarios that should be considered in the admixture modelling for the Early Bronze Age Xinjiang (at least), and possibly later Bronze Age. Narasimhan et al (2019) show a clear genomic cline of shared ancestry between predominantly pre-Bronze Age steppe communities (coined WSHG, here ANE) and southwestern C. Asian communities (modelled as Iranian_neolithic) distributed along the IAMC in a bidirectional admixture as early as ~3000 BC. Specifically, Narasimhan et al. 2019 model the EBA genome from Dali (Dali_EBA), which dates to the late Afanasievo period (ca. 2800 BC) as roughly 80% WSHG (similar to ANE) and ~20% Iranian (note: Jeong et al 2020 describe this Iranian

neolithic component as “BMAC”, even though Dali predates the BMAC by ~500yrs). This southern Central Asian ancestry can also be modeled using Sarazm, or Iran_EN). *[Note, The ancestry pie graph in the current article includes an additional element at Dali, modeling which is not discussed?] Other key genomes along the IAMC cline include Sarazm, Aigyrzhal-2, Dali, & Kanai, none of which are considered.

We now present admixture modeling involving Sarazm_EN, Aigyrzhal_BA, Dali_EBA, and Kanai_MBA (Supplementary Data S1D, S1F-G).

Comment 12: Admixture modeling of the Dzhungarian_EBA using these IAMC sites would at very least serve to confirm whether the small percentages of “Iranian” ancestry (like those found at northern sites along the IAMC) with abundant ANE ancestry (though smaller % at Sarazm) were contributing to populations in Xinjiang, either in the EBA or later. While the authors clearly show BMAC ancestry may be too distant genetically, admixed IAMC genomes with earlier Iranian elements may be more significant (given their reduced percent of Iranian_Neolithic ancestry). Parkhai (eneolithic) may also be useful for modeling.

Following the reviewer’s suggestion, we performed extensive admixture modeling of Dzungaria_EBA groups with IAMC and pre-BMAC populations (including Parkhai_EN) as a source. Still, we find that Dzungaria_EBA groups are best modeled as Tarim_EMBA + Afanasievo + DevilsCave_N/Baikal_EBA, with no further input from BMAC/pre-BMAC/IAMC populations. Genetically speaking, BMAC/pre-BMAC/IAMC populations share (partially for the IAMC groups) an ancestry component closely related to the ancient Iranian/Caucasus populations. Our modeling confirms that no such ancestry component is required to explain the Dzungaria_EBA individuals.

Comment 13: Interestingly, Jeong et al 2020 directly relate the ancestry of the Dali_EBA sample (ca 2800 BC on the IAMC) to EBA communities of the southern Chemurchek in the Altai/NW Xinjiang, pointing out the existence of Iranian ancestry there ca. 2700 BC. Notably this ancestry mixture is mirrored in correlations of ceramic finds at both Dali and Chemurchek sites (see Betts et al 2019 for more discussion). In the outgrouping analysis of the current study, (Extended data Fig 2) Dali_EBA ranks in the top range for both Dzhungaria_EBA1, EBA2, and Chemurchek. Yet in the current paper the the EBA genome from Dali was not included in any of the qpAdm modeling (at least it is not listed in the supplementary tables) in spite of the fact that the site is only ~250 km west from the site of Nileke (Authors’ sample Dzhungaria_EBA2), and is only 100 yrs different in chronological terms, and Chemurchek sites are modelled/illustrated wholly differently from Jeong et al, 2020 without a clear explanation as to why (in figure 3 for example)? This is confusing and should be explained, if there was an error in the Cell paper concerning the Chemurchek ancestry, or why such modeling has changed (see figure 3c).

We updated our modeling of Chemurchek, Dali and other IAMC populations from the ones presented in Jeong et al 2020 to the current one mainly because Tarim_EMBA, newly reported in this study, is a pivotal population to model Chemurchek and other BA Inner Asian populations. We now explain the difference between Jeong et al 2020 models and our current ones in the main text. Our updated model presented in this manuscript shows both Afanasievo-related and pre-BMAC/Iranian-related ancestry components in Chemurchek and the IAMC populations (Supplementary Data S1F-G). We accordingly updated the Figure 3C.

Comment 14: Given the above, I was also confused as to why the authors chose to model Dzhungaria_EBA1 and EBA2, using exogenous Okunev ancestry, since Okunev genomes are much later (+800 yrs) than the EBA samples and not clearly linked archaeologically.

We agree with the reviewer's opinion that Okunevo may not be relevant to model Dzhungaria_EBA. In the previous version, we used Okunevo only as a distal source. We now removed the parts from the main text to prevent readers' confusion.

Comment 15: *[Please also note, the location of the Dali site is in the wrong place on Figure 1.], it is located in the northern Dzhungar Mountains.

Thank you for finding our mistake. We corrected the location of the Dali site on Figure 1.

Comment 16: Summary. Given that there are archaeologically sound reasons to think that populations along the IAMC spread key domesticates to the Altai and Xinjiang through their relationships with neighboring groups (such as at sites for Dzhungaria_EBA1 and 2, Chemurchek, etc.), it seems important to include Dali_EBA, Sarazm, Aigyzhal-2, Kanai and other sites along the IAMC in the qpAdm model, to test/rule out these intermediate IAMC sites as genetic sources for either the EBA sites and later 2nd mill BC sites in Xinjiang. Given the abundant cultural ties between the IAMC and Xinjiang sites evident at (later) sites such as Xiaohe, Lop Nor, Gumougou and more (grains, ceramics, textiles, etc.), the high ANE component at Xiaohe needs to be explored as part of a wider geography of communities who are unrelated to Afanasievo ancestry. Since agropastoralism (at least key domesticates) clearly passed through the IAMC into western China around the end of the 3rd millennium and continued to do so throughout the 2nd mill. BC, the authors should engage with this more nuanced "compromise" model that might serve as an intermediation between the more essential "BMAC" or "Afnasievo" explanations.

We now present updated admixture modeling results including Dali_EBA, Sarazm_EN, Aigyrzhal_BA, Kanai_MBA, and other relevant sites in and around IAMC. As described above, these populations do not serve as a source to explain the genetic composition of Tarim_EMBA (Xiaohe): instead, Tarim_EMBA is found to be representative of the ANE source in these populations together with Afanasievo- and pre-BMAC-related ancestry components, explaining the observed close relationship between IAMC and Tarim_EMBA. We report these findings and discuss their implication in the main text.

Comment 17: If after being modeled, there are no genetic components in Xinjiang that can be reliably linked to genomes from IAMC sites, then the authors should contextualize this fact in light of the abundant cultural transmissions that tied Xinjiang to the Mountain Corridor throughout the 3rd and 2nd mill. BC. Since the IAMC genomic cline has clear parallels in cultural transmission processes that have strong archaeological correlates in Xinjiang – it would be interesting to test if there was shared ancestry through time. The outgroup analysis would suggest as much, as well as the fact that the Dzhungaria_EIA sample has Iranian admixture.

Please see the above. With regard to Dzungaria_EIA, we caution that the Iranian-related ancestry in Iron Age Central Asians may not be transmitted from the Early Bronze Age phenomenon: this is observed in a wide region across the Eurasian steppe, as reported in Jeong et al 2020. Although we do not find Iranian/BMAC/IAMC related ancestry in EBA Dzungarian populations, we do find this ancestry in later Chemurchek populations.

Comment 18: One final note: It is true that there has been abundant popular speculation about the supposed “western” origins of the Tarim mummies, and much of this is spurred by relatively conjectural ideas about the “looks” of the mummies, their (apparent) hair color and other phenotypical factors (noted in lines 87-93). One should bear in mind that this conjectural line of argumentation has – to this reviewer’s mind – never amounted to more than loose speculation and proponents of such speculations are usually taken with abundant “grains of salt”, at least by regional specialists. In as much, I would not recommend devoting as many words to such viewpoints as viable reasons for this study, since they do not represent notions of the majority of archaeologists who have done research on the region in the last 10-15 years using actual scientific methods. Thus, while Prof. Mair, who is cited extensively on this point at the start of the paper, is a pioneering and preeminent Sinologist, he not a formally trained archaeologist and his popularized viewpoints espousing wildly distant western origins at Xiaohe (and Xinjiang generally, eg. Scotland on account of tartan textiles!) are not taken very seriously by many in the field. Therefore, one might not want to spotlight this in the current study and I recommend shortening or removing this portion of the lead-in to make room for more serious issues of debate, which the authors quickly turn to in their introduction. Anyway, I am sure such points will be raised during the media coverage on the article, and it is in this context that such speculations can be easily dismissed (esp. given the results presented here).

We shortened our discussion of the physical appearance of the Tarim mummies and added more information about the cultural connections between IAMC and Xinjiang.

Comment 19: In general, I congratulate the authors on a fascinating piece of research and, with some updates and repositioning of the data, this paper will serve to illustrate many new things about the prehistoric population history of Xinjiang and Eurasia more broadly. I suggest that the paper be reviewed again before forwarding it for publication.

Thank you for your comments.

Comment 20: A few references mentioned (and not referenced in submission):

- Doumani-Dupuy, P. et al. 2017 Eurasian textiles: Case studies in exchange during the incipient and later Silk Road periods. *Quat. International* 468 <https://doi.org/10.1016/j.quaint.2016.09.067>
- Frachetti MD (2012) Multiregional emergence of mobile pastoralism and nonuniform institutional complexity across Eurasia. *Curr Anthropol* 53:2–38. <https://doi.org/10.1086/663692>
- Frachetti MD, Spengler RN, Fritz GJ, Mar’yashev AN (2010) Earliest direct evidence for broomcorn millet and wheat in the central Eurasian steppe region. *Antiquity* 84:993–1,010

- Li, Yuqi. Agriculture and palaeoeconomy in prehistoric Xinjiang, China (3000–200 BC). *Veget Hist Archaeobot* (2020). <https://doi.org/10.1007/s00334-020-00774-2>
- Liu X, Jones PJ, Motuzaite Matuzeviciute G et al (2019) From ecological opportunism to multi-cropping: mapping food globalisation in prehistory. *Q Sci Rev* 206:21–28

We have added the following recommended references to relevant sections of the main text: Frachetti 2012, Frachetti et al. 2010, and Li 2020. We did not add Doumani-Dupuy et al because it focuses on periods after those in our study. We did not add Liu et al because it focuses on a macroregional perspective, and we instead cited other archaeobotanical studies directly focusing on the IAMC and Xinjiang.

Referee #2, paleogenomics expertise (Remarks to the Author):

The current manuscript reports ancient DNA from the Xinjiang area of Central Asia. The paper seems to solve a major research question, namely the ancestry of the Tarim mummies in the light of their ‘Western’ physical appearance, association with Tocharian languages and noteworthy cultural elements. The study finds that, rather than supporting the predictions that would arise from two major hypothesis about the origins of the Tarim mummies, the ancestry of the group is neither derived from BMAC-associated individuals or Afanasievo steppe-related ancestry, but instead show strong links with Pleistocene 18-24 kya individuals from the Lake Baikal regions (Mal’ta and Afontova-Gora), which were the first ancient DNA to be described from this region (Raghavan et al. 2014). The ADMIXTURE analysis is striking, where Afontova-Gora3 appears to be a nearly perfect match to the Tarim mummies, despite the time and genetic drift separating them. This is very interesting, and sheds new light on the origins of this group for which there exists extensive archaeological information. In particular so, because it provides a plausible explanation for some of the previous observation, the ‘Western’ appearance is explicable by the indeed western genetic affinities of the Mal’ta and Afontova-Gora ~20kya individuals, but these were previously not known to have as close descendants present in the Holocene. In contrast, individuals from the Dzungarian basin show stronger affinities with Afanasievo. I have several comments about the study as performed and reported:

Comment 21: I think the term indigenous should not be used in this manuscript. Surely indigeneity is primarily a concept relating to identity, and I do not see it as useful to apply it to ancient groups, even if here there is a suggestion of long-standing occupation. We do not know how the ancient individuals self-identified, and indeed people in prehistory with Afanasievo ancestry might have self-identified as indigenous as well. Indigenous could also be interpreted in different ways, including those which lose the scientific basis of the Tarim groups being closely related to Pleistocene Baikal and groups in Central Asia. Are present-day groups in the region only indigenous if they show substantial ancestry from the EMBA groups? In this particular case, there is a risk of harm being made (upon communication of the study to the public) to a currently persecuted group in a disputed region, the Uyghur, and so the communication involved is of prime importance. In this context, indigeneity becomes a concept that may be primarily about politics and identity, and of little or no scientific use for an ancient DNA study.

We understand that the term indigenous is a relative term, and we use it here to refer to an ancestry profile that has existed within a place for millennia. We do not use it with the intent of indicating self-identity, but rather to distinguish the ancestry component observed in the Tarim population compared

to the ancestry components observed in other groups who migrated more recently into the region. We have added a clarifying statement to the text.

Comment 22: The qpAdm ancestry models are of central importance to the conclusions about ancestry, and all other analyses (PCA, ADMIXTURE, outgroup-f3 etc) are primarily explorative in comparison. However, when different populations were tested as sources in qpAdm, other candidate source populations were not included in the list of reference populations. This 'rotation' is valuable as it shows that good-fit source populations are good fits also in context of other groups in direct model competition (Harney et al. 2020). Otherwise, if the choice of reference populations does not include the source populations that are competing, there are questions e.g. if the number of SNPs included makes a difference in rejected/non-rejected models.

We now present additional qpAdm modeling results that include alternative source populations as extra outgroups to verify stability of key admixture models.

Comment 23: Additionally, given how central the qpAdm models are, I would suggest including more text or display items describing how the competitive model testing was set up, which populations were included as candidate source populations, which were included as reference/outgroup populations, and how the rationale for model-testing proceeded. This should not be relegated to supplementary excel tables, and if space is needed, other analyses that do not add much, e.g. PCA, can be removed from the main article.

We prefer to keep descriptive analyses like PCA in the main text because they provide a useful visual summary of the findings and help general readers understand the main message of the study. Because qpAdm models are difficult to visualize in a way that is straightforward to understand, we have instead updated our explanations of the qpAdm modeling in the main text to make our logic more straightforward and clear.

Comment 24: The main result is that the Tarim mummies are largely descendants of Pleistocene Baikal populations, but the authors do not cite the paper that centrally revealed this ancestry (Raghavan et al. 2014) in the main text. Furthermore, I find the ANE terminology unnecessary, in this case the link to LGM Baikal groups is fairly direct and the broader term doesn't seem needed.

We cited Raghavan et al 2014 in the Methods section, but now we add a citation for it when the Mal'ta genome is first mentioned (line 255). We would like to keep the term ANE because this ancestry does not seem to have been limited to the Baikal region, as shown by our finding of Tarim_EMBA in Xinjiang and the earlier report of the Afontova Gora genomes in the Upper Yenisei River Basin.

Comment 25: I am puzzled by the apparently extreme patterns of low conditional nucleotide mismatch rate between the Tarim individuals. The authors note that the rate is similar to 1st degree relatives in other groups, but the groups don't show recent runs of homozygosity. I believe this result should be featured in the main text. The Extended Data Figure 4A also seems to be missing some colours in the legend. Is the mismatch rate as low as the most low-diversity groups today, e.g. in the Amazon or

Oceania? What is the mismatch rate to Mal'ta and Afontova Gora: could the low diversity have originated in those Pleistocene populations?

We provide an updated version of the Extended Data Figure 4. We add pairwise mismatch rates of present-day Karitiana (from Amazon) and Papuan, as well as that of MA1-AG2-AG3. Pleistocene ANE hunter-gatherers (MA1-AG2-AG3) and Holocene Inner Eurasian populations (Afanasiovo, Baikal_EN, Baikal_EBA, Botai_CA, West_Siberia_N) all show substantially higher pairwise mismatch rates than Tarim_EMBA, supporting our interpretation that it reflects a population bottleneck specific to Tarim_EMBA. The genetic diversity within Tarim_EMBA is comparable to contemporary Amazonian (Karitiana) and Oceanian (Papuan) individuals.

Comment 26: Clarify that by formation of the Tarim_EMBA1 gene pool you are referring to the mixture between the two inferred sources. A date of 183 generations into the past seems quite ancient, beyond the limit of ~130 generations that have often been used as a rule-of-thumb in human genetics (see Loh et al. 2013, ALDER, and Moorjani et al., ROLOFF). Can the authors rule out that this admixture linkage disequilibrium is not confounded by the extremely low genetic diversity of Tarim_EMBA1? See e.g. Loh et al. 2013, Figure 5, for an analysis showing that intra-population linkage disequilibrium in the Andamanese can misleadingly be interpreted as admixture linkage disequilibrium.

The method we used for dating admixture in Tarim_EMBA1, DATES (<https://github.com/priyamoorejani/DATES>), presented by Narasimhan et al 2019, is based on a principle different from that of ALDER/ROLLOFF. In short, it models each individual's genotype as a linear combination of the allele frequencies of the two source populations and uses the regression residual as a proxy for its local ancestry. Then, it fits an exponential decay of the correlation of local ancestry state between markers over genetic distance in each individual, instead of decay of admixture LD across individuals. We highlight that the ALDER paper (the method mentioned by the reviewer) presents simulation results up to 200 generations with successful inference of admixture dates (Table S1 in Loh et al. 2013). Likewise, the above-mentioned figure 5 of Loh et al is for a special case in ALDER that the target itself is used as one of the two sources. The paper states "... We observed distinct weighted LD curves when analyzing the Onge, an indigenous population of the Andaman Islands. However, this curve is present only when using Onge themselves as one reference ...". Because our results are neither based on ALDER nor using the target itself as a source, we do not think the reviewer's critique is relevant to our analysis.

Comment 27: I find the paper well and objectively written, and do not see indications of politics influencing the study. However, we can note that a natural question that remains unaddressed in this manuscript is the relationship between the ancient groups in the Tarim and Dzungarian basins to modern-day populations in Xinjiang, including notably the Turkic-speaking Uyghur which are widely recognized as persecuted by the Chinese government. Indeed, the question of the Tarim mummies has been political for decades. A higher-level explorative analysis with ADMIXTURE is reported in Extended Data Figure 3, but finer-resolution K clustering, or qpAdm ancestry models, that studies Xinjiang present-day populations and asks whether the Tarim and Dzungarian ancient individuals are closely related to them or not would be mandated, as the question of relationships to later ancestry arises naturally from the revelations in this ancient DNA study.

We believe that a direct connection between Bronze Age Xinjiang individuals and Uyghurs is unlikely given the results of previous studies and written historical records indicating multiple waves of human migration and changes in gene pools between 4,000 years ago and today in Xinjiang. Therefore, we believe that an attempt to model the ancestry of present-day Uyghur with Tarim_EMBA/Dzungaria_EBA is unsound and prone to political misuse.

Comment 28: Is Figure 1E with a human remains necessary? Human remains should only be displayed if scientifically motivated.

We replaced figure 1E to show a coffin boat burial that illustrates the various features described in the text (e.g., woolen garments).

Comment 29: The authors write " we successfully retrieved ancient genome sequences from 5 EBA Dzungarian individuals (3000-2800 BCE) and 13 Early-Middle Bronze Age (EMBA) Tarim individuals (2100-1700 BCE)" . However, this is incorrect as these are not all genome sequences. Only 12 of the samples processed by the Jilin lab were directly whole-genome (shotgun) sequenced, and the ones processed at IVPP were only enriched for a selected panel of 1240k SNPs. Given the importance of this material and the destructive sampling, it would be motivated to directly sequence whole-genome data from all individuals that are amenable to such analysis (e.g. using the >10% endogenous human DNA threshold applied in the Jilin processing).

We corrected the text to "... 5 EBA Dzungarian individuals (3000-2800 BCE) and genome-wide data from 13 Early-Middle ...".

Referee #3, archaeology expertise (Remarks to the Author):

A. Summary of the key results.

Comment 30: I see the key results of this paper being that the "Tarim mummies represent a culturally cosmopolitan but genetically isolated indigenous population" (Line 300-301). This strong statement, well supported by the data, demonstrates the great significance of this targeted study of the genomic history of the Tarim Basin Bronze Age populations. The findings challenge many longstanding hypotheses about the BA in this region, and based on this fact, I recommend it be published after some changes are made. Please see below for more details.

Thank you for the reviewer's positive assessment of our work.

B. Originality and significance: if not novel, please include reference.

Comment 31: Very significant but not well outlined. Emphasis needs to be shifted to make a greater impact and make the contribution clear for the journal's wider audience. This paper, if published, will be one among several aDNA studies newly published and therefore needs to move beyond the stated goal of "reconstructing the population histories" (Line 112) of a region. More contextualization with the broader problems of archaeology are needed, as well as clear comparison to the contrast/comparison in findings from other papers in this field.

We have significantly revised our introduction and discussion to add contextualization and to make our arguments and hypotheses more straightforward and clear.

C. Data & methodology: validity of approach, quality of data, quality of presentation.

Comment 32: -Validity of approach: extremely valid approach, essential in order to address years of unsolved questions about human origins, population movement and large-scale/long-distance interaction during prehistory.

We thank the reviewer for their positive assessment.

Comment 33: -Quality of data: High particularly given the challenges of access in Xinjiang.

We thank the reviewer for their positive assessment.

Comment 34: -Quality of presentation: Moderate. Significance and aims of the study could be stated more clearly. As a regional specialist it is clear to me why this study is important, but for the wider readership of this journal, the contribution of the study may not be recognized.

We have significantly revised our introduction and discussion to make our significance and aims more clear.

D. Appropriate use of statistics and treatment of uncertainties

Comment 35: Not qualified to comment on appropriate use of statistics beyond the approach appearing to meet current standards for high quality aDNA analysis, and results are able to be compared with other large studies of aDNA, etc.

Checked.

E. Conclusions: robustness, validity, reliability

Comment 36: Conclusions are strong, but do not link well to the Introduction or Summary. The organization of the paper is such that the main point of the paper is not clear until the end. Authors are relying too heavily on the reader consulting the supplementary material where the main sites, archaeology, important geographic features, chronology, and data sets are well laid out. However, the primary text does not adequately orient the reader.

We now introduce our main questions in the introduction, and we have significantly revised our introduction and discussion to make the connections between these sections stronger.

F. Suggested improvements: experiments, data for possible revision

Comment 37: I have several suggestions to improve the readability and clarify of the manuscript. They are primarily organizational and suggestions for greater emphasis being placed on the conclusions/discussion.

-The introduction provides a brief description of the geographic area of study, both north and south Xinjiang and a discussion on the problem of the origin of the Tocharian language and associated populations. This focus does not adequately demonstrate the relevance for this choice of discussion. This could be moved down or rephrased to keep the reader on point.

We have rewritten the introduction to introduce our main questions and to better outline the problem we are solving. We have also reduced the focus on Indo-European language family dispersal and have added additional archaeological context.

Comment 38: -May include a clearer statement of research goals and question in the introduction. What is the problem being addressed? For instance, at Line 111, it's still unclear what the aims and research questions of this paper are beyond reconstructing population history (as either supporting the Steppe or the BMAC hypothesis).

We have significantly updated the introduction to better explain our research goals and questions.

Comment 39: -(Line 95) Authors present the 2 main hypotheses for Xiaohe origins, which largely shape their hypothesis testing. However, the archaeology of Xinjiang (and neighboring regions) has developed significantly since these hypotheses were initially proposed, and hence the authors should contextualize their study with more recent opinions and proposed models.

We have updated our hypotheses to also include an extended discussion of the Inner Asian Mountain Corridor and related population models.

Comment 40: -While the authors offer a brief description of the geography, it is not made clear where the sites that are part of their study are located (or what the sites are called). While this information is well outlined in the supplementary text, it should be included somehow in the main text body. A possible solution: Lines 89-93 mention the relevant TB sites, and so the authors could state that these are also the sites being used in their own study. They must do the same for the Dzungarian sites somewhere in the Introduction.

We have added the site names to the Introduction and we have also updated Figure 1 to make our sampling and study design more clear.

Comment 41: -The reason for conducting a combined aDNA and proteomics study is not well explained in the introduction, nor is the relevance for discussing the Tocharian language problem.

We have updated the text to clarify that although subsequent Tarim populations practiced dairy pastoralism, it was not known whether the founding Xiaohe population was also dairying. This question is important for understanding the cultural connections of the Xiaohe to neighboring populations.

Comment 42: -Lines 124-143 describe the genetic profiles of samples, but it's unclear to me what is original data and what is pre-existing published data that is helping to build interpretations.

We have updated Figure 1 and the discussion to make these distinctions more clear.

Comment 43: -Line 155 mentions Okunev for the first time. General readers will have to consult supplementary text in order to follow.

We have removed this analysis because it is unnecessary. See response to Comment 14.

Comment 44: -The findings on lactase persistence within the Tarim Basin populations as well as the range of dairy sources are very interesting. But how does it impact on their research question?

Lactase persistence is often presumed to be a prerequisite for dairy-based subsistence. However, we show that the Tarim population lacks this genetic trait and yet still practices dairying. This adds to a growing body of evidence that lactase persistence was not necessary for dairy-based subsistence in prehistory. This directly challenges current medical models of lactose intolerance.

Suggestions based on the Discussion content:

Comment 45: -Line 256 of the main text gives the first mention of the sites used for the study. Too late.

We now describe the sites in the introduction.

Comment 46: -The 3rd paragraph of discussion is very strong, but it should be moved (or something similarly composed) to the start of the paper to replace the current focus on geography and Tocharian language, etc.

We now introduce these main points in the introduction.

Comment 47: -Authors don't give due emphasis to the impact of their findings on the long-term adherence to migration-focused models to explain the population history and archaeological patterns of Bronze Age Eurasia (particularly for Xinjiang). For instance, they suggest that the Tarim Basin mummies (of their study) provide the clearest representation of pre-pastoralist populations in eastern Eurasia, hence we have an indigenous population! This is a highly significant finding and the opportunity to really tackle these issues should not be overlooked. For example, how can we think differently about the processes of cultural exchange and technology transfer given such findings?

We have added greater emphasis for these findings in the results and discussion sections.

Comment 48: -Authors could give more time to discussing the millennia gap between the Afanasevo ancestry and the indigenous ancestry of the Tarim Basin populations (which they do briefly). For instance, what happens to the Afanasevo populations, what do they contribute, what is the relationship between the 2?

We have added a discussion of ancestry modeling for the Chemurchek, a population that succeeds the Afanasievo in the Dzungarian Basin.

Comment 49: -The implications of this study's findings are significant, but they are not adequately showcased by the authors. I mention this because:

1. This robust and scientific study will make it possible to speak with much greater certainty about the BA in Xinjiang (+central Eurasia/Asia/east Asia). The traditional emphasis on "Western" influence on the prehistory of Xinjiang/population/material culture origins has shaped the academic and public dialogue since Mallory and Mair first wrote about it. Now these views are greatly challenged a strong scientific backing. This point should be emphasized.

We thank the reviewer for their positive assessment and we have added greater emphasis to these points in the Results and Discussion.

Comment 50: 2. We can put a lot of earlier hypotheses to bed and start to consider new possibilities for how the social dynamics played out in the BA. These are going to be more challenging to develop, but they are absolutely crucial if we are to produce to an accurate picture of what was happening.

We thank the reviewer for their positive assessment.

Comment 51: 3. Migration narratives – we now have a growing body of scientific evidence to suggest that populations movements were more varied. How do the author’s findings differ from what other larger scale populations studies are finding/suggesting?

Our study findings identify complex and contrasting patterns of migration and admixture in different populations. For example, we find that Xiaohe represents a local population that remained isolated without recent admixture, despite extensive population mixture occurring among their neighboring populations. Additionally, we note that Afanasievo in the Altai-Sayan region typically show very little admixture with local groups (which may reflect a true, long-distance migration event), while those in Dzungaria extensively admixed with local groups (which perhaps reflects a slower more gradual process of expansion). Admixture among IAMC populations likely proceeded slowly over many generations without discrete admixture events, but these groups did not mix with those in Dzungaria until the formation of the Chemurchek. We have tried to highlight these findings more clearly in the text.

Comment 52: 4. The site of Nileke is interesting because it suggests the lack of correlation between material culture and genetics. The new questions that emerge about the Afanasevo ‘spread’ based on this data is significant and could be emphasized.

The Nileke individuals (Dzungaria_EBA2) are genetically similar to the Dzungaria_EBA1 individuals (Ayituoan and Songshugou), but with a higher proportion of Tarim_EMBA1-related ancestry and with a lower proportion of Afanasievo-related one. We have made this point and its implications more explicit in the Results and Discussion.

G. References: appropriate credit to previous work?

Comment 53: Betts et al. 2019 (ref No.13) should be referenced for its examination and investigation of Tarim genetic ancestry, not just for the geographic description. Perhaps at Line 252.

We have added an extended analysis of Betts et al.’s IAMC model and have added additional citations in the text.

H. Clarity and context: lucidity of abstract/summary, appropriateness of abstract, introduction and conclusions

Comment 54: The clarity and context of the study is sound. However, the presentation weakens this piece. This could be remedied with some reorganization and a shifted emphasis away from the points presented in the introduction and summary to those that are focused on in the discussion. (which I outline in detail above).

We have reorganized the introduction and discussion to emphasize the points originally only highlighted in the Discussion.

Referee #4, population genetics and evolutionary genomics expertise (Remarks to the Author):

Comment 55: I believe this is an extremely well-written manuscript that adds an important piece of the puzzle in our understanding of human prehistory. I would like to say upfront that the size of the dataset (18 low coverage genomes) is not super impressive and the study is also not driven by methodological innovation. So we need to ask instead if results related to these particular mummies are important enough to warrant publication in the highest ranking journal? To answer that I hope that we have some archaeologists and/or linguists included in the reviewer panel that can comment on this. All I can say is that ever since I entered the field of ancient DNA research, I have heard talk about the Tarim mummies, and how they are central to the language origin debate, so despite not having a lot of knowledge on these mummies myself, I do believe that these results will have a wide impact in the fields of ancient DNA, archaeology and linguistics. The labwork meets the highest ancient DNA standards and the bioinformatic and analytical work is high quality too.

Thank you for the reviewer's positive assessment on our work.

Comment 56: In my view (as an aDNA researcher), the paper presents two major results:

- 1) Shows a genetic link between the Afanasievo culture in Altai and the Dzungarian Basin
- 2) Shows that the Tarim mummies have no genetic link to Afanasievo

I think, the first result is just as important as the second one, despite not getting the same level of attention in the manuscript. The first result confirms a major hypothesis proposed in both archaeological and genetic literature about the Yamnaya-Afanasievo migrations extending beyond Altai and all the way to China. What a fantastic result! Congratulations!

Thank you for the reviewer's positive assessment on our work.

Comment 57: Why not have a map showing the extend of this massive east-west connection across the continents? It seems to me that the authors are a bit too eager to reject the Tarim-Afanasio link that they forget just how amazingly close the previous archaeological and genetic studies have been in their predictions. For the same reason it would be interesting hearing more about these Dzungarian individuals in the main paper. How do they differ from Tarim culturally? Are they also mummies? Is the Afanasio link a surprise (a quick look in the archeological supplement tells me that this is not a major surprise, but this is very relevant information to have in the main paper!).

The Dzungarian individuals indeed belong to the Afanasio culture based on the archaeological contexts (described in the SI Text). We now clarify this and add sentences to discuss its implication in the main text (see the section “an extensive early Afanasio genetic influence in northern Xinjiang”). We have also updated Figure 1 to more clearly indicate the east-west connectivity of the continent.

Comment 58: The second result is mainly interesting in the light of how iconic these mummies are and their central position in the language debate. To put this into perspective though, they lived 1000 years after Afanasio, so perhaps it is not super surprising that they don't have Afanasio ancestry? I'm aware that this is not your own hypothesis that you are rejecting here, but it is still worth reflecting a bit on this gap in time? How likely is it actually that someone with Afanasio ancestry brought Tocharian to the Tarim basin if the population in Tarim was not established until 1000 years after Afanasio vanished? In the light of your results, I think this gap in time deserves a few comments in the manuscript.

In our Discussion, we highlight that the study of ancient Tarim populations that are temporally closer to the Tocharian texts is critical. We agree with the reviewer that many other scenarios are feasible that link EBA Afanasio impact in Dzungaria with later Tocharian speakers without posing the early Xiaohe horizon population as an intermediate link. However, our results reject the genetic Afanasio link of the early Xiaohe horizon in the representative sites of Xiaohe and Beifang.

Comment 59: Secondly, the sampled Tarim individuals lived 3000 years before the Tocharian texts were produced (if I am reading the information in this manuscript correctly), so perhaps it is also worth reflecting on just how well these particular mummies could represent potential Tocharian speakers? As readers we need a better understanding as to why these mummies could be directly related to manuscripts produced by monks 3000 years later? I'm sure there have been many books written about this, but I think readers of Nature should be provided with this information directly in your manuscript.

We agree with the reviewer that a link between Afansio, Tarim mummies, and the Tocharian speakers (which is not our hypothesis) is overly simplistic and multiple scenarios are currently feasible due to limited genetic, archaeological, and linguistic information. We rephrased our Introduction to present this idea as a (widely cited) hypothesis, but not a well-proven fact from either archaeology, linguistics, or genetics.

Comment 60: Also, given that the history of the Tarim mummies cover 1800 years, could it not easily be that some of the later mummies from this region (closer in time to the production of the texts) would indeed display Afanasio ancestry and spoke Tocharian? I'm aware that this is a counter-argument to my comment above about the 1000-year-gap, but it's just to highlight that there are large gaps on either

side of “your” mummy samples when compared to the two events central to your discussion (i.e., Afanasievo presence in Altai, and production of manuscripts).

We acknowledge that it is possible that the later Tarim mummies may show a different genetic profile from the earliest ones that we investigated in this study. We believe that it is not a topic that we can answer with certainty based on our data. We call for further archaeogenomic studies of Xinjiang to fill in the temporal gap in our Discussion.

Comment 61: Similarly, could there be spatial population structure in the Tarim basin, so you would potentially have neighboring populations with Afanasievo ancestry? The mummies carry signatures of small population sizes and you also write somewhere that this is a very dry area with only few oases. Taken together this could suggest highly structured populations in this area? If you want to entirely reject an Afanasievo link to “everything Tarim”, then it is important to discuss how confident you are that your 13 Tarim mummy samples are indeed representative for the populations in this basin through space and time? I think addressing these questions are essential to really understand the impact of your second finding. If not sufficient space in the main text, then perhaps draft a paragraph in the supplement section directly addressing these concerns.

Although we acknowledge that a potential population stratification in the Tarim Basin cannot be completely excluded, the tight genetic connection between Xiaohe and Gumugou in the eastern edge of the Tarim Basin and Beifang in the central Tarim Basin, a distance of 600 km, provides a substantial clue on the shared genetic history of the Xiaohe horizon sites across the Tarim Basin. In addition, the three cemeteries (Xiaohe, Beifang and Gumugou) investigated in this study are the only Bronze Age cemeteries excavated so far in the Tarim Basin. In the Discussion, we also call for further genetic studies on the ancient Tarim Basin populations.

Comment 62: Aside from the points above, I just want to emphasize again that it is indeed a result of major importance to be able to establish the ancestry of these mummies. As a reader I just need some context that allows me to evaluate to what extent these 13 early mummies are good representatives of the whole 1800-year history covered by the Tarim mummies.

Please see our responses in the above.

Comment 63: For the same reasons I am not a big fan of this statement in the discussion: “While the arrival and admixture of Afanasievo populations in the Dzungarian Basin of northern Xinjiang ca. 3000 BCE may have plausibly introduced Indo-European languages to the region, the material culture and genetic profile of the Tarim mummies from ca. 2100 BCE onwards call into question simplistic assumptions about the link between genetics and language and leaves unanswered the question of whether the Bronze Age Tarim populations spoke a form of proto-Tocharian.” It is very challenging to draw tight links between DNA and language because they are two proxies that can never fully be aligned (unless perhaps you find an actual book on the ancient individual you have sequenced...). In this case, we have no written sources from the Yamnaya/Afanasievo and we have no written sources from the Tarim mummies (as I understand it), but we are still trying to infer what language they spoke and how the language could have arrived there. Given the research history of these mummies, I do agree that your result has linguistic implications but I don’t think you use the result correspondingly. In your study,

you argue that Afanasievo likely introduced some form of indo-european southwards but clearly they did not have contact with Tarim. In that case is the most parsimonious conclusion not that these mummies had nothing to do with indo-european language? Since you are essentially rejecting the null-hypothesis that clearly had language implication (i.e. the Afanasievo-indo-european link), shouldn't the alternative hypothesis also have language implications? Given your whole setup, I don't think it works well to sit on the fence with this and simply say that more research is needed.

We agree that genetics and language do not necessarily correlate, and this is a point we wish to emphasize. Previous researchers have attempted to make explicit and uncritical links between the Tocharian language family, the Afanasievo, and the Tarim mummies and our results problematize such findings by showing that the Afanasievo and Tarim mummies have no biological relationship and by questioning the logic of uncritical genetic-linguistic assumptions. We have rephrased the Introduction and Discussion to make our critique more clear.

Some minor comments:

Comment 64: Figure 1: Can you make it clearer on these figures which samples are from your own project? Somehow make a visual distinction between your data and those published previously.

We provide a new Figure 1 that clearly marks newly reported ancient individuals.

Comment 65: There is something wrong with this bit: "Its eastward spread has been plausibly linked to the expansion of steppe pastoralists such as the Yamnaya (3500-2500 BCE) of the Pontic-Caspian steppe and the Afanasievo (3150-2750 BCE) of the Altai-Sayan region^{15,16}, who have close archaeological and genetic links¹⁷."

We have rephrased this sentence.

Comment 66: Mathieson et al. 2015 (ref 17) did not provide genetic data on the Afanasievo culture. The paper you have to cite here is Allentoft et al. (2015) which sequenced Afanasievo skeletons for the first time and thus established the close genetic link to Yamnaya. Your work is building directly on this study by extending the Afanasievo/Yamnaya ancestry even further (as predicted in the same study from 2015). Here is a relevant citation from Allentoft et al. (2015) that appear somewhat ignored in your current manuscript: "It seems plausible that Afanasievo, with their genetic western (Yamnaya) origin, spoke an Indo-European language and could have introduced this southward to Xinjiang and Tarim (38). Importantly, however, although our results support a correspondence between cultural changes, migrations, and linguistic patterns, we caution that such relationships cannot always be expected but must be demonstrated case by case."

Thank you for detecting our mistake. We replaced the citation (ref #17) to Allentoft et al 2015.

Reviewer Reports on the First Revision:

Referee #1 (Remarks to the Author):

Review Overview:

This paper is much improved from the first version, and a number of new models have been illustrated. My comments are focused primarily on the IMAC modeling aspect of the paper, which was newly added on account of my earlier comments. The updated results are fascinating and highly illuminating, but the authors should make some further clarification concerning their use of the IAMC model, and not just add it as simple additional hypothesis to test. This is primarily an issue of re-framing the introduction of the IMAC model to note explicitly that it is NOT a "migration" model in any sense, so adding some explanatory nuance to the mechanisms underlying the IAMC as a bridging model that can explain cultural transmission without necessary genetic transmission. This will serve then to reinforce the way it is brought up in the discussion at the end of the paper.

Detailed comments:

Line 114-117:

This is an oversimplification the IAMC model since the IAMC, as proposed by Frachetti 2012 (and used by Li 2020) does NOT make any claims about "founder populations" for Xiaohe (or Xinjiang) or anywhere, in terms of genetics. Rather Frachetti's argument is that the IMAC is a geographic corridor and vector for cultural influence, interaction, technology and trade, among (agro) pastoralists channeling innovations along the mountains between the Steppe, Southern Central Asia, and Xinjiang, starting in the 4th millennium BC and continuing in complex ways throughout the 3rd and 2nd millennia BC. (The authors correctly cite the model in their paper on line 320.)

Geographically, it is important to note that the IAMC extends to the Altai mountains and incorporates Afanasievo communities in the economic sense (wheat and barley have been documented at the Afanasievo site TongTian Cave, around 3200 BCE.

New archaeological evidence suggests such interactions may have taken place even earlier, in the Neolithic with the coming of domesticated animals to the IAMC around 6000 BC in Kyrgystan (Taylor et al, 2021). Evidence for this vector of interaction in the Early and middle Bronze Age is now very robust, especially in terms of cultural practices such as farming, wheat, barley and millet grains, and other aspects of agro-pastoral economy, such as herding (Taylor et al 2021, Hermes et al 2018). Other archaeological data show, in a number of material forms from the Early Middle and Late Bronze age, that the flows of interaction are complex and multi-directional during each period – as described by Frachetti 2012. Thus, in the archaeological sense, there are undeniable data to show linkages between, for e.g. the IAMC and the Chemurchek and IAMC and later Bronze Age sites in Xinjiang, and THIS is the argument made again by Li 2020 (but he does NOT make a genetic origins argument).

Neither Frachetti nor Li, in fact, make any comment about the genetic source of the people of Xinjiang, only the social, economic, and cultural connections. Thus, introducing the IAMC model, as suggested in my first review of this paper, was not intended to be a test for genetic sources per se, but as a way to understand the degree to which genetics and these cultural linkages were related.

The authors actually achieve this goal quite well, and the contribution of your paper then, should not be to prove or disprove models of genetic origin, especially since the "IAMC model" was not presented as a genetic migration model, and definitely not what Frachetti and later authors ever

actually argued.

Rather your paper can productively show how that there were different mechanisms of human sociality (meaning genetic admixture on the one hand, and cultural exchanges on the other), and these are decoupled in the context of late Bronze Age Xinjiang. In the earlier phase, as you note in the the discussion, the process is different, as there are cultural traits that accompany the Afanasievo genetics, along with local admixture.

Your data ultimately show that the genetic processes and cultural processes can be different at different times and that interaction in cultural terms does not ALWAYS move along with genetics. I would suggest that you reposition your use of the IAMC model as such, since it was not intended to be a genetic model. You have this worded better at the end of the paper, than at the beginning. Editing the hypothesis testing portion at the beginning to match with the conclusions at lines 300-350, would make your argument less dismissive and more productive in terms of how we understand the different mechanism underlying the genomic record you present for the early and later Bronze ages.

In fact, in this way the IAMC model actually provides a good working framework in favor of your paper, since it provides a mechanism to explain how many cultural practices can be shared -- e.g. the proteomics data you have for pastoralism is an example of this -- WITHOUT demanding a direct genetic link among the people who are exchanging. The IAMC model fully allows for genetics and proteomics to work together better.

In the Early Bronze Age, the situation appears more complicated. There appear to be BOTH archaeological links between northern IAMC (Dali_EBA) and Chemurchek, as well as genetic diverse links within that population (as described in Jeong et al. 2020 and the current paper). Yet, these genetic links are NOT evident in your Dzhungarian EBA samples- even though they are geographically closer. This suggests a complex set of regional affiliations whereby Afanasievo communities integrated to different degrees with local communities (as you discuss this in part in lines 185-187). Again, framing the discussion of your data in this fashion will make it more explanatory for archaeologists reading your paper.

I suggest that the authors try to reword how they introduce and use the IAMC model (in distinction from the Steppe or BMAC model) as a regional geographic context for cultural exchange, and leverage it to explain how genetics and culture do not necessarily match. Again, as it was never proposed as a genetic source, but rather a geographic arena for social complexity. It should definitely NOT be coupled with the BMAC model (it is currently written with a slash, which has little logic behind it).

That said, Betts et al. do look to the geography of the IAMC as a potential source for the Tarim Middle Bronze Age genetics, yet she is basing her hypothesis mostly on the cultural factors which I just noted, and she then only suggests a possible genetic link between them. Your data illustrates that a genetic affinity was not the case, that is interesting, but her model is most fundamentally an interaction model for the origins of Xiaohe cultural traits (namely agropastoralism). Indeed, she argues that much of the Xiaohe culture is likely derived from a local population. As it stands, it seems like you are disproving a model that few of us are claiming to be the case in the first place.

She states: "The Xiaohe culture appears to have emerged in the Tarim Basin without obvious antecedents but, like the Qiemu'erqieke culture, it too may have absorbed elements of a pre-Bronze Age indigenous population".

She also states "How the distinctive Qiemu'erqieke culture developed is still uncertain, but its ancestral derivation in the Altai is clear."

Rather than simply set up her argument to dismiss it, it would be more productive to use her

observations to say something like "although the IAMC appears to be a vector for cultural and economic factors evident throughout Xinjiang, the known sites from the IAMC do not provide a direct source of ancestry for later bronze Age populations in Xinjiang, these are interestingly traced to an ANE substrate that was present throughout Eurasia, likely in the early Holocene". As just explained above.

It is important that the authors use and cite these archaeological works carefully, and familiarize themselves more deeply with the nuanced arguments made in them. If not, they risk misrepresentation by oversimplifying them.

Line 251-266:

This is an important observation but one that can be somewhat confusing from an archaeological perspective. The new proposed qpADM model uses, for example, Tarim EMBA1 as a source/significant component for modeling Botai, and other EBA steppe genomes. But this is somewhat misleading since Tarim_EMBA POST-dates Botai by nearly 2000 years, and others by 1000 yrs or more. I understand that this is an exercise to demonstrate that Tarim EMBA1 can be used as proxy for an earlier, primordial ANE population on account of its isolation, but the argumentation here can appear somewhat circular to the non-specialist. Fact: The later Bronze age Xinjiang genomes do appear to be dominated by a primordial ANE ancestry. Fact: We don't know when this widespread ANE Neolithic substrate begins to differentiate in different regions of Eurasia. We only know that ANE was shared between the Central steppe (Botai), IAMC (Dali-EBA), and (later) Xiaohe (among others).

These sites span from 3500, 2600, and 1800BC respectively. So, it appears that the dominate ANE substrate begin to first diversify in the 4th-3rd millennium – specifically with the arrival of the Yamnaya/Afanasievo genetics in Altai, and Iranian Genetics in the IAMC, and yet NEITHER of these admixtures appear to impact the ANCESTRY in Xiaohe, as you note it remains isolated genetically. However, modeling the ANE component of earlier steppe groups with Tarim EMBA is a bit odd, since ideally, we would have an earlier source that matches. I am not sure how to solve this, but the issue seems to be illustrating that steppe populations with high ANE ancestry have a common (and not entirely clearly located) source population that was isolated and persisted without significant admixture ONLY in the Tarim basin region.

The paper ends on a note that makes a lot of sense, and it seems that the collective data presented from Xiaohe and other Tarim EMBA sites has opened the door for more conversation. Bravo!

Main changes requested.

1) Reframe how the IAMC model is introduced in the start – change it from a “genetic source” to test, to a process that helps understand how the many aspects of the cultural character of the region were formed. Edit the wording and fix the citations to reflect what the IMAC model actually proposes, and differentiate its use by Frachetti 2012 from Betts et al. Betts et al cites the model as an interaction model and goes on to propose that the geography might provide a source for genetics (note, this could still be the case if we had earlier Neolithic samples reflecting the ANE substrate with pre-Iranina admixture). At present, we don't know that. As you note in the end, (line 320) The IAMC works best in this explanatory fashion and can be used to bridge the archaeology and the genetic findings.

2) Testing the IAMC sites as sources for your samples in Xinjiang has utility (thank you!) and it is important to state that IAMC sites are not good fits as sources for Xinjiang (generally), either in the EBA or MBA periods. However, you should briefly contextualize this as a case where evidence for social and cultural interaction and genetics do not fit neatly, and say a few more words about the links that do exist, namely with Qiermuekek genomes.

3) Clarify the new qpADM modeling section, at least to acknowledge that there is a chronological and geographic issue with the use of TarimEMBA as a source of earlier steppe genomes. At least make it clearer, so that the remodeled steppe sites are not misunderstood by non-specialists.

Referee #2 (Remarks to the Author):

I thank the authors for their responses, but have to note that the revised manuscript is only marginally improved as most of the previous comments seem rejected. The central conclusion continues to remain sound and highly fascinating.

-I continue to think that the use of the term indigenous is not needed and stands the risk of misrepresentation to the public, as it has a very specific meaning in many regions of the world. I have not seen it commonly used in ancient DNA studies, but reserved for modern groups that self-identify as indigenous. Other alternatives are primordial as reviewer 1 uses, or possibly autochthonous.

-I continue to hold the opinion that the qpAdm analyses are key to the paper, but note that they are in the revision still not primarily done in the state-of-the-art approach of rotating populations from the outgroup/reference position to the source position. Such rotation is the primary way to let different models compete with each other. Further the authors reject the suggestion to highlight these analyses as main figures.

-I also note that the authors reject generating the highest quality-standard data that is clearly possible for many samples in this material, shotgun sequencing. The current data is sufficient to answer the main hypotheses, but not the best possible data that could have been economically produced and left as legacy for the many researchers that will be interested in incorporating this information to their own studies.

Referee #3 (Remarks to the Author):

The authors have placed much effort into addressing the many concerns raised by the referees, mine included. My initial concerns had to do with a lack of: clear statement of significance/broad impacts/importance; contextualization of the relevant archaeology, attention given to recent population models and hypotheses related to their topic; and overall organized of the paper. The authors have provided a major reworking of the introduction, results and discussion to address these concerns to my satisfaction. They have also shifted the paper's emphasis away from language/Indo-European dispersals, which were problematic. The revised Figure 1 is also very well done and suitable for readers to follow.

I just have one criticism to make. The authors offer conflicting descriptions of the burial architecture for Xiaohe and Gumugou. They state that oar-shaped planks were erected at the head of female burials and single posts erected at the head of male burials. But, then in another section of the paper the description is inverted (the oar-shaped planks are assigned to the males, and regular posts to females). While it's most often the case that the poles represent males and the paddles/oars females – sometimes these are reversed. Hence, we can't generalize, or else, keep consistent. The authors can reference the paper by Abuduresule et al. 2019 (in The Cultures of

Ancient Xinjiang, Western China: Crossroads of the Silk Roads, Editors Betts et al.).

Referee #4 (Remarks to the Author):

I believe the authors have addressed my concerns and suggested edits very well and I have no further comments. I believe this manuscript is well suited for publication in Nature.

Author Rebuttals to First Revision:

Referee #1, archaeology expertise (Remarks to the Author):

This paper is much improved from the first version, and a number of new models have been illustrated. My comments are focused primarily on the IMAC modeling aspect of the paper, which was newly added on account of my earlier comments. The updated results are fascinating and highly illuminating, but the authors should make some further clarification concerning their use of the IAMC model, and not just add it as simple additional hypothesis to test. This is primarily an issue of re-framing the introduction of the IMAC model to note explicitly that it is NOT a “migration” model in any sense, so adding some explanatory nuance to the mechanisms underlying the IAMC as a bridging model that can explain cultural transmission without necessary genetic transmission. This will serve then to reinforce the way it is brought up in the discussion at the end of the paper.

Comment 1: Line 114-117: This is an oversimplification the IAMC model since the IAMC, as proposed by Frachetti 2012 (and used by Li 2020) does NOT make any claims about “founder populations” for Xiaohe (or Xinjiang) or anywhere, in terms of genetics. Rather Frachetti’s argument is that the IMAC is a geographic corridor and vector for cultural influence, interaction, technology and trade, among (agro) pastoralists channeling innovations along the mountains between the Steppe, Southern Central Asia, and Xinjiang, starting in the 4th millennium BC and continuing in complex ways throughout the 3rd and 2nd millennia BC. (The authors correctly cite the model in their paper on line 320.)

We agree that Frachetti 2012 and Li 2020 do not make explicit claims about founder populations, but Betts et al. 2019 build upon this IAMC research and do specifically identify agropastoral IAMC populations as the proposed founder population for the Xiaohe horizon. We clarify in lines 104-105 that the IAMC hypothesis we test is that based on Betts et al.’s formulation of the Inner Asian Mountain Corridor island biogeography hypothesis, and we have changed “IAMC hypothesis” to “IAMC-IB hypothesis” to make this distinction clearer. See also response to Comment 6.

To avoid ambiguity and further distinguish the IAMC-IB hypothesis from other formulations, we have added the following statement to the introduction: “In contrast to these three migration models, the greater IAMC, which spans the Hindu Kush to Altai mountains, may have alternatively functioned as a geographic arena through which cultural ideas, rather than populations, primarily moved (Frachetti 2012).

We have also added a statement to the Discussion clarifying that the function of the IAMC as a geographic corridor and cultural vector, as originally described in Frachetti 2012 and later expanded by Li 2020, is supported by this study: “This finding is consistent with earlier arguments that the IAMC served as a geographic corridor and vector for regional cultural interaction that connected disparate populations from the 4th to the 2nd millennium BCE (Frachetti 2012; Li 2020).

Comment 2: Geographically, it is important to note that the IAMC extends to the Altai mountains and incorporates Afanasievo communities in the economic sense (wheat and barley have been documented at the Afanasievo site TongTian Cave, around 3200 BCE.

This is now indicated in Figure 1, where the geographic distribution of the IAMC is shown to extend to the Altai mountains.

Comment 3: New archaeological evidence suggests such interactions may have taken place even earlier, in the Neolithic with the coming of domesticated animals to the IAMC around 6000 BC in Kyrgystan (Taylor et al, 2021). Evidence for this vector of interaction in the Early and middle Bronze Age is now very robust, especially in terms of cultural practices such as farming, wheat, barley and millet grains, and other aspects of agro-pastoral economy, such as herding (Taylor et al 2021, Hermes et al 2018). Other archaeological data show, in a number of material forms from the Early Middle and Late Bronze age, that the flows of interaction are complex and multi-directional during each period – as described by Frachetti 2012. Thus, in the archaeological sense, there are undeniable data to show linkages between, for e.g. the IAMC and the Chemurchek and IAMC and later Bronze Age sites in Xinjiang, and THIS is the argument made again by Li 2020 (but he does NOT make a genetic origins argument). Neither Frachetti nor Li, in fact, make any comment about the genetic source of the people of Xinjiang, only the social, economic, and cultural connections. Thus, introducing the IAMC model, as suggested in my first review of this paper, was not intended to be a test for genetic sources per se, but as a way to understand the degree to which genetics and these cultural linkages were related. The authors actually achieve this goal quite well, and the contribution of your paper then, should not be to prove or disprove models of genetic origin, especially since the "IAMC model" was not presented as a genetic migration model, and definately not what Frachetti and later authors ever actually argued. Rather your paper can productively show how that there were different mechanisms of human sociality (meaning genetic admixture on the one hand, and cultural exchanges on the other), and these are decoupled in the context of late Bronze Age Xinjiang. In the earlier phase, as you note in the the discussion, the process is different, as there are cultural

traits that accompany the Afanasievo genetics, along with local admixture.

We believe these points are now addressed by the changes made in response to Comment 1.

Comment 4: Your data ultimately show that the genetic processes and cultural processes can be different at different times and that interaction in cultural terms does not ALWAYS move along with genetics. I would suggest that you reposition your use of the IMAC model as such, since it was not intended to be a genetic model. You have this worded better at the end of the paper, than at the beginning. Editing the hypothesis testing portion at the beginning to match with the conclusions at lines 300-350, would make your argument less dismissive and more productive in terms of how we understand the different mechanism underlying the genomic record you present for the early and later Bronze ages. In fact, in this way the IMAC model actually provides a good working framework in favor of your paper, since it provides a mechanism to explain how many cultural practices can be shared -- e.g. the proteomics data you have for pastoralism is an example of this -- WITHOUT demanding a direct genetic link among the people who are exchanging. The IMAC model fully allows for genetics and proteomics to work together better.

The original formulation of the IMAC model by Frachetti 2012 does not discuss the Tarim Basin. As described in response to Comment 1, we have more specifically clarified that the variant of the IMAC hypothesis that we are testing is the IMAC-IB hypothesis as described by Betts et al. 2019, as this variant explicitly addresses the Tarim Basin. Since its original formulation by Frachetti 2012, various aspects of the IMAC model's formulation have been supported, refuted, or modified over the past decade (e.g., Allentoft et al. 2015; Mathieson et al. 2015; Gaunitz et al. 2018; Narasimhan et al. 2019; Betts et al. 2019; Jeong et al. 2020; Taylor et al. 2021). Because of this, we have endeavored to be as precise as possible in describing and contextualizing our results. We agree that genetic and cultural processes are not always coincident, and we find that to be the case here. We have clarified in the discussion that the argument that the IMAC functioned as an important cultural corridor during the 4th through 2nd millennia BCE (as described in Frachetti 2012 and Li 2020) is supported by our study, and therefore this aspect of the original IMAC model is upheld. We believe our revised draft makes both the hypotheses that we test and the findings more explicit and clearer.

Comment 5: In the Early Bronze Age, the situation appears more complicated. There appear to be BOTH archaeological links between northern IAMC (Dali_EBA) and Chemurchek, as well as genetic diverse links within that population (as described in Jeong et al. 2020 and the current paper). Yet, these genetic links are NOT evident in your Dzhungarian EBA samples- even though they are geographically closer. This suggests a complex set of regional affiliations whereby Afanasievo communities integrated to different degrees with local communities (as you discuss this in part in lines 185-187). Again, framing the discussion of your data in this fashion will make it more explanatory for archaeologists reading your paper.

In the revised ancestry modeling presented now in our study, we show that the Chemurchek (2500-1700 BCE), an EBA pastoralist culture that succeeds the Afanasievo (3000-2600 BCE) in both the Dzungarian Basin and Altai mountains, derives from three ancestry streams: descendants of a local autochthonous population (Tarim_EMBA1), descendants of populations with high Iranian ancestry in Central Asia (Geoksyur_EN, a BMAC/IAMC proxy), and descendants of the Afanasievo (Dzungaria_EBA1). This new modeling helps to explain both the IAMC/BMAC-related ancestry previously noted in Chemurchek individuals as well as their reported cultural and genetic affiliations to Afanasievo groups. This new modeling supercedes the models we presented in Jeong et al. 2020. In that study, we were unable to detect the Afanasievo component, mainly because by then the autochthonous proxy, Tarim_EMBA, was not available, and also in part because it does not derive from the Afanasievo directly, but rather via their admixed Dzungarian descendents (Dzungaria_EBA). With our new modeling, we can confirm that the Chemurchek have a tripartite ancestry, which we have now more precisely modeled in time and space.

Comment 6: I suggest that the authors try to reword how they introduce and use the IAMC model (in distinction from the Steppe or BMAC model) as a regional geographic context for cultural exchange, and leverage it to explain how genetics and culture do not necessarily match. Again, as it was never proposed as a genetic source, but rather a geographic arena for social complexity. It should definitely NOT be coupled with the BMAC model (it is currently written with a slash, which has little logic behind it). That said, Betts et al. do look to the geography of the IAMC as a potential source for the Tarim Middle Bronze Age genetics, yet she is basing her hypothesis mostly on the cultural factors which I just noted, and she then only suggests a possible genetic link between them. Your data illustrates that a genetic affinity was not the case, that is interesting, but her model is most fundamentally an interaction model for the origins of Xiaohe cultural traits (namely agropastoralism). Indeed, she argues that much of the Xiaohe culture is likely derived from a local population. As it stands, it seems like you are disproving a model that few of us are claiming to be the case in the first place. She states: "The Xiaohe culture appears to have emerged in the Tarim Basin without obvious antecedents but, like the Qiemu'erqieke culture, it too may have absorbed elements of a pre-Bronze Age indigenous population". She also states "How the distinctive Qiemu'erqieke culture developed is still uncertain, but its ancestral derivation in the Altai is clear." Rather than simply set up her argument to dismiss it, it would be more productive to use her observations to say something like "although the IAMC appears to be a vector for cultural and economic factors evident throughout Xinjiang, the known sites from the IAMC do not provide a direct source of ancestry for later bronze Age populations in Xinjiang, these are interestingly traced to an ANE substrate that was present throughout Eurasia, likely in the early Holocene". As just explained above. It is important that the authors use and cite these archaeological works carefully, and familiarize themselves more deeply with the nuanced arguments made in them. If not, they risk misrepresentation by oversimplifying them.

The logic behind referring to IAMC and BMAC populations together (IAMC/BMAC) is that from a genetic perspective, individuals analyzed to date at agropastoral BMAC and IAMC sites are closely related to each other. For example, Eneolithic individuals from IAMC (e.g. Sarazm_EN) and BMAC

regions (e.g. Parkhai_EN), as well as later BMAC individuals, have a nearly indistinguishable genetic profile (see our Figure 2A). We refer to these two populations together only when discussing genetic modeling, not when discussing cultural traits. Thus we believe the usage of IAMC/BMAC is appropriate in these contexts.

In our analyses, we test the IAMC-IB model as it is described in Betts et al. 2019. Betts et al. 2019 describe the “original Xiaohe people” as an admixed group of “East and West Eurasian ancestral populations... (with) some traces of Indian lineages and probably some from Central Asian oases” in which the “Eurasian ancestry is more dominant”. They base this characterization partly on prior genetic work and partly on cultural traits, and they argue that the “founder population of the Xiaohe people came from among the transhumant pastoralists of the mountain ranges bordering Xinjiang to the west...the Inner Asian Mountain Corridor”, with the “western Altai and Semirech’ye as likely interim ancestral homelands”. That this involves the movement of people (“incomers”), and not just culture, is made explicit in their summary: “In summary, the broad picture within Xinjiang from the late 3rd millennium BCE shows a probable absorption of indigenous Neolithic hunter-gatherers into new agropastoral economies through the presence of incomers...” We test this explicit hypothesis by analyzing transhumant pastoralists excavated from the Semirech’ye region (Dali_EBA) and Altai (both Afanasievo and Chemurchek). We find that the ancestry profile of the founding Xiaohe population does not derive from either of these sources.

Thus, we are not setting up Betts et al.’s argument simply to dismiss it. We are taking seriously this explicit hypothesis that represents the most recently proposed explanation of the origins of the Xiaohe founding population. As the reviewer pointed out in the previous review, we would be remiss to ignore this recently published hypothesis.

The main contribution of our study is that we can clarify the genetic makeup of the Xiaohe population as a local autochthonous population (not an admixed population), allowing a fresh reinterpretation of the archaeological data. We have added the following statement to the discussion: “Although the IAMC may have been a vector for transmitting cultural and economic factors into the Tarim Basin, the known sites from the IAMC do not provide a direct source of ancestry for the Xiaohe populations. Instead, the Tarim mummies belong to an isolated gene pool whose Asian origins can be traced to the early Holocene.”

Comment 8: Line 251-266: This is an important observation but one that can be somewhat confusing from an archaeological perspective. The new proposed qpADM model uses, for example, Tarim EMBA1 as a source/significant component for modeling Botai, and other EBA steppe genomes. But this is somewhat misleading since Tarim_EMBA POST-dates Botai by nearly 2000 years, and others by 1000 yrs or more. I understand that this is an exercise to demonstrate that TARim EMBA1 can be used as proxy for an earlier, primordial ANE population on account of its isolation, but the argumentation here can appear somewhat circular to the non-specialist. Fact: The later Bronze age

Xinjiang genomes do appear to be dominated by a primordial ANE ancestry. Fact: We don't know when this widespread ANE Neolithic substrate begins to differentiate in different regions of Eurasia. We only know that ANE was shared between the Central steppe (Botai), IAMC (Dali-EBA), and (later) Xiaohe (among others). These sites span from 3500, 2600, and 1800BC respectively. So, it appears that the dominate ANE substrate begin to first diversify in the 4th-3rd millennium – specifically with the arrival of the Yamnaya/Afanasievo genetics in Altai, and Iranian Genetics in the IAMC, and yet NEITHER of these admixtures appear to impact the ANCESTRY in Xiaohe, as you note it remains isolated genetically. However, modeling the ANE component of earlier steppe groups with Tarim EMBA is a bit odd, since ideally, we would have an earlier source that matches. I am not sure how to solve this, but the issue seems to be illustrating that steppe populations with high ANE ancestry have a common (and not entirely clearly located) source population that was isolated and persisted without significant admixture ONLY in the Tarim basin region.

See response to Comment 12 for our changes in the text. We also clarify that we present models with late Pleistocene individual from the Afontova Gora site (AG3) as a temporally preceding distal ANE proxy, in parallel with the models involving Tarim_EMBA. We chose to discuss the Tarim_EMBA1 model in the main text because Tarim_EMBA1 is a better ANE proxy than AG3 for Dzungaria_EBA and other Holocene populations we discussed here. That is, Tarim_EMBA is an isolated descendent of a temporally preceding early Holocene population, whose genomes are currently unavailable, that contributed to Botai and other central Asian populations.

Comment 9: The paper ends on a note that makes a lot of sense, and it seems that the collective data presented from Xiaohe and other Tarim EMBA sites has opened the door for more conversation. Bravo!

We thank the reviewer for this positive assessment.

Comment 10: Main changes requested. 1) Reframe how the IAMC model is introduced in the start – change it from a “genetic source” to test, to a process that helps understand how the many aspects of the cultural character of the region were formed. Edit the wording and fix the citations to reflect what the IMAC model actually proposes, and differentiate its use by Frachetti 2012 from Betts et al. Betts et al cites the model as an interaction model and goes on to propose that the geography might provide a source for genetics (note, this could still be the case if we had earlier Neolithic samples reflecting the ANE substrate with pre-Iranina admixture). At present, we don't know that. As you note in the end, (line 320) The IAMC works best in this explanatory fashion and can be used to bridge the archaeology and the genetic findings.

See response to Comments 1, 4, and 6.

Comment 11: 2) Testing the IAMC sites as sources for your samples in Xinjiang has utility (thank you!) and it is important to state that IAMC sites are not good fits as sources for Xinjiang (generally), either in the EBA or MBA periods. However, you should briefly contextualize this as a case where evidence for social and cultural interaction and genetics do not fit neatly, and say a few more words about the links that do exist, namely with Qiermuekek genomes.

We have modified the Discussion to include the following statement: “the material culture and genetic profile of the Tarim mummies from ca. 2100 BCE onwards call into question simplistic assumptions about the link between genetics, culture, and language...”. We have added the Chemurchek to the Discussion: “... their extreme genetic isolation differs from the EBA Dzungarian, IAMC, and Chemurchek populations, who experienced substantial genetic interactions with the nearby populations mirroring their cultural links...”

Comment 12: 3) Clarify the new qpADM modeling section, at least to acknowledge that there is a chronological and geographic issue with the use of TarimEMBA as a source of earlier steppe genomes. At least make it clearer, so that the remodeled steppe sites are not misunderstood by non-specialists.

We have added the following clarifying statement “even though Tarim_EMBA1 postdates these populations in time” to the relevant statement:

“Interestingly, we observe that most Bronze Age and pre-Bronze Age populations with substantial ANE ancestry, such as Botai_CA from Eneolithic northern Kazakhstan, Kumsay_EBA and Mereke_MBA from western Kazakhstan, West_Siberia_N from Neolithic southern Russia, Okunevo_EMBA from the Minusinsk Basin, Chemurchek from EMBA Altai mountains, and Aigyrzhal_BA, Dali_EBA, Kanai_MBA from the IAMC region, show the highest outgroup- f_3 value with Tarim_EMBA1, suggesting that the Tarim mummies are currently the best representative of the pre-pastoralist ANE-related population that once inhabited Central Asia and southern Siberia (Extended Data Fig. 2A), even though Tarim_EMBA1 postdates these populations in time.”

Referee #2, paleogenomics expertise (Remarks to the Author):

Comment 13: I thank the authors for their responses, but have to note that the revised manuscript is only marginally improved as most of the previous comments seem rejected. The central conclusion continues to remain sound and highly fascinating. I continue to think that the use of the term indigenous is not needed and stands the risk of misrepresentation to the public, as it has a very

specific meaning in many regions of the world. I have not seen it commonly used in ancient DNA studies, but reserved for modern groups that self-identify as indigenous. Other alternatives are primordial as reviewer 1 uses, or possibly autochthonous.

We have changed the term “indigenous” to “autochthonous” throughout.

Comment 14: I continue to hold the opinion that the qpAdm analyses are key to the paper, but note that they are in the revision still not primarily done in the state-of-the-art approach of rotating populations from the outgroup/reference position to the source position. Such rotation is the primary way to let different models compete with each other. Further the authors reject the suggestion to highlight these analyses as main figures.

We have provided rotation-approach qpAdm results for Chemurchek and IAMC populations in our previous version (Supplementary Data S1F-G). In the revised manuscript, we provide rotation-approach results for the remaining groups: northern Xinjiang (Data S1D), southern Xinjiang (Data S1E), Central-South Asia (Data S1H), and pre-Bronze Age (Data S1I). We provide a new Extended Data Table 3 highlighting the key qpAdm models including results from the rotating population approach. Unfortunately, due to space constraints, it is not possible to show all of these results in the main text.

Comment 15: I also note that the authors reject generating the highest quality-standard data that is clearly possible for many samples in this material, shotgun sequencing. The current data is sufficient to answer the main hypotheses, but not the best possible data that could have been economically produced and left as legacy for the many researchers that will be interested in incorporating this information to their own studies.

We have generated high quality, genome-wide data for the individuals in this study. Using a combination of shotgun sequencing and target enrichment, we were able to generate a sufficient dataset to robustly answer the study’s genetic and archaeological questions. We understand that the reviewer would like us to perform additional data generation for the purpose of making it easier to address different questions in future studies, but this was beyond the budget and scope of this study’s project design and research questions.

Referee #3, archaeology expertise (Remarks to the Author):

Comment 16: The authors have placed much effort into addressing the many concerns raised by the referees, mine included. My initial concerns had to do with a lack of: clear statement of

significance/broad impacts/importance; contextualization of the relevant archaeology, attention given to recent population models and hypotheses related to their topic; and overall organized of the paper. The authors have provided a major reworking of the introduction, results and discussion to address these concerns to my satisfaction. They have also shifted the paper's emphasis away from language/Indo-European dispersals, which were problematic. The revised Figure 1 is also very well done and suitable for readers to follow.

We thank the reviewer for their positive assessment.

Comment 17: I just have one criticism to make. The authors offer conflicting descriptions of the burial architecture for Xiaohe and Gumugou. They state that oar-shaped planks were erected at the head of female burials and single posts erected at the head of male burials. But, then in another section of the paper the description is inverted (the oar-shaped planks are assigned to the males, and regular posts to females). While it's most often the case that the poles represent males and the paddles/oars females – sometimes these are reversed. Hence, we can't generalize, or else, keep consistent. The authors can reference the paper by Abuduresule et al. 2019 (in *The Cultures of Ancient Xinjiang, Western China: Crossroads of the Silk Roads*, Editors Betts et al.).

We have removed the word "typical" from the legend of Extended Data Fig. 1. We have updated the main text to say "...they also developed distinctive cultural elements not found among other cultures in Xinjiang or elsewhere, such as boat-shaped wooden coffins covered with cattle hides and marked by timber poles or oars..." We have added the citation to Abuduresule et al. 2019 to the main text.

Referee #4, population genetics and evolutionary genomics expertise (Remarks to the Author):

Comment 18: I believe the authors have addressed my concerns and suggested edits very well and I have no further comments. I believe this manuscript is well suited for publication in Nature.

We thank the reviewer for their positive assessment.

Reviewer Reports on the Second Revision:

Referee #1 (Remarks to the Author):

The authors have addressed all the issues raised in my previous reviews, and I recommend publishing this article in Nature.

Congratulations on a superb piece of research!

Michael Frchetti

Referee #2 (Remarks to the Author):

I am happy with the authors' replies, and have no further comments.

Referee #3 (Remarks to the Author):

The authors have given careful consideration to all concerns raised by the referees. I have no further comments and support the publication of this manuscript in Nature.